# Learning representations of chromatin contacts using a recurrent neural network identifies genomic drivers of conformation

Kevin B. Dsouza[1✉], Alexandra Maslova[2], Ediem Al-Jibury [3,4], Matthias Merkenschlager[3], Vijay K. Bhargava[1] & Maxwell W. Libbrecht [2✉]

Despite the availability of chromatin conformation capture experiments, discerning the relationship between the 1D genome and 3D conformation remains a challenge, which limits our understanding of their affect on gene expression and disease. We propose Hi-C-LSTM, a method that produces low-dimensional latent representations that summarize intra-chromosomal Hi-C contacts via a recurrent long short-term memory neural network model. We find that these representations contain all the information needed to recreate the observed Hi-C matrix with high accuracy, outperforming existing methods. These representations enable the identification of a variety of conformation-defining genomic elements, including nuclear compartments and conformation-related transcription factors. They furthermore enable in-silico perturbation experiments that measure the influence of cis-regulatory elements on conformation.

[1] Department of Electrical and Computer Engineering, University of British Columbia, Vancouver, Canada. [2] School of Computing Science, Simon Fraser University, Burnaby, Canada. [3] MRC, London Institute of Medical Sciences, Institute of Clinical Sciences, Faculty of Medicine, Imperial College London, London, UK. [4] Department of Computing, Imperial College London, London, UK. ✉email: kevin@ece.ubc.ca; maxwl@sfu.ca

The organization of the genome in 3D space inside the nucleus is important to its function. Chromosome conformation capture (3C) techniques, developed in the last couple of decades, have enabled researchers to quantify the strength of interactions between loci that are nearby in space. Hi-C[1] uses a combination of chromatin conformation capture and high-throughput sequencing to assay pairwise chromatin interactions genome-wide. This rich source of data promises to help elucidate the influence of 3D structure on gene expression and thereby on development, evolution and disease. However, we lack a complete understanding of how the 1D genome influences 3D conformation.

The machine learning technique of representation learning[2] provides a way to link the 1D genome to 3D conformation. Representation learning aims to summarize high dimensional datasets into a low-dimensional representation. It has become a valuable tool for finding compact and informative representations that disentangle explanatory factors in diverse data types. Representation learning has recently driven advances in a variety of tasks including speech recognition[3], signal processing[4], object recognition[5], natural language processing[6,7] and domain adaptation[8]. Representation learning has recently been applied to genomic sequences[9,10] and Hi-C data[11–14].

In order to understand the 1D–3D relationship and thereby link 3D conformation to genetic variation and disease, we need representations for Hi-C data that can summarize the contact map into a locus-level summary. Such a representation would encapsulate all the contacts from each genomic position to the others into a small number of features per locus, such that the contacts can be reproduced using just the features. Reducing the Hi-C map to locus-level representations in this way would allow us to study the effect of sequence elements on chromatin conformation, identify genomic drivers of 3D conformation and predict the effect of genetic variants.

Two methods for representation learning of Hi-C data have previously been developed, SNIPER[11] and SCI[12]. SNIPER uses a fully connected autoencoder[15] to transform the sparse Hi-C inter-chromosomal matrix into a dense one row-wise, the bottleneck of which is assigned as the representation for the corresponding row. SCI[12] treats the Hi-C matrix as a graph and performs graph embedding[16], aiming to preserve the local and the global structures to form representations for each node.

These existing methods for Hi-C representations have two weaknesses that limit their applicability. First, SNIPER takes only inter-chromosomal contacts as input and therefore its representations cannot incorporate intra-chromosomal contact patterns that are most important for the regulation of gene expression, such as topological domains and promoter-enhancer looping. Second, the Hi-C representations produced by both SNIPER and SCI do not account for the inherent sequential nature of the genome. As we demonstrate in this work, these two weaknesses limits existing methods' informativeness and makes them unable to accurately identify conformation-defining elements or predict how those elements influence structure.

Hi-C-LSTM primarily forms Hi-C representations. Learning methods like SNIPER[11] and SCI[12] have been proposed that can form representations of Hi-C. SNIPER forms Hi-C representations using a feed-forward neural network autoencoder. While SNIPER predicts high-resolution Hi-C contacts using low-resolution contacts as input, Hi-C-LSTM predicts Hi-C contacts using just the genomic positions as input. SCI forms Hi-C representations by performing graph network embedding on the Hi-C data. SCI is similar to Hi-C-LSTM in that it can be used to identify elements, however, it differs in the underlying structure it uses to represent the genome. SCI represents the genome using a graph, whereas Hi-C-LSTM treats the genome as a sequence. We compare Hi-C-LSTM with these two methods as they are most similar to what we are trying to achieve.

The first Hi-C representations were formed using principal component analysis (PCA)-based methods, introduced in Lieberman-Aiden et al.[1]. These methods cluster the Hi-C matrix into A and B compartments based on the first principal component of the intra-chromosomal contact matrix. Imakaev et al.[17] later showed that PCA-based reduction is inaccurate at classifying compartments and Rao et al.[18] used a Gaussian hidden Markov model (HMM) to obtain latent features that were better at locating compartments. We treat the PCA-based method developed in Lieberman-Aiden et al.[1] as a baseline.

Some methods form chromatin representations but are not directly comparable to ours. REACH-3D[19] forms internal Hi-C representations using manifold learning combined with recurrent autoencoders, however, these are three dimensional and mainly used for 3D chromatin structure inference. MATCHA[14] forms representations using hypergraph representation learning and uses them to distinguish multi-way interactions from pairwise interaction cliques. We do not compare Hi-C-LSTM with MATCHA because MATCHA works with multi-way interaction data (SPRITE and ChIA-Drop) whereas we use pair-wise interaction data (Hi-C).

Another related task is that of imputing unseen Hi-C data sets, for which several methods have been developed. Such imputation methods include SNIPER[11], DeepHiC[20], HiCPlus[21], Higashi[22], and scHiCluster[23]. SNIPER imputes high-coverage Hi-C using moderate-coverage Hi-C at the input. DeepHiC predicts high-resolution Hi-C contact maps from low-coverage sequencing data using generative adversarial networks. HiCPlus infers high-resolution Hi-C matrices from low-resolution Hi-C data using deep convolutional neural networks. Both DeepHiC and HiCPlus, do not form position specific representations that accomplish various downstream tasks, therefore, are not comparable to our method. Higashi enhances scHi-C data quality using hypergraph representation learning. scHiCluster studies cell type-specific chromosome structural patterns in scHi-C. These methods (apart from SNIPER) cannot be used for the task of bulk Hi-C representation learning because they do not form position-specific representations and, in the case of Higashi and scHiCluster, require single-cell data.

Note that, while existing methods for Hi-C representation learning (including SCI, SNIPER and Hi-C-LSTM) utilize a reconstruction loss that aims to reconstruct the input Hi-C data, they cannot in general be used for imputation.

Many methods have been proposed for predicting Hi-C contacts. Some methods try to predict the chromatin contacts by using either the nucleotide sequence or chromatin accessibility and histone modifications or both[24–31]. Akita in particular[31], is a convolutional neural network that predicts chromatin contacts from the nucleotide sequence alone, and can be used to perform in-silico predictions. In addition to these, the maximum entropy genomic annotation from biomarkers associated to structural ensembles (MEGABASE) coupled with an energy landscape model for chromatin organization called minimal chromatin model (MiChroM), generates an ensemble of 3D chromosome conformations[32]. Though these methods are similar to Hi-C-LSTM in that they predict Hi-C contacts, we do not compare Hi-C-LSTM with them as none of them produce Hi-C representations.

In addition, many approaches have been developed to identify genomic features, such as histone modifications or other ChIP-seq measurements, that influence chromatin conformation[33–38]. This task is similar to conformation representation learning in that it links 1D to 3D genomic features. However, using histone modifications as a summary of the chromatin-defining features of a given locus may not fully encapsulate the conformation.

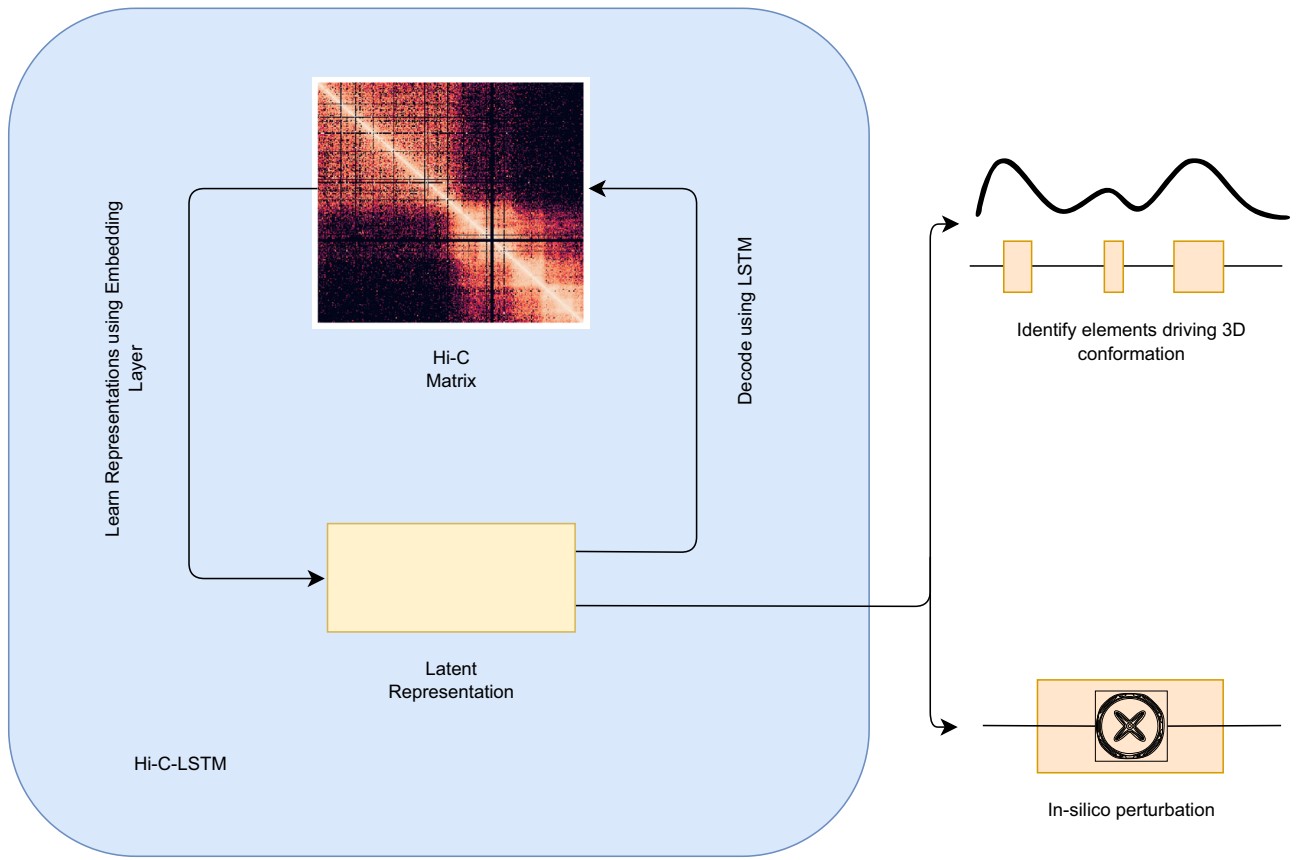

**Fig. 1 Overview of approach.** Hi-C-LSTM learns a *K*-length vector representation of each genomic position that summarizes its chromatin contacts, using an LSTM embedding neural network. The representations and LSTM decoder are jointly optimized to maximize the accuracy with which the decoder can reproduce the observed Hi-C matrix given just the representations. The resulting representations identify sequence elements driving 3D conformation through integrated gradients (IG) analysis, and they enable a researcher to perform in-silico perturbation experiments by editing the representations and observing the effect on predicted contacts.

In this work, we propose a method called Hi-C-LSTM that produces low-dimensional representations of the Hi-C intra-chromosomal contacts, assigning a vector of features to each genomic position that represents that position's contact activity with all other positions in the given chromosome. Hi-C-LSTM defines these representations using a sequential long short-term memory (LSTM) neural network model which, in contrast to existing methods like SNIPER and SCI, accounts for the sequential nature of the genome. A second methodological innovation of Hi-C-LSTM is that, instead of learning an encoder to create representations, we learn our representations directly through iterative optimization. We find that this approach provides a large improvement in information content relative to existing non-sequential methods, enables the use of intra-chromosomal interactions, and enables the model to accurately predict the effects of genomic perturbations (Fig. 1, see the section "Results").

We demonstrate the utility of Hi-C-LSTM's representations through several analyses. First, we show that our representations have information needed to recreate the Hi-C matrix and that this recreation is more accurate using an LSTM than alternatives. Second, we show that our representation captures cell type-specific functional activity, genomic elements, and regions that drive conformation. Third, we show that feature attribution of Hi-C-LSTM can identify sequence elements driving 3D conformation, such as binding sites of CTCF and Cohesin subunits[39,40]. Fourth, we show Hi-C-LSTM can perform in-silico perturbation of CTCF and Cohesin binding sites. Fifth,

we simulated a previously assayed 2.1 Mbp structural variant at the SOX9 locus and found that Hi-C-LSTM correctly reproduces experimentally derived contacts.

## Results

**Hi-C-LSTM representations capture the information needed to create the Hi-C matrix.** Hi-C-LSTM assigns a representation to each genomic position in the Hi-C contact map, such that a LSTM[41] that takes these representations as input can predict the observed contact map (Fig. 2). The representation and the LSTM are jointly trained to optimize the reconstruction of the Hi-C map. This process gives us position-specific representations genome-wide (see the "Methods" section for more details).

We find that Hi-C-LSTM achieves higher accuracy when constructing the Hi-C matrix compared to existing methods (Fig. 3a, c). The inferred Hi-C map matches the observed Hi-C map (Fig. 3g) closely, and differs from it by about 0.25 R-squared points on average. We adapt SNIPER to our task by replacing the feed-forward decoder that converts low-resolution Hi-C to high-resolution Hi-C with a decoder that reproduces the observed input Hi-C. We call this SNIPER-FC. Hi-C-LSTM outperforms SNIPER (SNIPER-FC) convincingly, by 10% higher *R*-squared on average (Fig. 3a). Hi-C-LSTM also outperforms SCI (SCI-LSTM) by 12% higher *R*-squared on average (Fig. 3a).

Two hypotheses could explain Hi-C-LSTM's improved reconstructions: (1) that Hi-C-LSTM's representation captures more information, or (2) that an LSTM is a more powerful decoder.

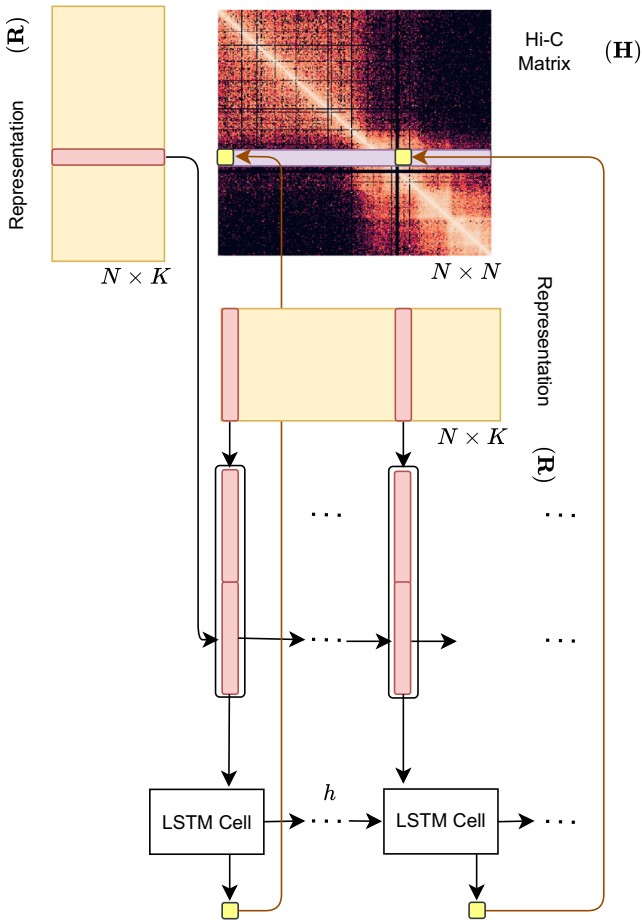

**Fig. 2 Overview of the Hi-C-LSTM model.** A trained Hi-C-LSTM model consists of a $K$-length representation $\mathbf{R}_i$ for each genomic position $i$ and LSTM connection weights (see the "Methods" section). To predict the contact vector of a position $i$ with all other positions, the LSTM iterates across the positions $j \in \{1...N\}$. For each $(i, j)$ pair, the LSTM takes as input the concatenated representation vector ($\mathbf{R}_i \, \mathbf{R}_j$) and outputs the predicted Hi-C contact probability $H_{i,j}$. The LSTM hidden state $h$ is carried over from $(i, j)$ to $(i, j + 1)$. This process is repeated for all $N$ rows of the contact map by reinitializing the LSTM states. The LSTM and the representation matrix are jointly trained to minimize the reconstruction error.

We found that both are true. To distinguish these hypotheses, we split each model, respectively, into two components—its representation and decoder—and evaluated each possible pair of components. We train the representations (Hi-C-LSTM, SCI, SNIPER) on all chromosomes and couple them with selected decoders (LSTM, CNN, FC). Using the representations as input, we re-train these decoders with a small subset of the chromosomes and test on the rest. (see the "Methods" section for more details). We compute the average $R$-squared value for creating the Hi-C contact matrix using each combination of selected representations and decoders

We found that the choice of decoder has the largest influence on reconstruction performance. Using a LSTM decoder performs best, even when using representations derived from SNIPER or SCI (improvement of 0.14 and 0.12 $R$-squared points on average over fully connected decoders, respectively, Fig. 3a). Furthermore, we found that Hi-C-LSTM's representations are most informative, even when using decoder architectures derived from SNIPER or SCI (Fig. 3a).

Though the Hi-C-LSTM representations capture important information from a particular sample, we wanted to verify

whether they capture real biological processes or irreplicable experimental noise. To check the effectiveness of Hi-C-LSTM representations in creating the Hi-C contact map of a biological replicate, we train the representations on one replicate (replicate 1), repeat the decoder training process on replicate 2 (see the "Methods" section for more details), and compute the average $R$-squared value for creating the Hi-C contact map of replicate 2 (Fig. 3b, d). The average $R$-squared reduces slightly for inference of replicate 2 due to experimental variability; however, the performance trend of the representation–decoder combinations is largely preserved (Fig. 3b, d). These results show that Hi-C-LSTM's improved performance is not merely driven by memorizing irreplicable noise.

We discovered a relationship between sequencing depth and model performance after training and evaluating Hi-C-LSTM on Hi-C datasets from GM12878 with a combined filtered reads of 300 million and 216 million (compared to Hi-C data from Rao et al. which had 3 billion combined filtered reads). We saw that Hi-C-LSTM $R^2$ worsened with reduced read depth, however, the reconstruction performance trend over distance was preserved (Fig. 3e, f).

We also trained and evaluated Hi-C-LSTM on data from 3 other tier 1 cell types from the 4DN Data Portal apart from GM12878, namely, H1-hESC (embryonic stem cell) (Fig. 3c, d), HFF-hTERT (foreskin fibroblast immortalized cell) (Supplementary: Fig. 1a, b), and WTC11 (induced pluripotent stem cell derived from skin leg fibroblast) (Supplementary: Fig. 1c, d). We found that difference in performance across these cell types can be explained by their differing read depths. These data sets have varying read depths, ranging from 150 million (HFFhTERT) to 900 million (H1hESC) filtered reads. We saw that the $R^2$ fell by 0.03 points on average when reads reduced from 3 billion to 1 billion (Fig. 3e). The performance further reduced by 0.4 on average when the reads reduced to 150 million. This amounted to a total $R^2$ decrease of 0.7 points on average with very low sequencing depth (Fig. 3e). We additionally found that the reconstruction performance trend between models is preserved across cell types (Supplementary: Fig. 1).

**Hi-C-LSTM representations locate functional activity, genomic elements, and regions that drive 3D conformation.** Considering that a good representation of Hi-C should contain information about the regulatory state of genomic loci, we evaluated our model by checking whether these genomic phenomena and regions are predictable from only the representation. Specifically, we test whether the position specific representations learned via the Hi-C contact-generation process are useful for genomic tasks that the model was not trained on, such as classifying genomic phenomena like gene expression[42] and replication timing[43–46], locating nuclear elements like enhancers, transcription start sites (TSSs)[47] and nuclear regions that are associated with 3D conformation like promoter–enhancer interactions (PEIs)[48–50], frequently interacting regions (FIREs)[51,52], topologically associating domains (TADs), subTADs, and their boundaries[18], loop domains and subcompartments[18]. We compared two classifiers, a boosted decision tree (XGBoost) model[53] to predict binary genomic features of GM12878 from representations, for each task separately, and a multi-class multi-label model with a simple linear layer and sigmoid, to predict all tasks from the representations simultaneously (see the "Methods" section for more details regarding comparison methods, baselines and classifiers).

We use mean average precision (mAP) (see the "Methods" section) to quantify classification performance (for additional classification metrics like area under the receiver operating characteristic curve (AuROC), $F$-score, and Accuracy, refer to the

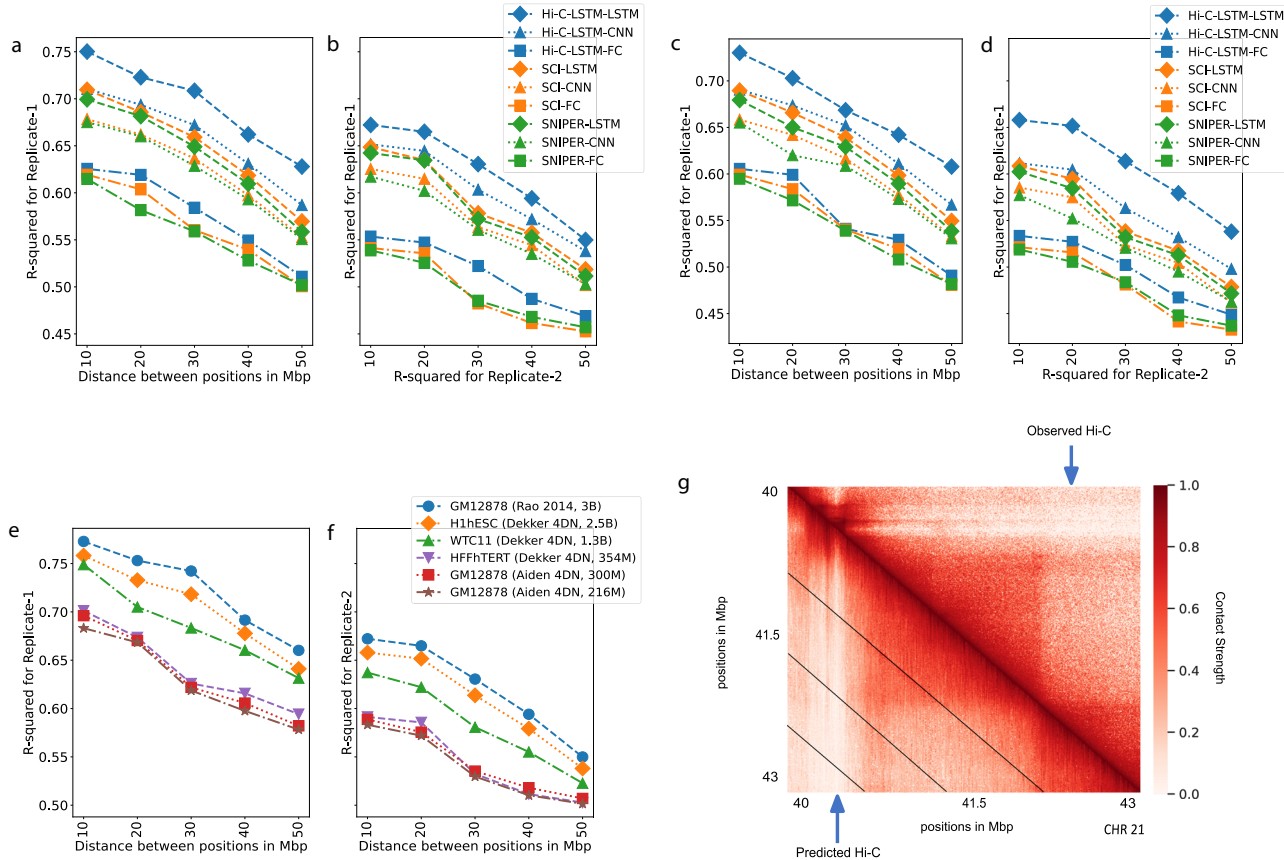

**Fig. 3 Accuracy with which representations reproduce the observed Hi-C matrix. a, b** The Hi-C $R$-squared computed using the combinations of representations from different methods and selected decoders for replicate 1 and 2 (GM12878). The horizontal axis represents the distance between positions in Mbp. The vertical axis shows the reconstruction accuracy for the predicted Hi-C data, measured by average $R$-squared. The $R$-squared was computed on a test set of chromosomes using selected decoders with the representations trained all chromosomes as input. The legend shows the different combinations of methods and decoders, read as *[representation]-[decoder]*. **c, d** Same as (**a, b**), but for H1-hESC. **e, f** Hi-C $R$-squared computed for different cell types. **g** A selected portion of the observed Hi-C map (upper-triangle) and the predicted Hi-C map (lower-triangle) in GM12878. The portion is selected from chromosome 21, between 40 and 43 Mbp. Diagonal black lines denote Hi-C-LSTM's frame boundaries (see the "Methods" section).

Supplementary, see the "Methods" section for definitions). We find that the models built using the intra-chromosomal representations achieve higher classification performance overall relative to ones trained on inter-chromosomal representations when predicting gene expression, enhancers and TSSs (Fig. 4a, b). This trend is likely due to the relatively close range of the elements involved in prediction. We verify this observation by running Hi-C-LSTM at different resolutions (see the section "Resolution"). In contrast, SNIPER is slightly better at predicting replication timing when compared to the rest of the intra-chromosomal models except Hi-C LSTM (SNIPER-INTER, Fig. 4a, b). While all methods achieve low absolute scores at predicting promoter–enhancer interactions, Hi-C-LSTM performs best (0.5 units on average, 0.1 unit higher on average than SCI) (Fig. 4a, b, d).

Both methods perform comparably in predicting the other interacting genomic regions like FIREs, TADs, subTADs, loops domains, and subcompartments (Fig. 4a, b). SNIPER-INTRA as well as SNIPER-INTER do not perform as well as Hi-C-LSTM and SCI on these tasks. One caveat of the model is that it loses CTCF interaction dots at loop boundaries because of its sequential prediction streaks (Supplementary Fig. 2).

The only task on which other methods outperform Hi-C-LSTM is at predicting subcompartments. Subcompartments were originally defined based on inter-chromosomal interactions, so representations based on such interactions outperform those

based on intra-chromosomal interactions such as Hi-C-LSTM (see Supplementary: Fig. 3 for confusion matrix). Also subcompartment-ID (SBCID) methods achieves perfect mAP by virtue of its design (Fig. 4a, b). Among the rest of the methods, we find that methods which were designed to predict subcompartments such as SCI and SNIPER-INTER, perform better than the others (Fig. 4a, b). Hi-C-LSTM does perform marginally better than SNIPER-INTRA. Overall, although Hi-C LSTM performs better than other models on most of the tasks, the performance of SCI and SNIPER are comparable to Hi-C-LSTM and all three models perform significantly better than the baselines on average (Fig. 4a, b).

Similar to reconstruction, when comparing classification performance across cell types, we saw that Hi-C-LSTM accuracy worsened with reduced read depth. However, the classification performance trend over tasks was preserved (Fig. 4c). We include results for all available data sets for each cell type. We omitted WTC11 from this analysis because most data sets are not available (see the "Methods" section for details regarding element specific data in cell types). We observed that the accuracy reduced by 0.6 units on average when the reads reduced to 150 million (Fig. 4c). Next, we compared the classification performance of Hi-C-LSTM with other methods (SCI, SNIPER) and baselines (PCA, SUBCOMPARTMENT-ID) in these cell types (Supplementary Figs. 4–6). Similar to $R^2$, we saw that the prediction score trend of methods is preserved across all these cell types.

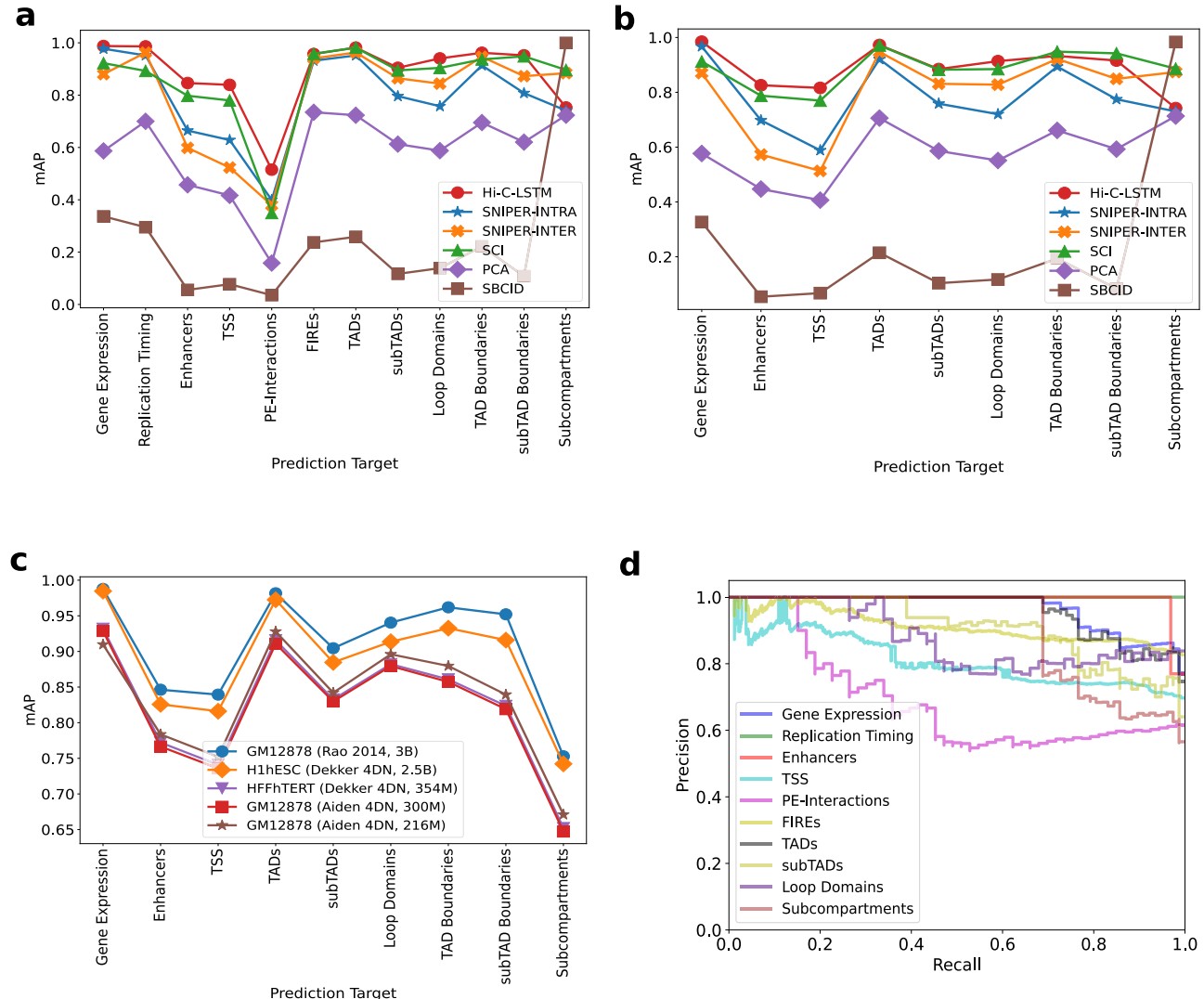

**Fig. 4 Genomic phenomena and chromatin regions are classified using the Hi-C-LSTM representations as input. a** Prediction accuracy for gene expression, replication timing, enhancers, transcription start sites (TSSs), promoter–enhancer interactions (PEIs), frequently interacting regions (FIREs), loop and non-loop domains, and subcompartments in GM12878. The *y*-axis shows the mean average precision (mAP), the *x*-ticks refer to the prediction targets, and the legend shows the different methods compared with. **b** Same as (**a**), but for targets in H1-hESC. **c** mAP using Hi-C-LSTM for targets compared across cell types. **d** The Precision-Recall curves of Hi-C-LSTM for the various prediction targets in GM12878. The *y*-axis shows the Precision, the *x*-axis shows the Recall, and the legend shows the prediction targets.

Understanding what kind of interactions the model is more likely to capture is vital. TADs are identified with a higher accuracy in all cell types compared to other larger chromatin structures like subcompartments (Fig. 4a, b; Supplementary: Fig. 4–6). On the other hand, Promoter-Enhancer interactions are hard to classify in all cell types (Supplementary: Fig. 4,5,6). This means that Hi-C-LSTM representations achieve higher accuracy for medium-scale structures such as TADs than for small-scale structures like promoter–enhancer interactions. This could be due to many factors including data resolution, model architecture, and conservation across cell types.

**Hi-C-LSTM recapitulates structures at different Hi-C resolutions**. To check if Hi-C-LSTM works at different resolutions of Hi-C data, in addition to our model trained at 10 kbp, we trained Hi-C-LSTM at three other resolutions of 2, 100, and 500 kbp. As expected, models at different scales detect different elements, classify differently, and attribute importance to different sites

depending on the resolution. The models achieved these by forming representations that allowed them to construct the Hi-C map at the given resolution. We investigate how these representations differ from the ones learned at 10 kbp.

To train the model at 2 kbp, we used only a subset of chromosomes due to memory and compute constraints but trained on the whole genome at other resolutions. A selected portion in chromosome 21 (Fig. 5a) shows that the predicted Hi-C values capture the fine structure of Hi-C even at 2 kbp resolution. The sparsity of available data at 2 kbp is a major constraint in enhancing the performance of the model at this resolution. Hi-C-LSTM captures the Hi-C macro-structures accurately at 500 kbp (Fig. 5b) and 100 kbp (Fig. 5c). This is because operating at this resolution with our sequence length allows it to span entire smaller chromosomes.

We found that representations at different resolutions predict chromatin structures of different scales. The classification performance (as measured in mAP) with gene expression, enhancers, TADs, subTADs, and subcompartments of models trained at different resolutions (Fig. 5d), shows that for small

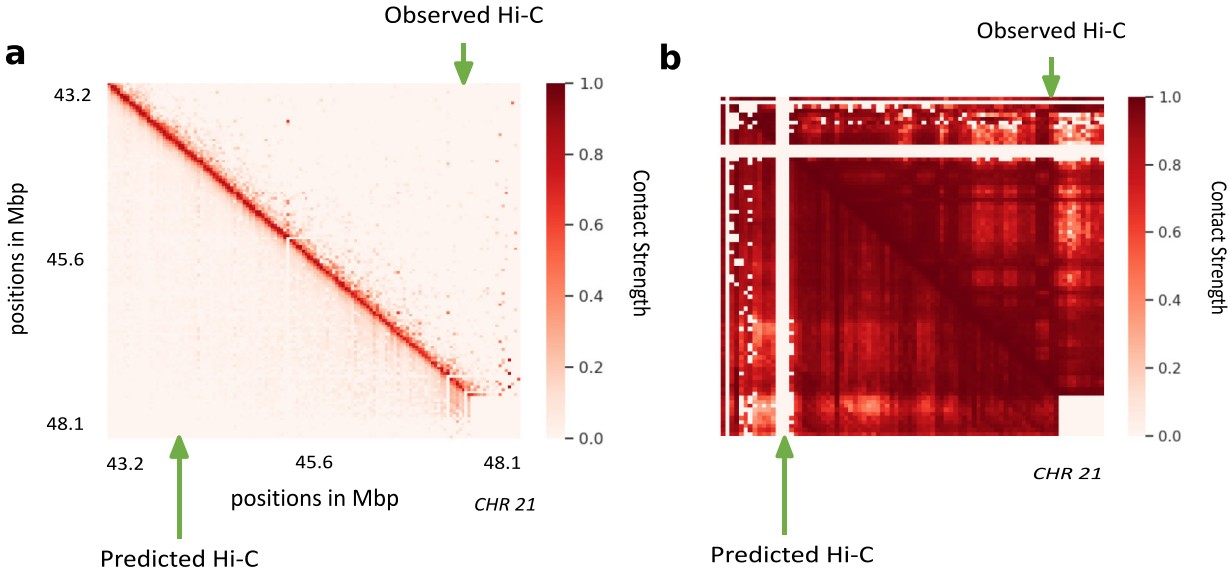

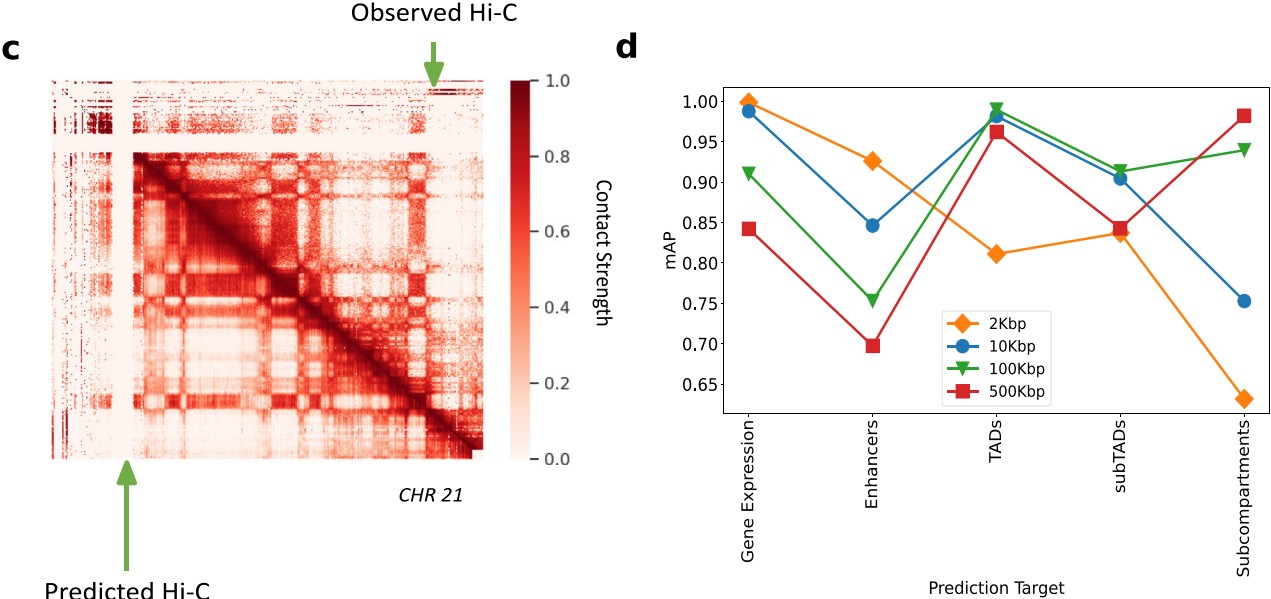

**Fig. 5 Hi-C-LSTM applied at different resolutions. a** Hi-C-LSTM predictions at 2 kbp resolution. A selected portion of the observed Hi-C map (upper-triangle) and the predicted Hi-C map (lower-triangle) in GM12878. The portion is selected from chromosome 21, between 43.2 and 48.1 Mbp. **b** Hi-C-LSTM predictions at 500Kbp resolution. The observed Hi-C map (upper-triangle) and the predicted Hi-C map (lower-triangle) in GM12878 for chromosome 21. **c** Hi-C-LSTM predictions at 100 kbp resolution. The observed Hi-C map (upper-triangle) and the predicted Hi-C map (lower-triangle) in GM12878 for chromosome 21. **d** The classification performance (as measured in mAP) with gene expression, enhancers, TADs, subTADs, and subcompartments of models trained at different resolutions.

scale phenomena and expression like gene expression and enhancers, the cumulative prediction score worsens with coarser resolution as expected. For enhancers, the prediction score drops by 0.22 units when the resolution goes from 2 to 500 kbp (Fig. 5d). Both with TADs and subTADs, the model at 100 kbp has the best prediction score, closely followed by the model at 10 kbp (Fig. 5d). We attribute this performance to the fact that these resolutions, combined with our frame length of 150, are close to the to the averages sizes of both these elements. The model at 500 kbp performs best at identifying subcompartments given that the average size of subcompartments is 300 kbp

(Fig. 5d). These results point to the idea that aggregating representations learnt at different Hi-C resolutions will likely increase prediction performance across elements of all sizes. Such aggregation will also potentially help in alleviating computational bottlenecks, as a model at a particular resolution need not take the broader context into account (see the section "Discussion").

**Feature attribution reveals association with genomic elements driving 3D conformation.** Given that our representations capture elements driving 3D conformation, we should be able to identify

those elements using our representations. To validate the ability of our representations to locate genomic regions that drive chromatin conformation, we identified which genomic positions have the largest impact on Hi-C contacts, using the technique of feature attribution. Feature attribution is a technique that allows us to attribute the prediction of neural networks to their input features. In this case, it identifies which genomic positions influence which Hi-C contacts. We ran feature attribution analysis on the Hi-C-LSTM and aggregated the feature importance scores across all the dimensions of the input representation to get a single score for each genomic position (see the "Methods" section for more details). We expected to see higher feature attribution for the genomic elements, regions, domains, and transcription factors (TFs) that are crucial for chromatin conformation.

The variation of the aggregated feature importance across interesting genomic regions helps us distinguish boundaries of domains and genomic regulatory elements (Fig. 6a, b). We observe the variation of the feature importance signal across

TADs and a selected portion of chromosome 21 (28–29.2 Mbp)[54] to check if we can isolate the boundaries of domains, genes and other regulatory elements. To deal with TADs of varying sizes, we partition the interior of all TADs into 10 equi-spaced bins and average the feature importance signal within these bins. We plot this signal along with the signal outside the TAD boundary 50 kbp upstream and downstream, averaged across all TADs (Fig. 6a). The feature importance has largely similar values in the interior of the TAD, noticeably peaks at the TAD boundaries, and slopes downward in the immediate exterior vicinity of the TAD (Fig. 6a). This trend validates the importance of TADs and TAD boundaries in chromatin conformation. We also consider a candidate region in chromosome 21 that is referred to in ref. [54] to observe the variation of feature importance across active genomic elements (Fig. 6b). For this selected region in chromosome 21, as we do not have to deal with domains of varying sizes, we just average the feature importance signal within a specified number of bins and plot this in the UCSC Genome Browser along with

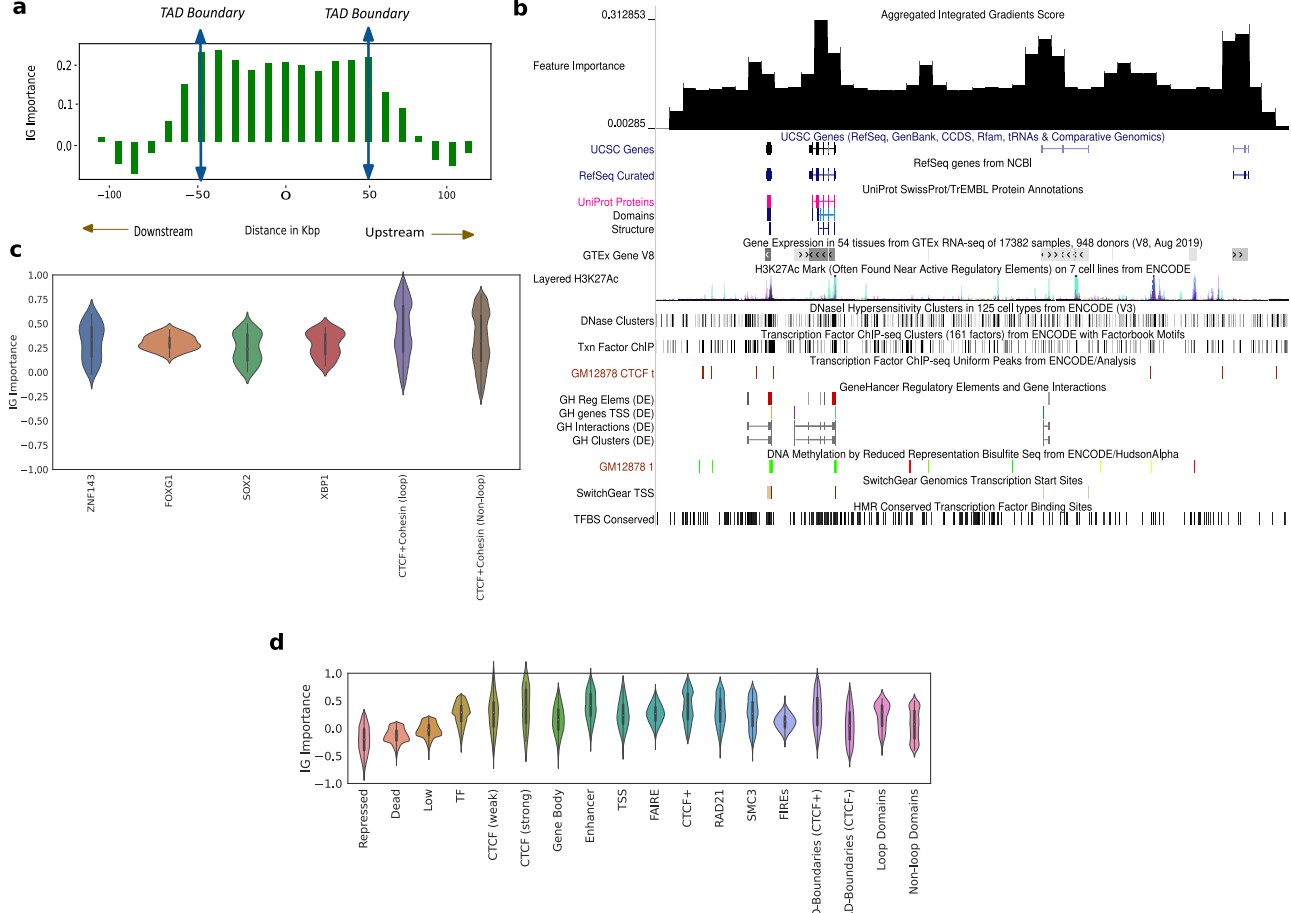

**Fig. 6 Hi-C-LSTM representations identify genomic elements involved in conformation through integrated gradients (IG) feature importance analysis.**
**a** The IG feature importance averaged across different TADs of varying sizes. The vertical axis indicates the average IG importance at each position and the horizontal axis refers to relative distance between positions in kbp, upstream/downstream of the TADs. **b** The IG feature importance for a selected genomic locus (chr21 28–29.2 Mbp) along with genes, regulatory elements, GC percentage, CTCF signal, and conserved TFBS among others in the UCSC genome browser. We see that the feature importance scores peak at known regulatory elements, higher GC percentage, and CTCF peaks. **c** Violin plots of aggregated feature attribution scores for top ranked transcription factor binding sites (TFBS). The x-axis shows the labels/elements and the y-axis displays the z normalized feature importance scores from Integrated Gradients. Both at loop and non-loop regions, the scores shown are aggregated only at shared sites. **d** Violin plots of aggregated feature attribution scores for selected elements. The x-axis shows the labels/elements and the y-axis displays the z normalized feature importance scores from Integrated Gradients. The scores for CTCF and Cohesin subunits are aggregated genome wide. In **c**, **d**, Violin plots present summary statistics where the white dot is the median, thick gray bar is the inter-quartile range, and thin gray line is the rest of the distribution. Kernel density estimation is shown on either side of the line. Sample size for the genomic elements are calculated genome wide by considering all observations of elements according to element specific data.

genes, regulatory elements, GC percentage, CTCF signal, and conserved TFBS among others. The feature importance peaks around genes, regulatory elements and domain boundaries (Fig. 6b), showing that they play a more important role in conformation than other functional elements. The feature importance peaks also correlate with CTCF peaks and GC percentage peaks (Fig. 6b).

We analyzed importance scores at TF binding sites (TFBS)[55] and saw that some TFBS have a larger positive importance score compared to others (Fig. 6c). Our motif enrichment analysis showed that the top 5 TFs according to importance score were: CTCF, ZNF143, FOXG1, SOX2, and XBP1 (Fig. 6c). As Cohesin is a known partner of CTCF, we looked for Cohesin-binding sites in the ranked list and found them in the top 15. The full ranked list of transcription factors is attached as a Supplementary file. All TFs in the top 5 are known to play a role in chromatin conformation. The genome folds to form "loop domains", which are found to be a result of tethering between two loci bound by CTCF and Cohesin subunits RAD21 and SMC3[40]. Among the many models of genome folding, Cohesin ring-associated complex that extrudes chromatin fibers and is delimited by CTCF is most promising. This extrusion model explains why loops do not overlap[39].

We found that CTCF + Cohesin sites at loop anchors show 10% higher mean importance score than CTCF + Cohesin sites at non-loop regions (we only considered the case where CTCF and Cohesin share sites) and in both cases they have a spread that is predominantly positive (Fig. 6c). Note that CTCF and Cohesin sites usually overlap, so we analyze them together. Specifically, 98% of loop anchor CTCF ChIP-seq peaks also harbor Cohesin peaks; 92% non-loop CTCF peaks do so[56,57]. The high feature importance scores observed at CTCF and Cohesin-binding sites reaffirms the crucial role they play in loop formation[39,40]. The importance of CTCF is further validated by the aggregated feature importance (Fig. 6d), showing a markedly positive score near CTCF-binding sites given by Segway[58], particularly the strong ones (mean importance score of 0.45).

Apart from CTCF, the other TFs in the top 5 are also known to play a role in conformation (Fig. 6c). There is a widespread role of C2H2-ZF proteins in chromatin structure and organization[59]. These TFs are known to promote local chromatin loosening, local chromatin condensation[60], and control chromatin accessibility through the recruitment of chromatin-modifying enzymes[59]. ZNF143 (2nd-most important) is a C2H2-ZF protein. It is known to bind directly to promoters, connect promoters to distal regulatory enhancers[61], and plays a partner role in establishing conserved chromatin loops[61]. Similarly, many FOXG1 (3rd-most important) and related TFs are considered pioneer factors which open closed chromatin and facilitate the binding of other TFs[62,63]. The last two TFs in our top 5, SOX2 and XBP1, are also known to play a role in conformation. SOX2 loss is seen to decrease chromatin interactivity genome-wide[64], and the genomic distribution of XBP1 peaks shows that it binds promoters and potential enhancers[65,66].

Along with the aforementioned TFs, we saw that the model places high importance on regulatory elements, particularly enhancers (mean importance score of 0.4) (Fig. 6d). The active domain types had a higher mean score and a spread that largely occupies the positive portion of the feature importance plot when compared to the inactive regions (Fig. 6d). This is further verified by segway-gbr[67] feature importance scores (Supplementary Fig. 7). This suggests that active regions may play a dominant role in nuclear organization, where the movement of repressed regions to the periphery is a side-effect.

Aggregated feature importance also demonstrates the largely positive feature attribution of genomic regions that are an integral part of 3D conformation like FIREs, topologically associating domain (TAD) boundaries with and without CTCF sites, loop and non-loop domains (Fig. 6d). TAD boundaries enriched with CTCF show a 20% higher mean importance score compared to TAD boundaries not associated with CTCF, pointing to the importance of CTCF sites at domain boundaries in conformation (Fig. 6d). Moreover, loop domains show a 20% higher mean importance score compared to non-loop domains, which is expected because of the increased contact strength on average and the presence of CTCF sites (Fig. 6d). P-values from the relevant comparisons for each group can be referred to in the Supplementary: Table 1.

**Hi-C-LSTM accurately predicts effects of a 2.1 Mbp duplication at the SOX9 locus.** To validate Hi-C-LSTM as a tool for in-silico genome alterations, we simulated a structural variant at the SOX9 locus that was previously assayed by Melo et al. [68]. This variant was observed in an individual with Cook's syndrome and comprises the tandem duplication of a 2.1 Mbp region on chromosome 17 that includes regulatory elements of SOX9 (chr17:67,958,880–70,085,143; GRCh37/hg19, Fig. 7a). To simulate a Hi-C experiment on a genome with this variant, we made a new Hi-C-LSTM representation matrix that includes a tandem copy of the representation at the locus in question and passed this representation matrix through the original Hi-C-LSTM decoder to produce a simulated Hi-C matrix on a post-duplication genome (Fig. 7b). Because Hi-C reads cannot be disambiguated between the two duplicated loci, we simulated mapping reads to the observed hg19 reference by summing reads originating from the two copies (see the "Methods" section). We evaluated Hi-C-LSTM's predictions according to the agreement between this predicted matrix and a Hi-C experiment performed by Melo et al. [68] (Fig. 7c).

We found that Hi-C-LSTM accurately predicted the effect of the duplication. The domains that existed pre-duplication ($D_1$, $D_2$, $D_3$, Fig. 7a) are correctly captured post-duplication. In addition, a new chromatin domain ($D_{New}$) that was introduced by the duplication is correctly predicted by Hi-C-LSTM (Fig. 7b). To quantitatively evaluate our predictions, we compared them to a baseline that predicts the observed pre-duplication Hi-C for the interactions between the upstream, downstream and duplicated regions, and the genomic average for the interactions of the duplicated region with itself (see the "Methods section). We found that Hi-C-LSTM's predictions significantly outperform this baseline overall (Fig. 7d). Note the baseline is a slightly better predictor of contacts between the upstream and downstream regions.

Hi-C-LSTM's predictions have the advantage that they describe contacts on the true post-duplication genome, in contrast to the reference genome used to map reads (Fig. 7c). Hi-C-LSTM's contacts recapitulate the post-duplication topological domain structure hypothesized by Melo et al. These duplication experiments validate the ability of Hi-C-LSTM to perform in-silico insertion and duplication.

Note that Hi-C-LSTM can simulate only *cis* effects such as structural variants, but not *trans* effects that arise from loss of diffusible entities such as transcription factors.

**Hi-C-LSTM can simulate knockout of transcription factor binding sites and TAD boundaries.** As Hi-C-LSTM is able to perform in-silico insertion/duplication (see the section "Duplication"), we wanted to investigate whether knockout or deletion of certain genomic loci would produce reliable changes in the resulting Hi-C contact map. In-silico knockout experiments have gained prominence lately, mainly in intercepting signal flows in signaling pathways[69] and drug discovery[70–72]. A Hi-C in-silico

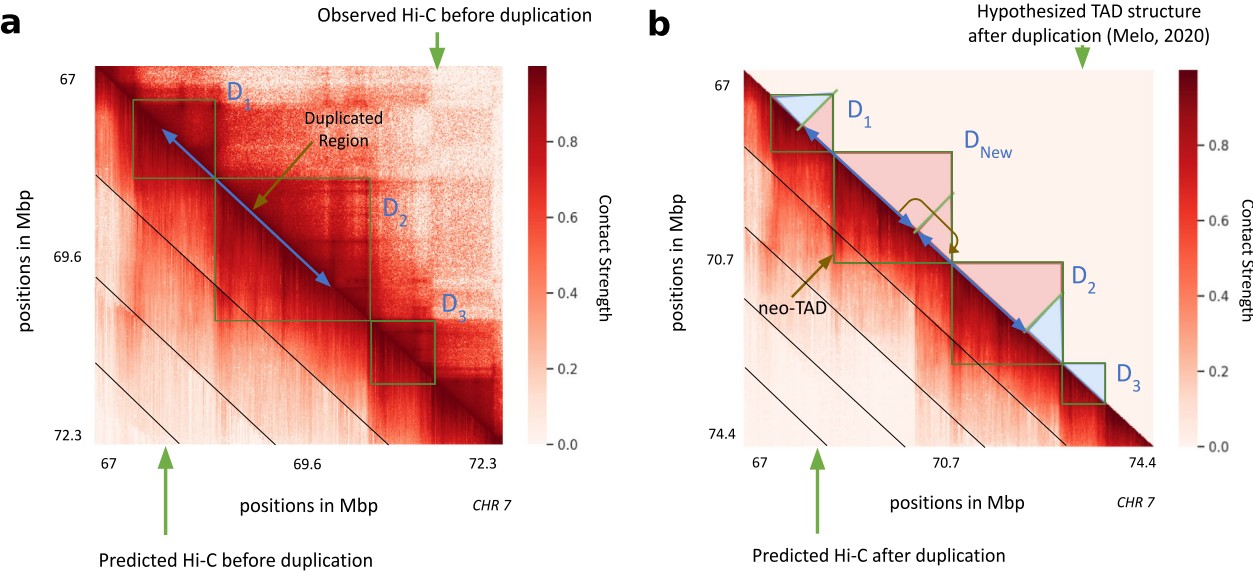

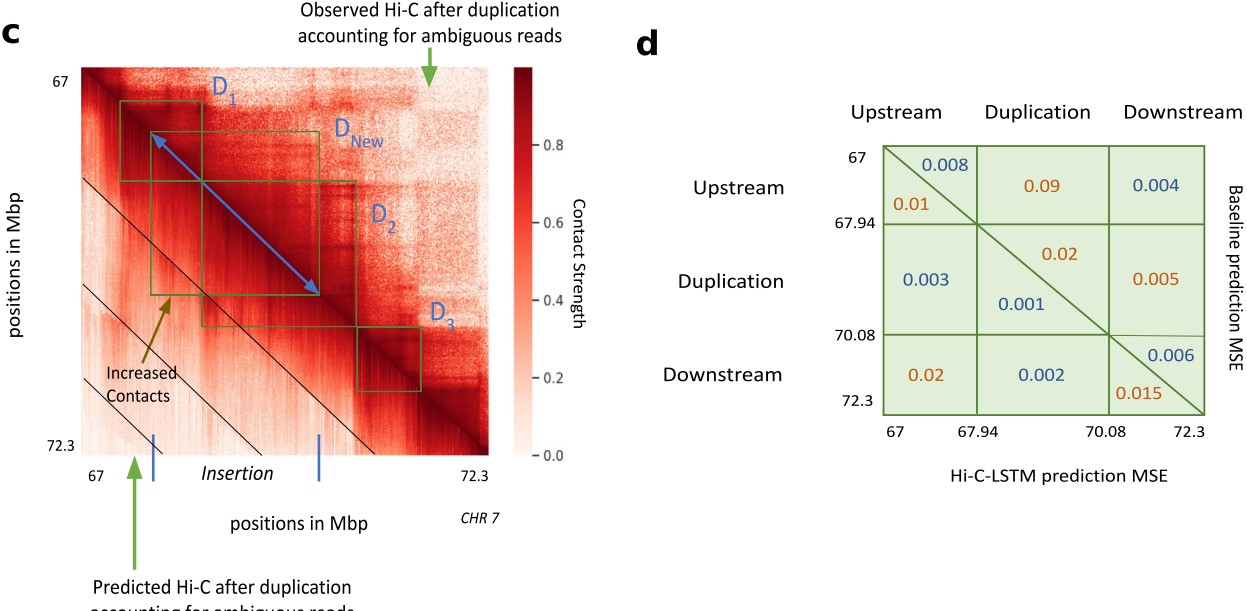

**Fig. 7 In-silico duplication of a 2.1 Mbp region on Chromosome 17.** In all subplots, upper and lower triangles denote observed and predicted Hi-C contact probabilities respectively, and diagonal black lines denote Hi-C-LSTM frame boundaries. **a** Observed and predicted Hi-C before duplication. $D_1$, $D_2$ and $D_3$ indicate the three pre-duplication topological domains. **b** Predicted Hi-C after duplication on a simulated reference genome that includes both copies. Lower triangle indicates Hi-C-LSTM predicted contacts. The true Hi-C contact matrix on this reference genome is not observable because the read mapper cannot disambiguate between the two copies. The upper triangle depicts the post-duplication topological domain structure hypothesized by Melo et al, which includes a novel topological domain $D_{New}$. **c** Observed and predicted Hi-C on the observed pre-duplication reference genome. Upper triangle shows observed post-duplication Hi-C data assayed by Melo et al. Lower triangle shows Hi-C-LSTM predictions, mapped to the pre-duplication reference by summing the contacts for the two copies (see the section "Results"). **d** Average mean-squared error (MSE) in predicting the observed data by (lower triangle) Hi-C-LSTM, and (upper triangle) a simple baseline (see the section "Results") at the upstream, duplicated, and downstream regions.

manipulation tool is of great value it enables researchers to identify the importance and influence of any genomic locus of interest to 3D chromatin conformation.

It is an open question how to simulate small-scale perturbations. We performed knockout using four different techniques at CTCF plus Cohesin-binding sites (see the section "Discussion"). The difference in inferred Hi-C between the CTCF plus Cohesin knockout and the no knockout using shifted representations (see

the section "Methods") shows the decrease in contact strength (7% lower on average) in the immediate neighborhood of the KO site (Fig. 8a). Other ways to simulate knockout like using the padding, zero and average representations (Supplementary: Fig. 8) exploit different properties of the model. We believe there is no one right way to perform knockout, however, we prefer the method of shifting all downstream representations from the knockout site upward (see the "Methods" section).

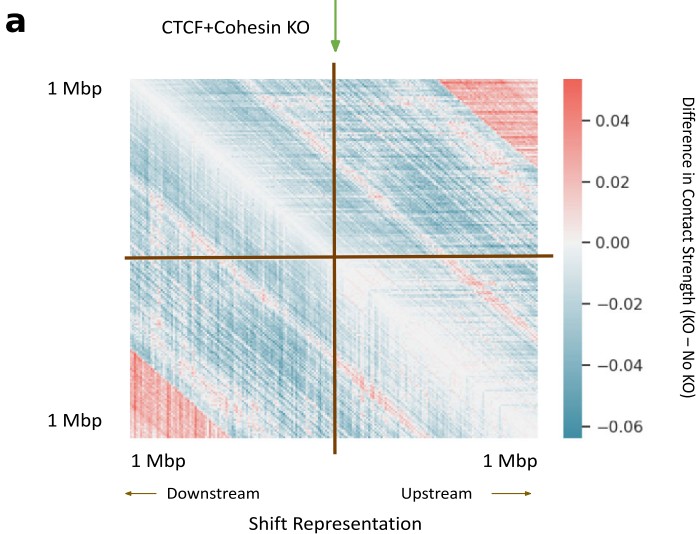

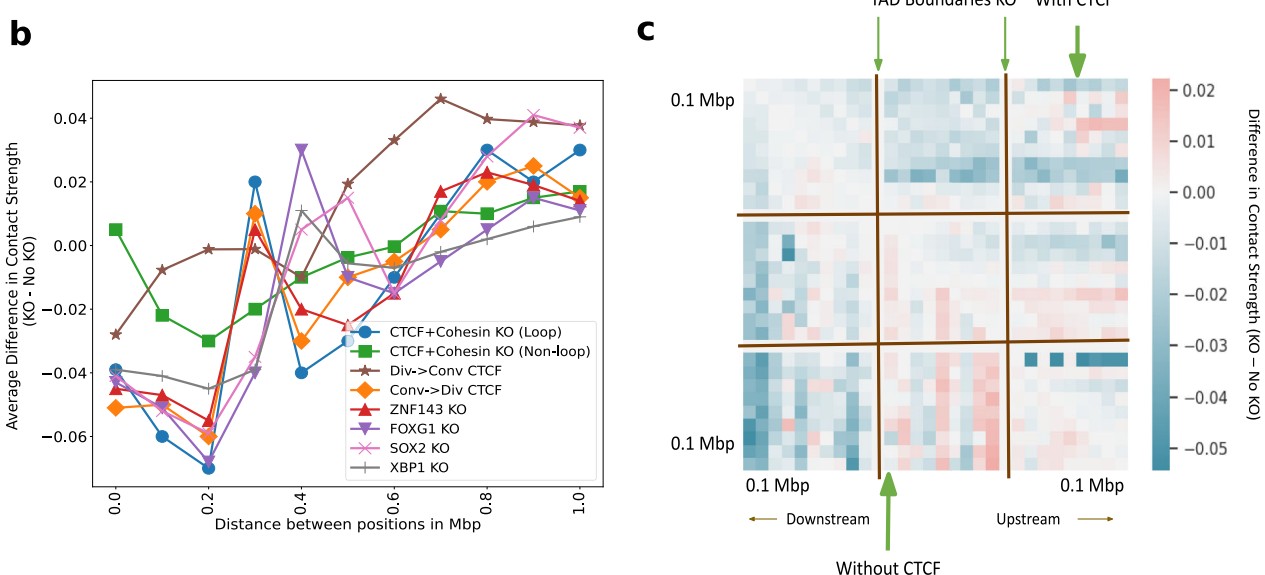

**Fig. 8 In-silico deletion of transcription factor binding sites (TFBS), orientation replacement of CTCF binding sites and TAD boundaries with and without CTCF. a** The average difference in predicted Hi-C contact strength between CTCF + Cohesin knockout (KO) and no knockout in a 2 Mb window. We simulate deletion by shifting the downstream representations upward. **b** Average difference in contact strength of the inferred Hi-C matrix between knockout and no knockout (y-axis) for varying distance between positions in Mbp (x-axis). The knockout experiments include TFBS knockout and convergent/divergent CTCF replacements (legend). **c** The genome-wide average difference in predicted Hi-C contact strength between TAD boundaries knockout and no knockout with CTCF (upper-triangle) and without CTCF (lower-triangle).

Previous work showed that altering even a single base pair near the loop anchors can make many loops and domains vanish, altering chromatin conformation at the megabase scale[39]. Given the crucial role played by CTCF and Cohesin subunits in conformation at loop anchors (see sections "Classification", "Attribution"), we hypothesized that knocking out CTCF and Cohesin subunit binding sites will alter the Hi-C contact map in the neighborhood. The average difference in predicted contact strength between no knockout and knockout at the site under consideration as a function of genomic distance is observed (Fig. 8b). After the combined CTCF and Cohesin knockouts, the average contact strength reduces by 7% in a 200 kbp window when compared to the no knockout case (Fig. 8b). CTCF knockout is seen to affect insulation and reflect possible loss of loops at 200 kbp (Fig. 8b). The knockout of CTCF and Cohesin subunit binding sites at non-loop regions[56,57] (just like feature

attribution, we only considered the case where CTCF and Cohesin share sites, and ignored the cases where CTCF binds alone, and Cohesin binds alone) produces markedly different effects with 2% lower average inferred strength after knockout at 200 kbp, hinting at the relative importance of loop and non-loop binding factors (Fig. 8b).

Along with CTCF, we knocked out the other 4 TF binding sites (TFBS) in the top 5 TFs according to the ranked list, namely, ZNF143, FOXG1, SOX2, and XBP1 (Fig. 8b). We see that the average predicted contacts after genome-wide knockout partially reflects the importance attributed to each TF by integrated gradients. FOXG1 binding site knockout reduces contacts by 7% on average, XBP1 binding site knockout reduces contacts by 4% on average, whereas ZNF143 and SOX2 binding site knockouts reduce contacts between 4% and 5% on average at 200 kbp. Most knockouts cause an increase in contacts at 300Kbp and a gradual

increase in contacts after 400 kbp. These results validate that Hi-C-LSTM knockout of TFBS captures the general idea of contacts depleting within the domain and connecting regions outside the domain.

The CTCF sites at loop anchors occur mainly in a convergent orientation, with the forward and reverse motifs together, suggesting that this formation maybe required for loop formation[18,73–78] (see Supplementary Fig. 9 for illustration). To check how important the orientation of CTCF motifs is to conformation, we conducted CTCF orientation replacement experiments at loop boundaries. The average difference in predicted contact strength between no replacement and replacement at the site under consideration as a function of genomic distance is observed (Fig. 8b). The replacement of convergent with the divergent orientation around loops is seen to behave similar to the case of CTCF knockout thereby validating observations made in[79] (Fig. 8b). On the other hand, replacement of divergent with the convergent orientation is seen to preserve loops at 200 Kbp and behave similar to the control (Fig. 8b).

TADs anchored with CTCF at their boundaries have a differential role to play in conformation compared to the TADs without CTCF. We wanted to check if Hi-C-LSTM can capture this differential behavior of TADs by knocking out their boundaries. To deal with TADs of varying sizes, we partition the interior of all TADs into 10 equi-spaced bins and average the predicted contacts within these bins. We show these along with the regions outside the TAD boundary 100Kbp upstream and downstream, averaged across all TADs (Fig. 8c). The average difference in inferred Hi-C between the knockout at TAD boundaries and the no knockout (Fig. 8c) shows largely decreased contacts for both TADs with and without CTCF in a 200 kbp window (3% lower on average). Within the TAD, however, we see increased contacts for TADs without CTCF (5% higher on average) and decreased contacts with CTCF (4% lower on average) (Fig. 8c).

**Simulating loop anchor deletions at the TAL1 and LMO2 loci Hi-C-LSTM predicts measured 5C data**. To further validate the ability of Hi-C-LSTM to predict experimental perturbations, we simulated the deletion of loop anchor regions at the TAL1 and LMO2 neighborhood boundaries in human embryonic kidney cells (HEK-293T) previously conducted by Hnisz et al.[80]. These deletions were observed in T-cell acute lymphoblastic leukemia (T-ALL) patients. The TAL1 anchor deletion was seen on chromosome 1 in the neighborhood of 47.7 Mbp (GRCh37/hg19, Fig. 9a), and the LMO2 anchor deletion was seen on chromosome 11 in the neighborhood of 34 Mbp (GRCh37/hg19, Fig. 9b)[80]. Both deletions included loop boundary sites. The authors hypothesized that deletions of loop boundary sites at these loci could cause activation of inactive proto-oncogenes within the loops[80]. To simulate a Hi-C experiment on a genome with these deletions, we first obtained the trained model from GM12878 and retrained it on the 5C data from the TAL1 and LMO2 segments[80]. We then made a new representation matrix that shifted the representations downstream from the knockout sites upward, and passed this representation matrix through the retrained Hi-C-LSTM decoder to produce a simulated Hi-C matrix (Supplementary Fig. 10a, b, lower-triangle) (see the "Methods" section for more details) and compared this with the 5C experiment performed by Hnisz et al.[80] (Supplementary Fig. 10a, b, upper-triangle).

They authors saw that the insulated neighborhoods of TAL1 and LMO2 were disrupted, which allowed activation of these elements by regulatory elements outside the loop, and caused rearrangement of interactions around the neighborhood. We found that Hi-C-LSTM's predicted contacts correlate with the

post-deletion interactions hypothesized by Hnisz et al. To evaluate our predictions, we investigated whether there is a correlation in the differences of knockout and no knockout between the observed and the predicted contacts (Fig. 9c, d). We found a noticeable correlation between Hi-C-LSTM's prediction differences between knockout and no knockout and the observed assayed contacts for TAL1 (Fig. 9c). The interactions across domain boundaries that did not exist pre-deletion in the TAL1 neighborhoods were correctly captured by Hi-C-LSTM (Fig. 9c). The correlation for LMO2 was not as strong as TAL1 (Fig. 9d) and the discrepancy was particularly at points where the post knockout contacts were same as the pre-knockout or higher. We see that Hi-C-LSTM accurately predicts decrease in post knockout contacts as decrease, but wrongly attributes some points of no-change and increase as decrease (Fig. 9d).

These anchor deletion experiments reaffirm that Hi-C-LSTM can perform in-silico alterations with moderate accuracy. Moreover, the results also point to the transfer learning ability of Hi-C-LSTM in cell types with limited data (see the section "Discussion").

## Discussion

In this work, we have proposed a recurrent model that uses intra-chromosomal contacts to form position-specific representations of chromatin conformation. These representations are able to capture a variety of genomic phenomena and elements and at the same time distinguish genomic regions, transcription factors and domains that are known to play an important role in chromatin conformation. They also elucidate the interplay between genome structure and function. The classification and feature attribution results validate the ability of the representations to locate vital regions such as CTCF and Cohesin-binding sites.

The primary contribution of this work is the application of a recurrent LSTM to the problem of forming representations for intra-chromosomal interactions. The Hi-C-LSTM not only outperforms the existing models like SCI and SNIPER that form representations in predicting genomic phenomena but also locates elements driving 3D conformation as revealed by feature importance analysis. In addition to these, the Hi-C-LSTM has few distinct advantages over its counterparts. One, it can be used as a contact generation model. It's observed that the Hi-C-LSTM representations are more informative in this regard and that sequential models like the LSTM perform much better at contact generation. Two, a low-dimensional Hi-C-LSTM representation is powerful enough to reasonably recreate the Hi-C matrix (see the section "Hyperparameters"). Three, the Hi-C-LSTM framework allows us to conduct in-silico experiments like insertion, deletion and reversal of elements driving 3D conformation and observe changes in contact generation. This would be extremely useful in fully understanding the role of CTCF and Cohesin-binding sites, other transcription factors, and TADs and their boundaries in chromatin conformation.

Although we do a good job of identifying TADs, we cannot use our method as a computational TAD caller (like Arrowhead, GMAP, HiCseg among others)[81] because we need a gold standard training set to train our classification model, which we obtain from the aforementioned TAD callers. As a result, we also cannot compare our TAD classification performance with these TAD callers.

We noticed that performing knockout of specific single genomic sites is not as straightforward as performing insertion of a larger genomic segment (as seen in the section "Duplication") for Hi-C-LSTM. This is primarily because of two reasons. First, because the row representation is fed throughout the column sequence, Hi-C-LSTM decoder is more dependent on the row

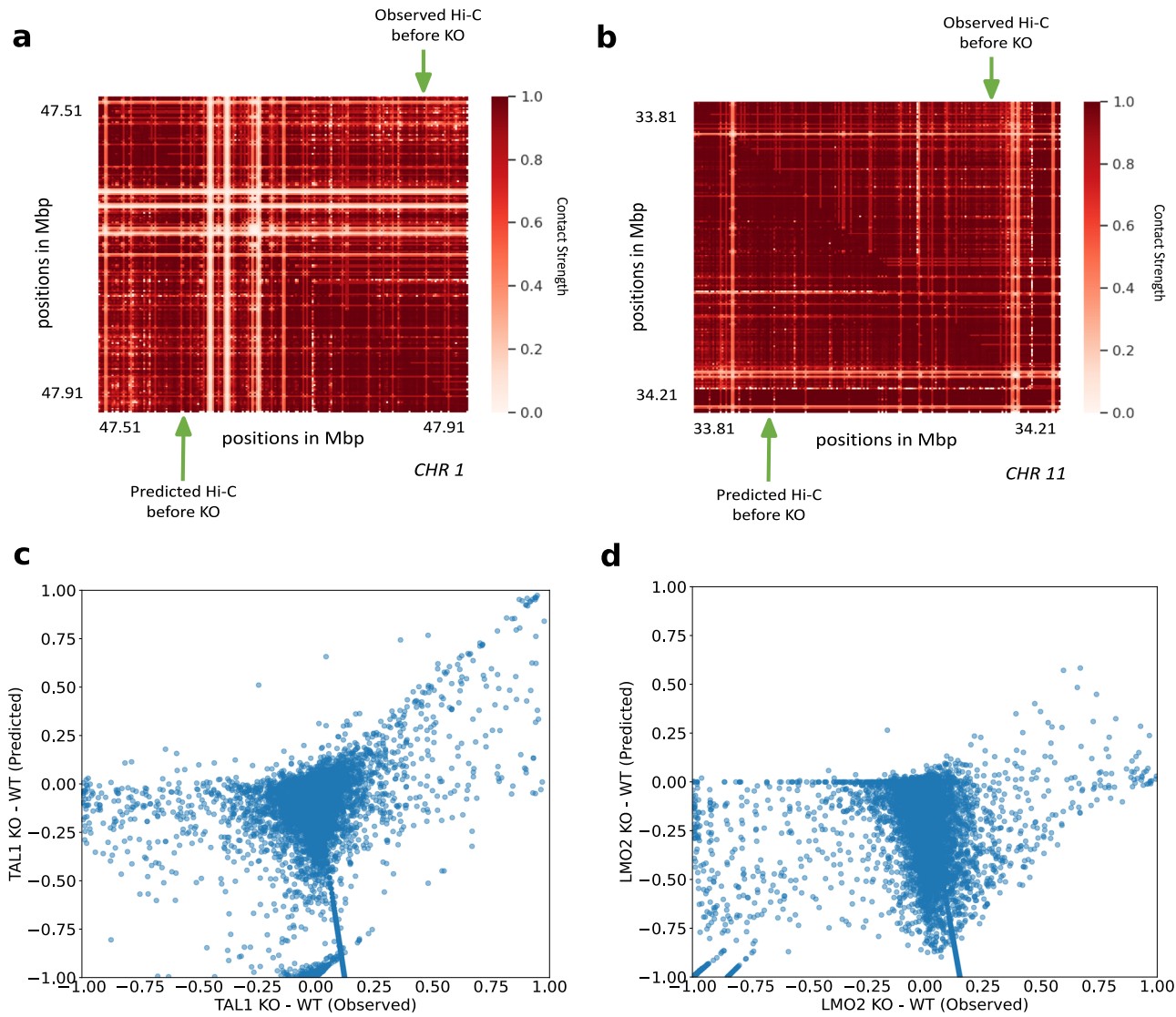

**Fig. 9 In-silico anchor deletions at the TAL1 and LMO2 loci. a**, **c** TAL1 anchor deletion on chromosome 1. **a** Observed Hi-C contacts before deletion (upper-triangle), and predicted Hi-C contacts before deletion (lower-triangle). **c** Scatter plot of differences in contacts after and before TAL1 deletion. The x-axis shows observed differences, and the y-axis shows predicted differences. **b**, **d** LMO2 anchor deletion on chromosome 11. **b** Observed Hi-C contacts before deletion (upper-triangle), and predicted Hi-C contacts before deletion (lower-triangle). **d** Scatter plot of differences in contacts after and before LMO2 deletion. The x-axis shows observed differences, and the y-axis shows predicted differences, KO knockout, WT wild-type.

representations than the column representations. Therefore, Hi-C-LSTM is less susceptible to manipulation of column representations alone, which is the case when inferring contacts for rows around the knockout site, and more reliable for the row pertaining to the knockout site. This issue of robustness to manipulation of column representations is less prominent during insertion because a contiguous segment gets inserted and in the post-insertion genome both the row and the column representations are affected. Second, single locus knockout is harder than knockout of a larger genomic segment because the sequential model is robust to slight perturbations in the input.

Moreover, there is no accepted standard way of simulating in-silico knockout in the Hi-C community in the context of manipulating sequential representations. There are four ways one can simulate the knockout of a locus. One, by replacing the representation by the zero representation. Two, by replacing the representation by the average representation in the neighborhood. Three, by replacing the representation by the representation of the padding input, and four, by shifting all downstream representations upward (see the

"Methods" section). We tried all four techniques and found shifting the representations to be most convincing.

An important limitation of Hi-C-LSTM's in silico experiment is that it can simulate only *cis* effects. Variation in chromatin structure can be caused either by *cis* or *trans* effects. *Cis* effects are caused by genetic variants on the same DNA molecule, whereas *trans* effects arise from diffusible elements like transcription factors. Hi-C-LSTM can model only *cis* effects because *trans*-acting cellular machinery is captured within the Hi-C-LSTM decoder, which cannot be easily modified. An example of a *cis*-effect is the duplication at the SOX9 locus, in which case we showed Hi-C-LSTM correctly models the resulting neo-TAD (see the section "Duplication")[68]. Hi-C-LSTM cannot model *trans* effects such as recent investigation of the removal of RAD21[40] and CTCF[82,83]. To directly validate the *cis*-knockout of CTCF-binding sites, to the best of our knowledge, there is no reliable post-CTCF-binding site knockout Hi-C available in our cell types of interest in humans. There are Hi-C experiments available after CTCF protein depletion[82–86]. There are also Cohesin depletion

experiments[40]. However, these experiments cannot be used to compare with our binding site knockout experiments as depleting the protein itself is not the same as knocking out the binding site the protein can bind to. There is one work that performs post CTCF-binding site knockout Hi-C, but this experiment is conducted in mice[87]. Therefore, instead of looking for a post-CTCF binding site knockout Hi-C, we decided to further verify our model using data from duplications (see the section "Duplication") and anchor deletions (see the section "Anchor").

Single-cell Hi-C (scHi-C) datasets are becoming increasingly valuable in providing us with cell level resolution of contacts. Hi-C-LSTM cannot utilize single-cell Hi-C data—unless that data is aggregated into pseudo-bulk Hi-C—because it takes as input a single Hi-C matrix. Additionally, the model would have to be trained on such data by taking into consideration the data resolution, mapping it to appropriate bins and mapping the contacts between those bins. The main challenge when analyzing scHi-C is that the data is extremely sparse. Pseudo-bulk scHi-C, where many cells are clustered into groups of similar types and pooled in silico, allows for the statistical validation of chromatin patterns. We trained and evaluated Hi-C-LSTM on a subset of chromosomes (15–22) using pseudo-bulk scHi-C data from Ramani et al.[88] (see the "Methods" section for details). A representative heatmap from chromosome 21 shows that Hi-C-LSTM is able to reconstruct contacts faithfully (Supplementary Fig. 11), however, the sparsity of scHi-C data might be a potential concern when using the representations from scHi-C models for other downstream tasks like classification and in-silico manipulation.

Transfer learning is an important goal for the Hi-C community because of the availability of a variety of disparate and sparse Hi-C datasets. Instead of training a new model from scratch for every new Hi-C, using existing models from other cell types can drastically speed up the training process and also deal with the sparsity of available data. We are able to perform Hi-C inference of fragments in a new cell type (HEK-293T) by using partial 5C fragments as input for retraining (Fig. 9a, b). Hi-C-LSTM accurately captures both the TAL1 and LMO2 5C observed fragments (Fig. 9a, b: upper-triangle) in its predicted Hi-C contacts (Fig. 9a, b: lower-triangle). This will allow the model to be rapidly used in cell types and tasks where the available contact data is scarce. Users can use transfer learning to apply Hi-C-LSTM to new data sets by fine-tuning the pre-trained model on the new data sets. However, if the amount of new data available is large, it may be preferable to train a fresh Hi-C-LSTM model.

Hi-C-LSTM, in its current form, is not designed to handle data from multiple cell types. We acknowledge that imputation is an important goal that deserves consideration, however, our goal with Hi-C-LSTM is not to impute data in new cell types but rather form cell type and position specific representations. Learning representations that help you reconstruct the Hi-C map can be useful for multiple reasons, namely, (a) the resulting model becomes a contact generation framework, (b) the resulting representations capture conformation defining elements, (c) the representations coupled with the model can be used for in-silico manipulation of genomic elements, and (d) the process can give us insights about which genomic sites are most important in construction. It is also important to note that we learn the representations in the process of reconstructing the matrix, i.e., reconstruction does not bring biological insight but is part of the process that forms the representations. The Hi-C reproduction evaluation shows how well these representations capture the information in the Hi-C matrix.

To reconstruct, we chose a shorter frame length of 150 because (1) LSTMs can typically work with sequences of lengths in the order of 100s but cannot handle very long sequences in the order of 1000s because of issues with gradient propagation.

(2) A shorter frame length helps us fit our model in memory and speeds up training time drastically. (3) At our choice of 10 kbp resolution it allows us to identify other important large chromatin structures like loop domains, TADs, and subTADs. A shorter frame length is one of the reasons our model does not do well at identifying long-range interactions like subcompartments, however, our goal is not just to identify long-range interactions but design a model that can identify both short and long range interactions satisfactorily. Hi-C-LSTM is able to achieve this trade-off because of its shorter frame length. In future work, we plan to work with longer sequences efficiently by: (1) Creating hierarchical representations from initial representations. (2) Using models like Transformers that can handle longer sequences. (3) Aggregating representations learnt at different Hi-C resolutions.

The good performance of Hi-C-LSTM suggests several avenues for future work. First, extending the mode to incorporate data from multiple cell types and the resulting representations may yield insights into differences in chromatin organization across development. Extending this framework to work with multiple cell types at the same time may be possible with the addition of "cell type id" as an input parameter. Second, an in-depth analysis of Hi-C-LSTM performance on scHi-C warrants a detailed report of its own. Third, combining representations from models trained at varying resolutions to form a common representation would allow us to not only discover new elements at different scales but also form a comprehensive scale agnostic representation. Fourth, the success of a LSTM model suggests trying other sequential neural network models that can handle longer sequences such as Transformers[89], coupled with learning hierarchical representations. Fifth, a modified version of Hi-C-LSTM may be able to infer a 3D structure of chromatin. The Hi-C representations that we form currently are embedded on a lower-dimensional manifold that does not have any direct physical significance. However, a Hi-C-LSTM-like model trained to produce three-dimensional representations may be able to reproduce the true nuclear positions of chromatin.

## Methods
The code and data repository for this project, including training, evaluation, data handling, and generated data can be found in our GitHub repository[90].

**Data sets**. We generated the intrachromosomal Hi-C data set on the hg19 human reference genome assembly[91] at 10 kb resolution with KR (Balanced) normalization[92] using juicer tools with the command `java -jar juicer_tools.jar dump observed KR data/chr.hic chr chr BP 10000 chr.txt`, where `chr` refers to the chromosome being extracted. To extract data at 2, 100, and 500 kb, we simply replaced the resolution field in the juicer command. See the section "Data availability" for links to Hi-C datasets.

Following SCI[12], to mitigate the extreme range of magnitudes present in Hi-C read counts, we transformed Hi-C values into contact probabilities between 0 and 1. We calculated contact probabilities according to the exponential transformation (Eq. (1))

$$cf = \frac{1}{v + \delta}$$
$$CP = \exp(-a * cf), \tag{1}$$

where $v$ is the raw input contact strength, $\delta$ is a very small positive real number (we set $\delta$ to be $10^{-10}$), cf is the coefficient obtained, $a$ is the coefficient multiplier, and CP is the resulting contact probability. We chose $a = 8$ because it appeared to provide a good separation of low and high contact values.

For the classification task, each gene was considered to be active if its log mean expression value across the gene was >0.5[93,94]. See the section "Data availability" for links to gene expression data.

Frequently interacting region (FIRE) scores were converted to binary indicators using 0.5 as a threshold following[95]. See the section "Data availability" for links to FIRE data.

We could not find PEI, FIRE, and replication timing data for H1-hESC and HFF-hTERT, and hence excluded these elements from classification performance evaluation in H1-hESC and HFF-hTERT.

We ran FIMO[96] to get the CTCF motif instances using the command `fimo -oc output_directory motif_file.meme sequence_file.fna`. We use all default options while running fimo including the p-value threshold (`-thresh`) of $10^{-4}$. We ran FIMO after obtaining the human genome sequence file under mammals and the hg19 genome assembly. See the section "Data availability" for more details.

To classify CTCF and Cohesin peaks based on loop and non-loop regions, we used loop domain data for GM12878. We then segregated them as CTCF + Cohesin, CTCF only and Cohesin only sites in both loop and non-loop regions based on thresholds used in Hansen et al.[56]. See the section "Data availability" for more details.

For a full list of links to relevant datasets, refer to the section "Data availability" and Table 2 in the Supplementary.

**LSTM**. Long short-term memory (LSTM) networks were proposed as a solution to the vanishing gradient problem[97] in recurrent neural networks (RNNs)[98]. They are known to be a good candidate for modeling sequential data and have been widely used for sequential tasks[99–101]. An LSTM is made up of a memory state ($h_t$), a cell state ($c_t$), and three gates that control the flow of data: input ($i_t$), forget ($f_t$) and output ($o_t$) gates. The input and the forget gates together regulate the effect of a new input on the cell state. The output gate determines the contribution of the cell state on the output of the LSTM.

Let matrices $W$ and $U$ be the weights of the input and recurrent connections, and $b$ refer to the biases. There are four sets of weight matrices and biases in the LSTM. These include one for each of the three gates—forget gate ($W_f$, $U_f$, $b_f$), input gate ($W_i$, $U_i$, $b_i$) and output gate ($W_o$, $U_o$, $b_o$)—and one to form the cell state ($W_c$, $U_c$, $b_c$). The current cell state ($c_t$) is formed by the modulation of the previous cell state ($c_{t-1}$) by the forget gate ($f_t$) and combining it with the modulation of the current input ($x_t$) and the previous memory state ($h_{t-1}$) by the input gate ($i_t$). Finally, the current memory state ($h_t$) is formed by the modulation of the current cell state ($c_t$) by the output gate ($o_t$). The current memory state ($h_t$) is fed into a linear layer with a sigmoid function at each step which produces the final output interacting frequency at that step.

An LSTM's output is determined by the following series of operations[41].

$$
\begin{aligned}
f_t &= \sigma(W_f x_t + U_f h_{t-1} + b_f) \\
i_t &= \sigma(W_i x_t + U_i h_{t-1} + b_i) \\
o_t &= \sigma(W_o x_t + U_o h_{t-1} + b_o) \\
c_t &= f_t \circ c_{t-1} + i_t \circ \sigma(W_c x_t + U_c h_{t-1} + b_c) \\
h_t &= o_t \circ \sigma(c_t)
\end{aligned}
\tag{2}
$$

where ∘ is the Hadamard product and $\sigma$ refers to the sigmoid activation function.

**Hi-C-LSTM**. Hi-C-LSTM creates a representation given a pair of genomic positions in the Hi-C contact matrix using an embedding neural network layer (for an illustration see Supplementary Fig. 12) and predicts the contact strength at that particular pair via a LSTM[41] that takes these representations as input (Fig. 2). Hi-C-LSTM takes as input a $N \times N$ intra-chromosomal Hi-C contact matrix ($\mathbb{R}^{N \times N}$), for each chromosome, where $N$ is the chromosome length.

A trained Hi-C-LSTM model consists of LSTM parameters (see section "LSTM") and a representation matrix $R \in \mathbb{R}^{N \times M}$, where $M$ is the representation size. At each genomic position, $(i, j)$ pair is given as input to an embedding layer, which indexes the row and column representations $R_i, R_j \in \mathbb{R}^M$ and feeds these two vectors as input to the LSTM. The output of the LSTM is the predicted Hi-C contact probability $\hat{H}_{i,j}$ for the given $(i, j)$ pair.

The hidden states of the LSTM are carried over from preceding columns thereby maintaining a memory for the row. For the sake of memory usage, the hidden states are reinitialized after every each frame of 1.5 Mbp or 150 resolution bins (see section "Modeling Choices"). This process is repeated for each row of the Hi-C matrix (Eq. (3)).

$$
\hat{H}_{i,j} = \text{LSTM}((R_i, R_j), h_j, c_j) \quad
\begin{aligned}
&\text{for } j = 1, 2, \ldots, N \\
&\text{for } i = 1, 2, \ldots, N
\end{aligned}
\tag{3}
$$

where $h_j$ and $c_j$ are reinitialized at the beginning of each new frame.

The LSTM and the embedding neural network layer are jointly trained using the mean squared error (MSE) loss function which facilitates the faithful construction of the Hi-C intra-chromosomal matrix (Eq. (4)).

$$
\text{MSE}_i = \frac{1}{N} \left[ \sum_{j=1}^{N} \left( H_{i,j} - \hat{H}_{i,j} \right)^2 \right] \text{ for } i = 1, 2, \ldots, N
\tag{4}
$$

At the end of all the training iterations, the output of the embedding neural network layer at each row $i$ ($R_i$) is treated as the representation for that row. The Hi-C-LSTM framework infers the Hi-C contact matrix from pairs of position IDs and therefore is a transformation from linear sequential space to the Hi-C space. The linear position IDs are a convenient and useful modeling assumption which builds a framework that does not make any other transfer function assumptions.

**Modeling choices and training**. The LSTM model required us to make a few design choices. As layer normalization can significantly reduce the training time and is effective at stabilizing the hidden state dynamics in LSTMs, we used a unidirectional layer norm LSTM[102] with one hidden layer. We found that variants such as the bidirectional LSTM[103] and LSTM with multiple layers provided a marginal increase in test performance (Supplementary Fig. 13). The variants were also prone to overfitting. Therefore, we chose the single-layer unidirectional model over these variants accounting for computational efficiency and good generalization. Gradient clipping[97] and the *softsign* activation[104] were used at all nodes owing to their mitigating effect on hidden state saturation. The design choices were made after conducting ablation experiments which are elaborated in the following section "Hyperparameters". We used a batch size of 2000 and a sequence length 150 bins, both of which were observed to be data dependent and the best fit for our data. We used a learning rate of 0.01 for 5 epochs and 0.001 for 5 more epochs. We reinitialized the hidden states of the LSTM after every frame of length 150 and predicted each diagonal block of length 150 with fresh hidden states (Fig. 3c). The prediction error improved towards the end of the frame and increased at the start of the next frame (Supplementary: Fig. 14). We tried passing the hidden states across frames and saw that the convergence time significantly increased as the training graph had to be retained across iterations. So we chose to reinitialize the hidden states in each window instead.

We employed PyTorch, a Python-based deep learning framework and trained Hi-C-LSTM on GeForce GTX 1080 Ti GPUs with ADAM as the optimizer[105]. All parameters in PyTorch were set to their default values while training. As our primary goal was not to infer values for unseen positions but to form reliable representations for every chromosome, we trained our model on the full genome. For our Hi-C reproduction evaluation, we trained the representations on the full genome but the decoders only on a random subset. We chose to train the decoders on a random subset of the genome to prevent the decoder from overpowering the representations. The time taken to train and test all methods is included in the Supplementary Table 3 (Running Time).

**Hyperparameter selection**. To choose the representation size of our model, we performed an ablation analysis. We computed the average mAP across all downstream tasks with the Hi-C-LSTM model which consists of a single layer, unidirectional LSTM with layer norm in the absence of dropout[106] for odd chromosomes and used the even chromosomes to validate whether the choice of hyperparameters remained the same irrespective of chromosome set. We observed the mAP (see "Methods" section) of the Hi-C-LSTM vs. increasing representation size along with Hi-C-LSTM that is bidirectional, in the presence of dropout, without layer norm and 2 layers (Supplementary Fig. 13). While both the presence of dropout and the absence of layer norm adversely affected mAP, the addition of a layer and a complimentary direction did not yield significant improvements in downstream performance. We conducted a similar ablation experiment and computed the average Hi-C R-squared for the predictions with increasing representation size (Supplementary Fig. 13) and observed that the performance trend is preserved, which was indicative of the fact that recreating the Hi-C matrix faithfully aids in doing well across downstream tasks. These results were verified to be true for even chromosomes as well (Supplementary Fig. 13). For both odd and even chromosomes, even though the Hi-C prediction accuracy increased with hidden size, we noticed the elbow at a representation size of 16 for average mAP and therefore set our representation size to that value as a trade-off.

**Hi-C reproduction evaluation**. We investigated three hypotheses with following analysis. First, we asked whether the Hi-C-LSTM representations faithfully construct the Hi-C matrix. Second, whether the Hi-C-LSTM representation and the decoder are both powerful in generating the Hi-C map. Third, we evaluated the utility of the representations to infer a replicate map. In all cases, we computed the average prediction accuracy in reconstructing the Hi-C contact matrix, measured using $R$-squared, which represents the proportion of the variance of the observed Hi-C value that's explained by the Hi-C value predicted by the Hi-C-LSTM. We sampled the means of observed Hi-C values at different distances between positions and used that as a baseline.

In our first experiment, we trained both the representations and decoders on replicate 1 (Fig. 3a). We took representations trained using all chromosomes from Hi-C-LSTM, SCI and SNIPER and coupled these with some selected decoders, namely, a LSTM, a convolutional neural network (CNN) and a fully connected (FC) feed-forward neural network (used by SNIPER). We compared LSTM with CNN and FC decoders mainly because CNNs provided us with an alternative way of incorporating structure (using moving filters) and FC networks did not include any information about underlying structure. We re-trained these decoders using either of the representations as input, with a subset of the genome and tested on the rest. All the decoders were configured to have the same number of layers and hidden size per layer. As the decoders were separately trained, this process allowed us to check the power of the representations alone, moreover, as a subset of the genome was used to train the decoder, we reduced the possibility of the decoders overfitting.

In our second experiment (Fig. 3b), we trained the representations on replicate 1 using all chromosomes, and repeated the aforementioned decoder training process on replicate 2.

We conducted both these experiments in all 4 cell types, namely, GM12878 (Fig. 3a, b), H1-hESC (Fig. 3c, d), HFF-hTERT (Supplementary Fig.1a, b), and WTC11 (Supplementary Fig.1c, d).

**Comparison methods**. We compared our downstream classification results with five alternatives: two variations of SNIPER, one with inter-chromosomal (SNIPER-INTER) and the other with intra-chromosomal contacts (SNIPER-INTRA), SCI and two baselines, namely, the subcompartment-ID (SBCID) and principal component analysis (PCA). SNIPER-INTRA was the same as the original SNIPER-INTER, modified to take the intra-chromosomal row as input instead of the inter-chromosomal row. All the parameters for the two SNIPER versions and SCI were set as given in their respective papers[11,12]. The SBCID baseline used the one-hot-encoded vector of the subcompartment as the representation at the position under contention. The PCA baseline assigned the principal components from the PCA of the Hi-C matrix as the representations.

**Element identification evaluation**. We used the following analysis to evaluate the ability of a representation to identify genomic phenomena and chromatin regions.

For each type of element, we first trained a boosted decision tree classifier called XGBoost[53] on the representations. We tried tree boosting first as it is shown to outperform other classification models with respect to accuracy when ample data is available. Following Avocado[95], we used XGBoost with a maximum depth of 6 and a maximum of 5000 estimators and these parameters were chosen following ablation experiments with odd chromosomes as the training set and even chromosomes as the test set (Supplementary Fig. 15). N-fold cross-validation, with $n = 5$, was used to validate our training with and an early stopping criterion of 20 epochs. The rest of the XGBoost parameters were set to their default values.

For each task, the genomic loci under contention were assigned labels. All tasks were treated as binary classification tasks, except the subcompartments task, which was treated as a multi-class classification task. For tasks without preassigned negative labels, negative labels were created by randomly sampling genome-wide, excluding the regions with positive labels. We sampled negative labels until the number of negative labels equaled the number of positive labels to avoid class imbalance during classification. The XGBoost classifier was given the representations at these genomic loci as input and the assigned labels as targets.

We then compared the XGBoost classifier trained separately for each task with a multi-class multi-label classifier with a simple linear layer and sigmoid output. We observed that the multi-class classifier, which predicted regions/domains the given position belonged to, was much faster and gave more reliable results when compared to the XGBoost classifier. Therefore, we prefer the linear classifier for classification.

The classifiers were evaluated using four standard metrics for classification tasks, namely, mean average precision (mAP) (otherwise known as area under the Precision-Recall curve (AuPR)), area under the Receiver Operating Characteristic curve (AuROC), Accuracy ($A = \frac{TP+TN}{TP+FP+TN+FN}$), and $F$-score. AuROC is defined as the area under the curve that has true positive rate (TPR $= \frac{TP}{TP+FN}$) on the $y$-axis and false positive rate (FPR $= \frac{FP}{FP+TN}$) on the $x$-axis. mAP is defined as the average of the maximum precision ($P = \frac{TP}{TP+FP}$) scores achieved at varying recall levels ($R = $ TPR). $F$-score is defined based on precision and recall ($F = \frac{2P*R}{P+R}$). We compared these metrics for GM12878, H1-hESC, and HFF-hTERT (see Supplementary Figs. 4–6 for more details).

**Sequence attribution**. We validated the utility of the Hi-C-LSTM representations in locating genomic regions important for conformation using feature attribution analysis. Feature attribution was carried out on the intra-chromosomal representations using Integrated Gradients[107]. Integrated Gradients is a feature attribution technique that follows an axiomatic approach to attribution, adhering to the axioms of sensitivity and implementation invariance. Sensitivity implies that if the input and baseline differs in one feature and have different predictions, then the differing feature should be assigned a non-zero attribution. Implementation invariance requires that two networks, whose output is equal for every input despite having different implementations, should have the same attributions. We visualized our feature importance in the UCSC Genome Browser along with other genomic elements and signals. The 3D genome browser[108] is also an useful tool for visualization for contact map data.

We used Captum, a Integrated Gradients feature attribution framework in PyTorch that is generic and works with sequential models. The resulting feature attributions were summed across all features, giving us one importance score for every position in the genome. The feature importance scores were then subjected to min-max normalization (Eq. (5)) for both positive and negative values separately. Specifically, if IG is to the integrated gradients (IG) score, and $IG_{min}$, $IG_{max}$ are the minimum and maximum IG scores, then the normalized IG score $IG_{norm}$ is

defined as

$$IG_{norm} = \frac{IG - IG_{min}}{IG_{max} - IG_{min}}. \tag{5}$$

**In-silico perturbation**. The Hi-C-LSTM enables us to perform in-silico deletion, orientation replacement and reversal of genomic loci and predict changes in the resulting Hi-C contact map. We performed three types of experiments:: knockout, CTCF orientation replacement, and duplication. In a knockout experiment, we chose certain genomic sites (such as CTCF and Cohesin binding sites) and replaced their representations with a different representation depending on the method used to perform the knockout (Supplementary Fig. 8).

Among the four possible methods to perform knockout, we prefer the method of shifting the representations. Shifting the representations not only captures the true post-duplication genome but also avoids the noise that comes from zeroing or averaging the representations in the neighborhood (Supplementary Figs. 8, 16). It also is more interpretable than using the padding representation (Supplementary Figs. 8, 16) because we do not fully understand the role of padding representations in recreating the Hi-C matrix. The knockout of the representation at a particular row affects not just the Hi-C inference at columns corresponding to that row but also the succeeding rows because of Hi-C-LSTM's sequential behavior. The LSTM weights remain unchanged, but as the input to the LSTM is modified, the inferred Hi-C contact probability is altered based on the information retained by the LSTM about the relationship between the sequence elements under contention and chromatin structure.

In a CTCF orientation replacement experiment, we replaced the representations of downstream-facing CTCF motifs with the genome-wide average of the upstream-facing motifs and vice versa. This was done under the assumption that the average representation of the given orientation would encapsulate the important information regarding the role played by the orientation in chromatin conformation.

Our duplication experiment was carried out by creating a tandem duplication the representations from the 2.1 Mbp region between 67.95 and 70.08 Mbp in chromosome 7 region[68] and then passing the resulting representation matrix to the LSTM to infer contacts. Given our Hi-C resolution of 10 kbp, the duplicated region corresponds 214 bins, i.e., bin 6795 to bin 7008. Specifically, the duplicated representation matrix is defined as $\hat{R}_i := [R_{1:6794}, R_{6795:7008}, R_{6795:7008}, R_{7009:N}]$.

To enable comparison to Hi-C data mapped to the observed pre-duplication reference genome, we combined inferred contacts from both copies. This combination is required because Hi-C reads cannot be disambiguated between the two duplicated copies when they are mapped to the reference genome. Specifically, we passed the predicted contact probability $cp$ through the inverse exponential transformation to define predicted read counts $CS = \frac{1}{-\log cp/a} - \delta$ (see Eq. (1)). We summed predicted read counts from the two duplicated copies to simulate mapping reads from both copies to the same reference genome $CS'$, then re-applied the exponential transform to obtain predicted contact probability $cp'$.

Our baseline for the quantitative evaluation was the observed pre-duplication Hi-C for the interactions between the upstream, downstream and duplicated regions, and the genomic average for the interactions of the duplicated region with itself. We considered a window of 214 bins (length of the duplicated region), and computed the average genomic contact strength for the bins with themselves in a window of this size.

Our anchor deletion experiment was carried out by first obtaining the trained Hi-C-LSTM model from GM12878, and retraining it on the 5C data from the TAL1 and LMO2 segments in HEK-293T[80]. The TAL1 fragment is on chromosome 1 from 47.5 to 47.9 Mbp, and the LMO2 fragment is on chromosome 11 from 33.8 to 34.2 Mbp (GRCh37/hg19). After retraining the model with data from HEK-293T, we made a new representation matrix by shifting all the downstream representations upward (Supplementary Fig. 8), and passed this representation matrix through the retrained Hi-C-LSTM decoder to produce the inferred Hi-C matrix (Supplementary Fig. 10).

**Reporting summary**. Further information on research design is available in the Nature Research Reporting Summary linked to this article.

## Data availability

The representations generated in this work have been deposited in the GitHub repository[90]. The Hi-C text file for chromosome 22 is provided in the GitHub repository[90] as a minimum dataset for reproducibility. The transcription factor binding site feature attribution data generated in this study is provided as a Supplementary Data file.

The data that support the findings of this study are publicly available to download.

The Hi-C data for GM12878 was acquired using the GEO accession number GSE63525[18]. The Hi-C data for other tier 1 cell types was acquired from the 4DN Data Portal, like H1-hESC, WTC11, and HFF-hTERT. The Hi-C data for GM12878 with lower read depths were also downloaded from the 4DN Data Portal, such as 300M and 216M.

The intra-chromosomal Hi-C data set text file on the hg19 human reference genome assembly[91] was obtained at 10 kb resolution using juicer tools.

RNA-seq data for GM12878, H1-hESC, and HFF-hTERT was obtained from the Roadmap Consortium.

For GM12878, we defined promoter–enhancer interactions (PEI) as the ones that were used to train TargetFinder[109].

For GM12878, Frequently interacting region (FIRE) scores at 40 kbp resolution were downloaded from the additional material of ref. [51].

For GM12878, the replication timing data given by Repli-Seq[110] was downloaded from Replication Domain at 40 kbp resolution.

For GM12878, H1-hESC, and HFF-hTERT, locations of known enhancers and transcription start sites (TSSs) were obtained from FANTOM and ENCODE, respectively.

For GM12878, Loop Domains and Subcompartments were obtained from the results of Rao et al.[18] using the GEO accession number GSE63525. For H1-hESC and HFF-hTERT, Loop Domains were obtained by running HICCUPS[18] and Subcompartments were obtained by running Gaussian HMM[18].

Segway and Segway-GBR labels were obtained from Hoffmanlab and Noblelab, respectively.

CTCF, Cohesin peak calls for GM12878 were downloaded from ENCODE. The CTCF orientations were obtained by using the CTCF motif from the MEME suite (version 5.3.3) and running FIMO[96] to get the motif instances.

Other Transcription Factor binding sites (TFBS) for the feature importance evaluation were downloaded from The Human Transcription Factors repository.

For GM12878, topologically associating domains (TADs) were downloaded from TADKB[111] and subTADs were obtained by running GMAP[112]. For H1-hESC and HFF-hTERT, both TADs and subTADs were obtained by running GMAP[112].

For our duplication experiment, we obtained the duplicated Hi-C for the 2.1 Mbp region between 67.95 and 70.08 Mbp in chromosome 7 from Melo et al.[68].

For our anchor deletion experiment, we obtained the 5C data for the TAL1 and LMO2 fragments in chromosome 1 and 11 from Hnisz et al.[80].

Pseudo-bulk single-cell Hi-C (scHi-C) data was downloaded using the GEO accession number GSM2254215. We used the validPairs file to filter the data based on chromosome, positions and barcodes. We then obtained the number of reads aligned to hg19 corresponding to each distinct pair of bar codes from the percentages file.

Refer to the section "Methods" and Supplementary Table 2 for additional details.

## Code availability

The code repository for this project, including training, evaluation, data handling, and generated data can be found in our GitHub repository[90].

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

## Acknowledgements

K.B.D. is funded by the Four Year Fellowship (4YF) at the University of British Columbia. E.A.-J. is funded by the UK National Productivity Investment Fund Ph.D. studentship in Data Science and Artificial Intelligence. M.M. is funded by the Medical Research Council UK and a Wellcome Trust Investigator Award (099276/Z/12/Z). V.K.B. is funded by NSERC (AWD-001606). M.W.L. is funded by NSERC (RGPIN/06150-2018), Health Research BC (SCH-2021-1734) and Genome Canada (kdd-445).

## Author contributions

K.B.D. led ideation, genomic data processing, building and validating the machine learning model, wrote the first draft of the manuscript, and edited the manuscript. A.M. contributed towards ideation, data processing, parallelization of the model and model validation. E.A.-J. and M.M. provided inputs for downstream experiments and analyses. V.K.B. partially funded the project. M.W.L. supervised the project. All authors participated in the design of the study, the interpretation of results, and editing the manuscript. All authors read and approved the final manuscript.

## Competing interests

The authors declare no competing interests.
