## [Peer Review File · Nature Communications]

Reviewers' Comments:

Reviewer #1:

Remarks to the Author:

In this manuscript, Kevin B. Dsouza et al. proposed an LSTM-based model to embed the Hi-C intra-chromosomal contacts. The authors demonstrated that the learned representations can improve the reconstruction accuracy of the Hi-C matrix, and capture cell type-specific functional activity, genomic elements, and genomic regions that drive conformation. They also showed that Hi-C-LSTM can perform in-silico perturbation of CTCF and cohesin binding sites. However, there are several major issues to be addressed.

1. The authors emphasized that Hi-C-LSTM representations capture the information for the reconstruction of Hi-C maps. However, the motivation of reconstruction is not well justified. Why using the representations to recreate the Hi-C matrix? Can the recreated Hi-C matrix bring additional biological insights, using standard pipelines for Hi-C data analysis? Why not compare with Hi-C data imputation methods?
2. The performance of Hi-C-LSTM was evaluated on only one dataset, even though there are amounts of Hi-C data in literature. The authors need to systematically assess the reconstruction performance and the classification performance with multiple Hi-C datasets from various tissues.
3. In addition, it is interesting to test the performance on datasets with various sequencing depths, for example, by downsampling. Besides, can Hi-C-LSTM be used on single-cell Hi-C data?
4. The authors should elaborate on the data used for classification (Figure 4) in Methods Section. For example, enhancers of which cell type were used? How many negative samples were sampled? Besides, in addition to mAP, it is good to provide other metrics, e.g., Area Under the Receiver Operating Characteristic Curve and Area Under Curve.
5. It is good to provide the running time of different methods, and the robustness to Hi-C data resolution.
6. Feature attribution showed the trend for the identification of TADs. Can Hi-C-LSTM identify TADs via feature attribution but not just the trend? If so, will the performance outperform other methods for TAD identification?

Reviewer #2:

Remarks to the Author:

The submitted manuscript undertakes the challenge of discerning the genomic relationship between the 1D sequence's features and the 3D conformation to understand their effect on gene expression and disease. Using a recurrent long short-term memory (LSTM) neural network model, the authors develop Hi-C-LSTM. The new method summarizes the intra-chromosomal Hi-C contacts information into low-dimensional sequence latent representation. The authors show that the resulting sequence representations contain enough information to recap the original Hi-C matrix with high accuracy and present positive results compared to other existing methods. The method enables the identification of known genomic elements related to 3D structure conformation, including transcription factor (TF) binding sites associated with loop conformation and nuclear compartments. The authors show the use of the method for in-silico elements' perturbation experiments to measure the influence of genomic loci on the 3D conformation. This reviewer has several comments that the authors should address to improve the quality of the manuscript for further revision before considering the manuscript for acceptance.

Major:

- 1- The 3D Genome Browser [Wang, Y. Genome Biology. 2018] is a rich source of 3D chromatin interaction data that this paper does not utilize in any way. This reviewer believes that this study should further exploit the current availability of Hi-C data.
 - a) The authors only check the efficacy of Hi-C-LSTM representations by analyzing a biological

replicate in predicting a Hi-C contact map (Fig. 3a, b). TADs are conserved building blocks of genomes with regulatory functions. Thus, authors can enhance their analysis by investigating the method's performance using several Hi-C datasets to train the representations and decoders then obtain the corresponding prediction. That work could inform about what type of interaction is more likely to be captured by the method.

b) It would be of great value if the authors could provide statistical insight about the efficacy of the method recapitulating the 3D structures, breaking it down by length into TADs, subTAD, and smaller loops.

2- This reviewer is under the impression that the authors still do not show convincingly that the model learns anything besides the known presence of the CTCF-motifs in convergent orientation and cohesin complex within the anchor region.

a) In the section "Feature attribution reveals association with genomic elements driving 3D conformation", the authors show in Fig. 3 that the variation of the aggregated feature importance across relevant genomic regions can distinguish boundaries of domains and genomic regulatory elements. The objective of the method is to provide information about the 1D sequence feature associated with the 3D conformation. Therefore, it is relevant to complete the analysis by adding further motif enrichment analysis in the sequences with high aggregated feature importance. That can provide insight into other TF's binding sites besides CTCF related to 3D loop formation.

3- This reviewer considers that the in-silico knockout experiment is incomplete. As noted in above comment 2, the method's novelty relies on providing insight about 3D conformation beyond the well-known CTCF anchors. In the current analysis, the authors confirm that CTCF and cohesin (RAD21 and SMC3) are critical drivers for reducing 3D contact strength. This type of analysis looks encouraging, therefore would be interesting if the authors can extend it by:

a) Testing the in-silico knockout of CTCF and cohesin out of regions of IG importance peaks and within no loop regions to demonstrate the specificity of CTCF binding that the model picks up.

b) Testing the in-silico knockout of other TF bindings within IG importance peaks with no CTCF to evaluate potential new elements related to 3D contortion.

4- For validating the prediction accuracy using known structural variations that impact the 3D contact map, the authors used a 2.1 Mbp duplication at the SOX9 locus in individuals with Cook's syndrome. This reviewer notes the value of that positive validation, but that is a considerable duplication impacting a large fraction of the chromosome.

a) It will be interesting if the authors can validate their method using narrower rearrangements that disrupt the 3D conformation. This reviewer suggests using the two deletions at anchors of loops that contain the oncogenes TAL1 and LMO2 in T-ALL patients [Hnisz, D. et al. Science. 2016]. That paper showed that the deletions of the anchors increased interactions between enhancers and promoters that were isolated in the wild-type cells (anchors presence). That result has been validated previously by another machine learning approach [Trieu, T. et al. Genome Biology. 2020] thus, the authors will reinforce the utility of their result if they can use their method to evaluate the impact of these two deletions on the 3D contact map of these regions.

Minor:

1- In the analysis for Fig. 5a, it is unclear how many regions the authors use to compute the IG importance.

2- Fig. 5b should show the CTCF binding track.

3- Fig. 5c should show the p-value of the appropriate comparison for each group.

4- In the analysis for Fig. 6, it is unclear how many regions the authors use for the in-silico knockout test.

Reviewer #3:

Remarks to the Author:

The authors use embedding neural network layers to create 1D representation of genomes from their 3D Hi-C heatmaps. The 1D representation can then be converted to 3D interacting frequencies by a long short-term memory (LSTM) neural network model. This method is called Hi-C-LSTM. To compare their method (Hi-C-LSTM) with SNIPER and SCI (which also predict 1D representation of genomes), the authors divide the trained model into two parts. The first part is 1D representation, which is actually the embedding neural network layer with trained parameters. The second part is a decoder, which is actually the LSTM with trained parameters. The authors find

that not only the 1D representation of Hi-C-LSTM captures more genome information than those of SNIPER and SCI, but also the LSTM as a decoder, predicts the 3D interacting frequencies more precisely than alternative decoders such as a convolutional neural network (CNN) or a fully connected (FC) feed-forward neural network. To further show that the 1D representation of Hi-C-LSTM is more informative than those of SNIPER and SCI, the authors train a boosted decision tree classifier called XGBoost by 1D representations versus gene expression, enhancers, TSSs, and so on. The 1D representation gives better classifications on all genomic phenomena and regions except subcompartments. The authors claim that this is due to the intra-chromosome design of Hi-C-LSTM. There are no experimental confirmations for all claims.

By feature attribution analysis, the authors find that genomic elements related to 3D conformation, such as CTCF, cohesin, TAD boundaries, have higher IG scores based on the 1D representation of genomes predicted by Hi-C-LSTM.

Finally by doing in-silico knockout experiments on the 1D representation of genomes, Hi-C-LSTM allows one to predict the resulting 3D conformation without wet experiments. Knockout of CTCF or cohesin binding sites and CTCF orientation replacement give predicted interacting variances consistent loop extrusion model. Moreover, Hi-C-LSTM "accurately" predicts effects of a 2.1 Mbp duplication at the SOX9 locus revealed in previous wet assays.

Here are some points to consider.

1. In lines 243-290, the authors do in-silico knockout experiments on both CTCF and cohesin binding sites based on the data set in lines 393-396. However, the cohesin does not have known binding motifs. In fact, the binding sites of cohesin highly overlap with those of CTCF because CTCF blocks cohesin. The authors should pay more attention to it.
2. The in-silico knockout experiments on CTCF and cohesin are not validated by wet assays. Many data of experimental knocking out CTCF binding sites are available; however none of current models can explain all experimental data. Maybe the authors can compare their in-silico knockout experiments against these existing data or generate more strictly controlled data.
3. In lines 402-418, the authors illustrate long short-term memory (LSTM) networks. Then LSTM is used in Eq. 3 as a decoder to predict the interacting frequency from the 1D representations of two loci. The symbols used in lines 402-418 and Eq. 3 seem to be inconsistent.

LSTM needs initial values for both the cell and memory (or hidden) states, and at each step, it calculates a new pair of the cell and memory states. In Eq. 3, R_1 and R_2 seem to be the current input, so where is the cell state? Also, which gate (the input, output, forget gate, or even none of them) gives the final interacting frequency? Eq. 3 does not clarify this.
4. I think a simple illustration of the embedding neural network layer is also necessary.
5. Perhaps Eq. 4 is incorrect.
6. Hi-C-LSTM performs worse in classifying subcompartments than SNIPER and SCI. In my opinion, one possible reason is that the LSTM in this paper reinitializes the hidden states after every frame of length 150. Such a local design may be unsuitable for predicting long-range genomic phenomena such as the subcompartments.
7. This sentence has problems: "we show that our representation captures cell type-specific functional activity, genomic elements and identifies genomic regions that drive conformation."
8. Why Hi-C-LSTM cannot identify insulators?
9. Why Hi-C-LSTM maps lost CTCF interaction dots of Hi-C maps?

2 Responses to the Comments from Reviewer 1

In this manuscript, Kevin B. Dsouza et al. proposed an LSTM-based model to embed the Hi-C intra-chromosomal contacts. The authors demonstrated that the learned representations can improve the reconstruction accuracy of the Hi-C matrix and capture cell-type-specific functional activity, genomic elements, and genomic regions that drive conformation. They also showed that Hi-C-LSTM can perform in-silico perturbation of CTCF and cohesin binding sites. However, there are several major issues to be addressed.

1) *The authors emphasized that Hi-C-LSTM representations capture the information for the reconstruction of Hi-C maps. However, the motivation of reconstruction is not well justified. Why use the representations to recreate the Hi-C matrix? Can the recreated Hi-C matrix bring additional biological insights, using standard pipelines for Hi-C data analysis? Why not compare with Hi-C data imputation methods?*

We have now added discussion on the distinction between imputation and reconstruction in the revised manuscript (Change (1)). We have elaborated on the motivation behind using reconstruction and conducting the “Hi-C reproduction evaluation” in the revised manuscript (Change (2)). We have also added discussion about extending our method to simultaneously incorporate multiple cell types as future work (Change (3)).

Changes made to the paper:

- (1) The following text has been added to section Related Work of the revised manuscript, regarding background on imputation methods and the distinction between imputation and reconstruction.

Another related task is that of imputing unseen Hi-C data sets, for which several methods have been developed. Such imputation methods include SNIPER [11], DeepHiC [22], HiCPlus [23], Higashi [24] and scHiCluster [25]. SNIPER imputes high-coverage Hi-C using moderate-coverage Hi-C at the input. DeepHiC predicts high-resolution Hi-C contact maps from low-coverage sequencing data using generative adversarial networks. HiCPlus infers high-resolution Hi-C matrices from low-resolution Hi-C data using deep convolutional neural networks. Both DeepHiC and HiCPlus, do not form position specific representations that accomplish various downstream tasks, therefore, are not comparable to our method. Higashi enhances scHi-C data quality using hypergraph representation learning. scHiCluster studies cell type-specific chromosome structural patterns in scHi-C. These methods (apart from SNIPER) cannot be used for the task of bulk Hi-C representation learning because they do not form position-specific representations and, in the case of Higashi and scHiCluster, require single-cell data.

Note that, while existing methods for Hi-C representation learning (including SCI, SNIPER and Hi-C-LSTM) utilize a reconstruction loss that aims to reconstruct the input Hi-C data, they cannot in general be used for imputation.

- (2) The following text has been added to section Discussion of the revised manuscript, talking about the motivation behind using reconstruction.

Hi-C-LSTM, in its current form, is not designed to handle data from multiple cell types. We acknowledge that imputation is an important goal that deserves consideration, however, our goal with Hi-C-LSTM is not to impute data in new cell types but rather form cell type

and position specific representations. Learning representations that help you reconstruct the Hi-C map can be useful for multiple reasons, namely, (a) the resulting model becomes a contact generation framework, (b) the resulting representations capture conformation defining elements, (c) the representations coupled with the model can be used for in-silico manipulation of genomic elements, and (d) the process can give us insights about which genomic sites are most important in construction. It is also important to note that we learn the representations in the process of reconstructing the matrix, i.e., reconstruction does not bring biological insight but is part of the process that forms the representations. The Hi-C reproduction evaluation shows how well these representations capture the information in the Hi-C matrix.

- (3) The following text has been added to section Discussion of the revised manuscript, regarding the extension of the work to handle multiple cell types.

Extending this framework to work with multiple cell types at the same time may be possible with the addition of “cell type id” as an input parameter.

2) *The performance of Hi-C-LSTM was evaluated on only one dataset, even though there are amounts of Hi-C data in the literature. The authors need to systematically assess the reconstruction performance and the classification performance with multiple Hi-C datasets from various tissues.*

We thank the reviewer for the comment. We have now trained and evaluated Hi-C-LSTM on 3 other tier 1 cell types from the 4DN Data Portal apart from GM12878, namely, H1-hESC (embryonic stem cell), WTC11 (induced pluripotent stem cell derived from skin leg fibroblast), HFF-hTERT (foreskin fibroblast immortalized cell). We compared the reconstruction (Changes (4), (5), (8), (9), and (10)) and classification (Changes (6), (7), (8), (11), and (12)) performance of the models derived from these cell types. We also compared the performance of Hi-C-LSTM with other methods (SCI, SNIPER) and baselines (PCA, SUBCOMPARTMENT-ID) in these cell types. We observe that the performance in these cell types is linked closely to sequencing depth (Changes (9), (10), (11), and (12)).

Changes made to the paper:

- (4) Figure 1 (Fig. 3) has been added to section Results (Hi-C reproduction) of the revised manuscript, showing reconstruction performance in different cell types and experiments with reduced read depth.

- (5) The following text has been added to section Results (Hi-C reproduction) of the revised manuscript, talking about the reconstruction performance in different tier 1 cell types and experiments with reduced read depth.

We discovered a relationship between sequencing depth and model performance after training and evaluating Hi-C-LSTM on Hi-C datasets from GM12878 with a combined filtered reads of 300 million and 216 million (compared to Hi-C data from Rao. et al. which had 3 billion combined filtered reads). We saw that Hi-C-LSTM R2 worsened with reduced read depth, however, the reconstruction performance trend over distance was preserved (Fig. 3e,f).

We also trained and evaluated Hi-C-LSTM on data from 3 other tier 1 cell types from the 4DN Data Portal apart from GM12878, namely, H1-hESC (embryonic stem cell) (Fig. 3c,d), WTC11 (induced pluripotent stem cell derived from skin leg fibroblast) (Supplementary:

Figure 1: Accuracy with which representations reproduce the observed Hi-C matrix. **a,b)** The Hi-C R-squared computed using the combinations of representations from different methods and selected decoders for replicate 1 and 2 (GM12878). The horizontal axis represents the distance between positions in Mbp. The vertical axis shows the reconstruction accuracy for the predicted Hi-C data, measured by average R-squared. The R-squared was computed on a test set of chromosomes using selected decoders with the representations trained all chromosomes as input. The legend shows the different combinations of methods and decoders, read as *[representation]-[decoder]*. **c,d)** Same as **a,b**, but for H1-hESC. **e,f)** Hi-C R-squared computed for different cell types. **g)** A selected portion of the observed Hi-C map (upper-triangle) and the predicted Hi-C map (lower-triangle) in GM12878. The portion is selected from chromosome 21, between 40 Mbp to 43 Mbp. Diagonal black lines denote Hi-C-LSTM’s frame boundaries (Methods).

Fig. 10c,d), and HFF-hTERT (foreskin fibroblast immortalized cell) (Supplementary: Fig. 10a,b). We found that difference in performance across these cell types can be explained by their differing read depths. These data sets have varying read depths, ranging from 150 million (HFFhTERT) to 900 million (H1hESC) filtered reads. We saw that the R2 fell by 0.03 points on average when reads reduced from 3 billion to 1 billion (Fig. 3e). The performance further reduced by 0.4 on average when the reads reduced to 150 million. This amounted to a total R2 decrease of 0.7 points on average with very low sequencing depth (Fig. 3e). We additionally found that the reconstruction performance trend between models is preserved across cell types (Supplementary: Fig. 10).

(6) Figure 2 (Fig. 4) has been added to section Results (classification) of the revised manuscript,

showing classification performance in different cell types and experiments with reduced read depth.

Figure 2: Genomic phenomena and chromatin regions are classified using the Hi-C-LSTM representations as input. **a)** Prediction accuracy for gene expression, replication timing, enhancers, transcription start sites (TSSs), promoter-enhancer interactions (PEIs), frequently interacting regions (FIREs), loop and non-loop domains, and subcompartments in GM12878. The y-axis shows the mean average precision (mAP), the x-ticks refer to the prediction targets, and the legend shows the different methods compared with. **b)** Same as **a)**, but for targets in H1-hESC. **c)** mAP using Hi-C-LSTM for targets compared across cell types. **d)** The Precision-Recall curves of Hi-C-LSTM for the various prediction targets in GM12878. The y-axis shows the Precision, the x-axis shows the Recall, and the legend shows the prediction targets.

(7) The following text has been added to section Results (classification) of the revised manuscript,

talking about the classification performance in different cell types and its relationship to reduced read depth.

Similar to reconstruction, when comparing classification performance across cell types, we saw that Hi-C-LSTM accuracy worsened with reduced read depth. However, the classification performance trend over tasks was preserved (Fig. 4c). We include results for all available data sets for each cell type. We omitted WTC11 from this analysis because most data sets are not available (see Methods for details regarding element specific data in cell types). We observed that the accuracy reduced by 0.6 units on average when the reads reduced to 150 million (Fig. 4c). Next, we compared the classification performance of Hi-C-LSTM with other methods (SCI, SNIPER) and baselines (PCA, SUBCOMPARTMENT-ID) in these cell types (Supplementary: Fig. 7,8,9). Similar to R2, we saw that the prediction score trend of methods is preserved across all these cell types.

- (8) The following text has been added to section Methods (datasets) of the revised manuscript mentioning new datasets. Table 1 (Supplementary: Table 3) has been added to the supplementary showing links to datasets.

The Hi-C data for other tier 1 cell types like H1-hESC, WTC11, and HFF-hTERT was acquired from the 4DN Data Portal. The Hi-C data for GM12878 with lower read depths of 300M and 216M were also downloaded from the 4DN Data Portal. For a full list of links to relevant datasets, refer to Table 3 in the Supplementary.

RNA-seq data for GM12878, H1-hESC, and HFF-hTERT was obtained from the Roadmap Consortium (see Supplementary: Table 3).

We could not find PEI, FIRE, and replication timing data for H1-hESC and HFF-hTERT, and hence excluded these elements from classification performance evaluation in H1-hESC and HFF-hTERT.

For GM12878, H1-hESC, and HFF-hTERT, locations of known enhancers and transcription start sites (TSSs) were obtained from FANTOM and ENCODE respectively (see Supplementary: Table 3).

For GM12878, Loop Domains and Subcompartments were obtained from the results of Rao et al. [20] using the GEO accession number GSE63525 (see Supplementary: Table 3). For H1-hESC and HFF-hTERT, Loop Domains were obtained by running HICUPS [20] and Subcompartments were obtained by running Gaussian HMM [20].

For GM12878, Topologically-associating domains (TADs) were downloaded from TADKB [92] and subTADs were obtained by running GMAP [93]. For H1-hESC and HFF-hTERT, both TADs and subTADs were obtained by running GMAP [93].

- (9) Figure 3 (Supplementary: Fig. 10) has been added to the supplementary, showing R-squared in HFF-hTERT and WTC11.

- (10) The following text has been added to section Methods (Hi-C reproduction evaluation) of the revised manuscript.

We conducted both these experiments in all 4 cell types, namely, GM12878 (Figure 3a,b), H1-hESC (Figure 3c,d), WTC11 (Supplementary: Fig.10c,d), and HFF-hTERT (Supplementary: Fig.10a,b).

- (11) Figures 4, 5, 6 (Supplementary: Fig. 7,8,9) have been added to the supplementary, showing additional classification metrics in GM12878, H1-hESC, and HFF-hTERT. We ignored WTC11 as we could not find relevant data about elements.

Dataset	Link
GM12878 Hi-C	https://www.ncbi.nlm.nih.gov/geo/query/acc.cgi?acc=GSE63525
H1-hESC Hi-C	https://data.4dnucleome.org/experiment-set-replicates/4DNES2M5JIGV/
WTC11 Hi-C	https://data.4dnucleome.org/experiment-set-replicates/4DNESPDEZNX/
HFF-hTERT Hi-C	https://data.4dnucleome.org/experiment-set-replicates/4DNESVUMGLG2/
GM12878 Hi-C (300M)	https://data.4dnucleome.org/experiment-set-replicates/4DNESJFTAURO/
GM12878 Hi-C (216M)	https://data.4dnucleome.org/experiment-set-replicates/4DNESLQG7ZKJ/
Juicer Tools	https://github.com/aidenlab/juicer/wiki/Juicer-Tools-Quick-Start
RNA-seq (GM12878, H1-hESC, HFF-hTERT)	https://egg2.wustl.edu/roadmap/data/byDataType/rna/expression/
Promoter-Enhancer Interactions (GM12878)	https://github.com/shwhalen/targetfinder
Replication Timing (GM12878)	https://www2.replicationdomain.com
Enhancers (GM12878, H1-hESC, HFF-hTERT)	https://fantom.gsc.riken.jp/5
Transcription Start Sites (GM12878, H1-hESC, HFF-hTERT)	https://www.encodeproject.org/files/ENCFF140PCA
Loop Domains and Subcompartments (GM12878)	https://www.ncbi.nlm.nih.gov/geo/query/acc.cgi?acc=GSE63525
Segway Labels	https://segway.hoffmanlab.org
CTCF and Cohesin Peaks (GM12878)	https://www.encodeproject.org
CTCF Motif	https://meme-suite.org/meme/doc/fimo.html
Transcription Factor Binding Sites	http://humantfs.cabr.utoronto.ca/
Pseudo-Bulk Single-Cell Hi-C	https://www.ncbi.nlm.nih.gov/geo/query/acc.cgi?acc=GSM2254215

Table 1

Figure 3: Accuracy with which representations reproduce the original Hi-C matrix. **a,b)** The Hi-C R-squared computed using the combinations of representations from different methods and selected decoders for replicate 1 and 2 (HFF-hTERT). The horizontal axis represents the distance between positions in Mbp. The vertical axis shows the average R-squared for the predicted Hi-C data. The R-squared was computed on a test set of chromosomes using selected decoders with the representations trained all chromosomes as input. The legend shows the different combinations of methods and decoders, read as *[representation]-[decoder]*. **c,d)** Same as **a,b**, but for WTC11.

Figure 4: Additional classification metrics in GM12878 for gene expression, replication timing, enhancers, transcription start sites (TSSs), promoter-enhancer interactions (PEIs), frequently interacting regions (FIRES), topologically associating domains (TADs), subTADs, loop domains, TAD boundaries, subTAD boundaries, and subcompartments. **a, b, c**) The area under the receiver operating characteristic curve (AuROC), F-score, and Accuracy for different prediction targets. The y-axis shows the metrics, the x-ticks refer to the prediction targets, and the legend shows the different methods compared with.

Figure 5: Additional classification metrics in H1-hESC for gene expression, enhancers, transcription start sites (TSSs), topologically associating domains (TADs), subTADs, loop domains, TAD boundaries, subTAD boundaries, and subcompartments. **a, b, c**) The area under the receiver operating characteristic curve (AuROC), F-score, and Accuracy for different prediction targets. The y-axis shows the metrics, the x-ticks refer to the prediction targets, and the legend shows the different methods compared with.

Figure 6: Additional classification metrics in HFF-hTERT for gene expression, enhancers, transcription start sites (TSSs), topologically associating domains (TADs), subTADs, loop domains, TAD boundaries, subTAD boundaries, and subcompartments. **a, b, c, d**) mean average precision (mAP), The area under the receiver operating characteristic curve (AuROC), F-score, and Accuracy for different prediction targets. The y-axis shows the metrics, the x-ticks refer to the prediction targets, and the legend shows the different methods compared with.

- (12) The following text has been added to section Methods (element identification evaluation) of the revised manuscript, talking about the classification metrics.

The classifier was evaluated using four standard metrics for classification tasks, namely, mean average precision (mAP) (otherwise known as area under the Precision-Recall curve (AuPR)), area under the Receiver Operating Characteristic curve (AuROC), Accuracy ($A = \frac{TP+TN}{TP+FP+TN+FN}$), and F-score. AuROC is defined as the area under the curve that has true positive rate ($TPR = \frac{TP}{TP+FN}$) on the y-axis and false positive rate ($FPR = \frac{FP}{FP+TN}$) on the x-axis. mAP is defined as the average of the maximum precision ($P = \frac{TP}{TP+FP}$) scores achieved at varying recall levels ($R = TPR$). F-score is defined based on precision and recall ($F = \frac{2P*R}{P+R}$). We compared these metrics for GM12878, H1-hESC, and HFF-hTERT (see Supplementary: Fig. 7,8,9 for more details).

- 3) *In addition, it is interesting to test the performance on datasets with various sequencing depths, for example, by downsampling. Besides, can Hi-C-LSTM be used on single-cell Hi-C data?*

We thank the reviewer for pointing this out. We have now trained and evaluated our model on Hi-C datasets from GM12878 with 300 million and 216 million filtered reads (Erez Aiden, 4DN), compared to our original Hi-C dataset from Rao. et al., which had 3 billion filtered reads. Our observations regarding model performance and sequencing depth, and their relationship to performance in other tier 1 cell types now summarised (Changes (4) - (12)).

We agree with the reviewer that single-cell Hi-C datasets are valuable in providing us with cell level resolution of contacts. Hi-C-LSTM cannot utilize single-cell Hi-C data—unless that data is aggregated into pseudo-bulk Hi-C—because it takes as input a single Hi-C matrix. We have now included a sample result from training and evaluating Hi-C-LSTM on pseudo-bulk single-cell HiC data from Ramani et al. [2] (Changes (13), (14), and (15)). We leave representation learning for single-cell Hi-C data for future work (Change (16)).

Changes made to the paper:

- (13) Figure 7 (Supplementary: Fig. 12) has been added to the supplementary, showing Hi-C-LSTM predictions in scHi-C.

- (14) The following text has been added to section Discussion of the revised manuscript, talking about running Hi-C-LSTM on pseudo-bulk single-cell Hi-C data.

Single-cell Hi-C (scHi-C) datasets are becoming increasingly valuable in providing us with cell level resolution of contacts. Hi-C-LSTM cannot utilize single-cell Hi-C data—unless that data is aggregated into pseudo-bulk Hi-C—because it takes as input a single Hi-C matrix. Additionally, the model would have to be trained on such data by taking into consideration the data resolution, mapping it to appropriate bins and mapping the contacts between those bins. The main challenge when analysing scHi-C is that the data is extremely sparse. Pseudo-bulk scHi-C, where many cells are clustered into groups of similar types and pooled in silico, allows for the statistical validation of chromatin patterns. We trained and evaluated Hi-C-LSTM on a subset of chromosomes (15-22) using pseudo-bulk scHi-C data from Ramani et al. [82] (see Methods for details). A representative heatmap from chromosome 21 shows that Hi-C-LSTM is able to reconstruct contacts faithfully (Supplementary: Fig. 12), however, the sparsity of scHi-C data might be a potential concern when using the representations from

Figure 7: Hi-C-LSTM applied to pseudo-bulk single-cell Hi-C (scHi-C) data. A selected portion of the observed scHi-C map (upper-triangle) [1] and the predicted scHi-C map (lower-triangle). The portion is selected from chromosome 21, between 43.2 Mbp to 48.1 Mbp. Hi-C-LSTM does a good job of recapitulating the sparse structures in the observed scHi-C. The ability of Hi-C-LSTM to handle such sparse data alludes to its prowess in reconstructing data from minimal number of data points.

scHi-C models for other downstream tasks like classification and in-silico manipulation.

- (15) The following text has been added to section Methods (datasets) of the revised manuscript, regarding processing of single-cell Hi-C data.

Pseudo-bulk single-cell Hi-C (scHi-C) data was downloaded using the GEO accession number GSM2254215 (see Supplementary: Table 3). We used the validPairs file to filter the data based on chromosome, positions and barcodes. We then obtained the number of reads aligned to hg19 corresponding to each distinct pair of bar codes from the percentages file.

- (16) The following text has been added to section Discussion of the revised manuscript, pointing to a detailed analysis of Hi-C-LSTM on scHi-C as part of future work.

Second, an in-depth analysis of Hi-C-LSTM performance on scHi-C warrants a detailed report of its own.

4) The authors should elaborate on the data used for classification (Figure 4) in Methods Section. For example, enhancers of which cell type were used? How many negative samples were sampled? Besides, in addition to mAP, it is good to provide other metrics, e.g., Area Under the Receiver Operating Characteristic Curve and Area Under Curve.

We thank the reviewer for the comment. We have now included more details regarding the elements used for classification (Change (8)). We have also clarified the negative sampling strategy used for classification (Change (17)). Moreover, in addition to mAP (Area under the PR curve, AuPR), we have now included Area under the Receiver Operating Characteristic Curve (AuROC), Accuracy, and F-score as additional classification metrics (Change (12)). We also show additional metrics in different cell types (Change (11)).

Changes made to the paper:

- (17) The following text has been added to section Methods (element identification evaluation) of the revised manuscript, talking more about the negative sampling strategy.

We sampled negative labels until the number of negative labels equaled the number of positive labels to avoid class imbalance during classification.

5) It is good to provide the running time of different methods, and the robustness to Hi-C data resolution.

We thank the reviewer for the comment. We have now added the running times of different methods (Changes (18) and (19)).

We chose to focus on 10Kbp because, (1) we wanted the model to take into account the larger chromatin structures like TADs and loop domains that are integral to conformation, while at the same time not ignoring smaller elements like genes, and (2) running it at a very fine resolution was turning out to be prohibitively expensive for us. Following your comment, we have now trained Hi-C-LSTM and evaluated its classification performance on 3 different resolutions apart from 10Kbp, namely, 2Kbp, 100Kbp, and 500Kbp (Changes (20), (21), and (22)).

Changes made to the paper:

- (18) The following text has been added to section Methods (modeling choices and training) of the revised manuscript, pointing to the running times of different methods.

The time taken to train and test all methods is included in the Supplementary: Table 1 (Running Time).

- (19) Table 2 (Supplementary: Table 1) has been added to section Running Time of the supplementary, tabulating the training and test times of different methods.

Method	Training time in hrs	Test time in mns
Hi-C-LSTM	2.549	14.37
SNIPER	1.805	8.61
SCI	2.438	12.14

Table 2: Running times of different methods for the whole genome. The training time is given in hours and the test time is given in minutes. As SNIPER uses only a feedforward neural network without considering a sequential input, it has much better training and test running times compared to SCI and Hi-C-LSTM. Hi-C-LSTM running time is comparable to SCI both during training and inference. SCI uses a graph embedding algorithm called LINE (with parameters mentioned in SCI) that roughly takes the same time to run as an LSTM with frame length on the order of 100. We verify this by increasing LSTM frame length and observe that both training and inference times get worse with increasing frame length.

- (20) Figure 8 (Fig. 5) has been added to section Results (Resolution) of the revised manuscript, showing Hi-C-LSTM predictions with different Hi-C resolutions.
- (21) The following section has been added to Results of the revised manuscript, talking about results from running Hi-C-LSTM at different resolutions.

Hi-C-LSTM recapitulates structures at different Hi-C resolutions

To check if Hi-C-LSTM works at different resolutions of Hi-C data, in addition to our model trained at 10Kbp, we trained Hi-C-LSTM at three other resolutions of 2Kbp, 100Kbp and 500Kbp. As expected, models at different scales detect different elements, classify differently, and attribute importance to different sites depending on the resolution. The models achieved these by forming representations that allowed them to construct the Hi-C map at the given resolution. We investigate how these representations differ from the ones learned at 10Kbp.

To train the model at 2Kbp, we used only a subset of chromosomes due to memory and compute constraints but trained on the whole genome at other resolutions. A selected portion in chromosome 21 (Fig. 5a) shows that the predicted Hi-C values capture the fine structure of Hi-C even at 2Kbp resolution. The sparsity of available data at 2Kbp is a major constraint in enhancing the performance of the model at this resolution. Hi-C-LSTM captures the Hi-C macro-structures accurately at 100Kbp (Fig. 5c) and 500Kbp (Fig. 5b). This is because operating at this resolution with our sequence length allows it to span entire smaller chromosomes.

We found that representations at different resolutions predict chromatin structures of different scales. The classification performance (as measured in mAP) with gene expression, enhancers, TADs, subTADs, and subcompartments of models trained at different resolutions (Fig. 5d), shows that for small scale phenomena and expression like gene expression and enhancers, the cumulative prediction score worsens with coarser resolution as expected. For enhancers, the prediction score drops by 0.22 units when the resolution goes from 2Kbp to 500Kbp (Fig. 5d). Both with TADs and subTADs, the model at 100Kbp has the best prediction score, closely followed by the model at 10Kbp (Fig. 5d). We attribute this performance to the fact that these resolutions, combined with our frame length of 150, are close to the averages sizes of both these elements. The model at 500 Kbp performs best at identifying subcompartments given that the average size of subcompartments is 300Kbp (Fig. 5d). These results point to the idea that aggregating representations learnt at different Hi-C resolutions

Figure 8: Hi-C-LSTM applied at different resolutions. **a**) Hi-C-LSTM predictions at 2Kbp resolution. A selected portion of the observed Hi-C map (upper-triangle) and the predicted Hi-C map (lower-triangle) in GM12878. The portion is selected from chromosome 21, between 43.2 Mbp to 48.1 Mbp. **b**) Hi-C-LSTM predictions at 500Kbp resolution. The observed Hi-C map (upper-triangle) and the predicted Hi-C map (lower-triangle) in GM12878 for chromosome 21. **c**) Hi-C-LSTM predictions at 100Kbp resolution. The observed Hi-C map (upper-triangle) and the predicted Hi-C map (lower-triangle) in GM12878 for chromosome 21. **d**) The classification performance (as measured in mAP) with gene expression, enhancers, TADs, subTADs, and subcompartments of models trained at different resolutions.

will likely increase prediction performance across elements of all sizes. Such aggregation will also potentially help in alleviating computational bottlenecks, as a model at a particular resolution need not take the broader context into account (see Discussion).

(22) The following text has been added to section Discussion of the revised manuscript, elaborating on running Hi-C-LSTM at different resolutions.

Third, combining representations from models trained at varying resolutions to form a common representation would allow us to not only discover new elements at different scales but also form a comprehensive scale agnostic representation.

6) *Feature attribution showed the trend for the identification of TADs. Can Hi-C-LSTM identify TADs via feature attribution but not just the trend? If so, will the performance outperform other methods for TAD identification?*

We thank the reviewer for pointing this out. Hi-C-LSTM can identify TADs directly from its representations and does not need to rely on feature attribution scores. This is shown in our new classification plot (in the previous version TADs were combined with other domains but now we have separated them into TADs, subTADs, and Loop Domains)(Change (6)). The TAD classification performance is seen to be better when compared to representations from SCI, SNIPER.

We cannot compare our TAD classification performance with computational TAD callers (Arrowhead, GMAP, DI, TopDom, CHDF, 3DNetMod, and HiCseg among others) because we trained our classifier using these TAD callers as the gold standard training set. We have now elaborated on this in the revised manuscript (Change (23)).

Changes made to the paper:

(23) The following text has been added to the section Discussion of the revised manuscript, talking about comparison with computational TAD callers.

Although we do a good job of identifying TADs, we cannot use our method as a computational TAD caller (like Arrowhead, GMAP, HiCseg among others) [75] because we need a gold standard training set to train our classification model, which we obtain from the aforementioned TAD callers. As a result, we also cannot compare our TAD classification performance with these TAD callers.

3 Responses to the Comments from Reviewer 2

The submitted manuscript undertakes the challenge of discerning the genomic relationship between the 1D sequence’s features and the 3D conformation to understand their effect on gene expression and disease. Using a recurrent long short-term memory (LSTM) neural network model, the authors develop Hi-C-LSTM. The new method summarizes the intra-chromosomal Hi-C contacts information into low-dimensional sequence latent representation. The authors show that the resulting sequence representations contain enough information to recap the original Hi-C matrix with high accuracy and present positive results compared to other existing methods. The method enables the identification of known genomic elements related to 3D structure conformation, including transcription factor (TF) binding sites associated with loop conformation and nuclear compartments. The authors show the use of the method for in-silico elements’ perturbation experiments to measure the influence of genomic loci on the 3D Conformation. This reviewer has several comments that the authors should address to improve the quality of the manuscript for further revision before considering the manuscript for acceptance.

Major Comments:

1) *The 3D Genome Browser [Wang, Y. Genome Biology. 2018] is a rich source of 3D chromatin interaction data that this paper does not utilize in any way. This reviewer believes that this study should further exploit the current availability of Hi-C data.*

a) *The authors only check the efficacy of Hi-C-LSTM representations by analyzing a biological replicate in predicting a Hi-C contact map (Fig. 3a, b). TADs are conserved building blocks of genomes with regulatory functions. Thus, authors can enhance their analysis by investigating the method’s performance using several Hi-C datasets to train the representations and decoders then obtain the corresponding prediction. That work could inform about what type of interaction is more likely to be captured by the method.*

We thank the reviewer for the comment. We have now trained and evaluated Hi-C-LSTM on 3 other tier 1 cell types from the 4DN Data Portal apart from GM12878, namely, H1-hESC (embryonic stem cell), WTC11 (induced pluripotent stem cell derived from skin leg fibroblast), HFF-hTERT (foreskin fibroblast immortalized cell). We compared the reconstruction (Changes (24), (25), (28), (29), and (30)) and classification (Changes (26), (27), (28), (31), and (32)) performance of the models derived from these cell types. We also compared the performance of Hi-C-LSTM with other methods (SCI, SNIPER) and baselines (PCA, SUBCOMPARTMENT-ID) in these cell types. We observe that the performance in these cell types is linked closely to sequencing depth (Changes (29), (30), (31), and (32)).

We have now also included a sample result from training and evaluating Hi-C-LSTM on pseudo-bulk single-cell HiC data from Ramani et al. [2] (Changes (33), (34), and (35)).

It is important to note that Hi-C-LSTM is not designed to handle data from multiple cell types at the same time at the moment because it does not use “cell type id” as an input parameter. Our goal with Hi-C-LSTM is not to design a single model that works across cell types but rather form cell type and position specific representations. We have now added discussion on this in the revised manuscript (Changes (36), (37), and (38)).

We agree with the reviewer that understanding what kind of interactions the model is more likely to capture is vital. We have now added discussion in the revised manuscript that talks about this (Change (39)).

The 3D genome browser mostly houses contact map data and for our experiments we had already acquired contact map data from [3], 4DN Data Portal, and [2]. We use the UCSC genome

browser (see Q5b), however, we have now mentioned the 3D genome browser in the manuscript for the sake of completeness (Change (40)).

Changes made to the paper:

- (24) Figure 9 (Fig. 3) has been added to section Results (Hi-C reproduction) of the revised manuscript, showing reconstruction performance in different cell types and experiments with reduced read depth.

Figure 9: Accuracy with which representations reproduce the observed Hi-C matrix. **a,b)** The Hi-C R-squared computed using the combinations of representations from different methods and selected decoders for replicate 1 and 2 (GM12878). The horizontal axis represents the distance between positions in Mbp. The vertical axis shows the reconstruction accuracy for the predicted Hi-C data, measured by average R-squared. The R-squared was computed on a test set of chromosomes using selected decoders with the representations trained all chromosomes as input.

The legend shows the different combinations of methods and decoders, read as *[representation]-[decoder]*. **c,d)** Same as **a,b**, but for H1-hESC. **e,f)** Hi-C R-squared computed for different cell types. **g)** A selected portion of the observed Hi-C map (upper-triangle) and the predicted Hi-C map (lower-triangle) in GM12878. The portion is selected from chromosome 21, between 40 Mbp to 43 Mbp. Diagonal black lines denote Hi-C-LSTM's frame boundaries (Methods).

- (25) The following text has been added to section Results (Hi-C reproduction) of the revised manuscript, talking about the reconstruction performance in different tier 1 cell types and experiments with reduced read depth.

We discovered a relationship between sequencing depth and model performance after training and evaluating Hi-C-LSTM on Hi-C datasets from GM12878 with a combined filtered reads of 300 million and 216 million (compared to Hi-C data from Rao. et al. which had 3 billion combined filtered reads). We saw that Hi-C-LSTM R2 worsened with reduced read depth, however, the reconstruction performance trend over distance was preserved (Fig. 3e,f).

We also trained and evaluated Hi-C-LSTM on data from 3 other tier 1 cell types from the 4DN Data Portal apart from GM12878, namely, H1-hESC (embryonic stem cell) (Fig. 3c,d), WTC11 (induced pluripotent stem cell derived from skin leg fibroblast) (Supplementary: Fig. 10c,d), and HFF-hTERT (foreskin fibroblast immortalized cell) (Supplementary: Fig. 10a,b). We found that difference in performance across these cell types can be explained by their differing read depths. These data sets have varying read depths, ranging from 150 million (HFFhTERT) to 900 million (H1hESC) filtered reads. We saw that the R2 fell by 0.03 points on average when reads reduced from 3 billion to 1 billion (Fig. 3e). The performance further reduced by 0.4 on average when the reads reduced to 150 million. This amounted to a total R2 decrease of 0.7 points on average with very low sequencing depth (Fig. 3e). We additionally found that the reconstruction performance trend between models is preserved across cell types (Supplementary: Fig. 10).

(26) Figure 10 (Fig. 4) has been added to section Results (classification) of the revised manuscript, showing classification performance in different cell types and experiments with reduced read depth.

(27) The following text has been added to section Results (classification) of the revised manuscript, talking about the classification performance in different cell types and its relationship to reduced read depth.

Similar to reconstruction, when comparing classification performance across cell types, we saw that Hi-C-LSTM accuracy worsened with reduced read depth. However, the classification performance trend over tasks was preserved (Fig. 4c). We include results for all available data sets for each cell type. We omitted WTC11 from this analysis because most data sets are not available (see Methods for details regarding element specific data in cell types). We observed that the accuracy reduced by 0.6 units on average when the reads reduced to 150 million (Fig. 4c). Next, we compared the classification performance of Hi-C-LSTM with other methods (SCI, SNIPER) and baselines (PCA, SUBCOMPARTMENT-ID) in these cell types (Supplementary: Fig. 7,8,9). Similar to R2, we saw that the prediction score trend of methods is preserved across all these cell types.

(28) The following text has been added to section Methods (datasets) of the revised manuscript mentioning new datasets. Table 3 (Supplementary: Table 3) has been added to the supplementary showing links to datasets.

The Hi-C data for other tier 1 cell types like H1-hESC, WTC11, and HFF-hTERT was acquired from the 4DN Data Portal. The Hi-C data for GM12878 with lower read depths of 300M and 216M were also downloaded from the 4DN Data Portal. For a full list of links to relevant datasets, refer to Table 3 in the Supplementary.

RNA-seq data for GM12878, H1-hESC, and HFF-hTERT was obtained from the Roadmap Consortium (see Supplementary: Table 3).

Figure 10: Genomic phenomena and chromatin regions are classified using the Hi-C-LSTM representations as input. **a**) Prediction accuracy for gene expression, replication timing, enhancers, transcription start sites (TSSs), promoter-enhancer interactions (PEIs), frequently interacting regions (FIREs), loop and non-loop domains, and subcompartments in GM12878. The y-axis shows the mean average precision (mAP), the x-ticks refer to the prediction targets, and the legend shows the different methods compared with. **b**) Same as **a**, but for targets in H1-hESC. **c**) mAP using Hi-C-LSTM for targets compared across cell types. **d**) The Precision-Recall curves of Hi-C-LSTM for the various prediction targets in GM12878. The y-axis shows the Precision, the x-axis shows the Recall, and the legend shows the prediction targets.

We could not find PEI, FIRE, and replication timing data for H1-hESC and HFF-hTERT, and hence excluded these elements from classification performance evaluation in H1-hESC and HFF-hTERT.

For GM12878, H1-hESC, and HFF-hTERT, locations of known enhancers and transcription

start sites (TSSs) were obtained from FANTOM and ENCODE respectively (see Supplementary: Table 3).

For GM12878, Loop Domains and Subcompartments were obtained from the results of Rao et al. [20] using the GEO accession number GSE63525 (see Supplementary: Table 3). For H1-hESC and HFF-hTERT, Loop Domains were obtained by running HICCUPS [20] and Subcompartments were obtained by running Gaussian HMM [20].

For GM12878, Topologically-associating domains (TADs) were downloaded from TADKB [92] and subTADs were obtained by running GMAP [93]. For H1-hESC and HFF-hTERT, both TADs and subTADs were obtained by running GMAP [93].

(29) Figure 11 (Supplementary: Fig. 10) has been added to the supplementary, showing R-squared in HFF-hTERT and WTC11.

(30) The following text has been added to section Methods (Hi-C reproduction evaluation) of the revised manuscript.

We conducted both these experiments in all 4 cell types, namely, GM12878 (Figure 3a,b), H1-hESC (Figure 3c,d), WTC11 (Supplementary: Fig.10c,d), and HFF-hTERT (Supplementary: Fig.10a,b).

(31) Figures 12, 13, 14 (Supplementary: Fig. 7,8,9) have been added to the supplementary, showing additional classification metrics in GM12878, H1-hESC, and HFF-hTERT. We ignored WTC11 as we could not find relevant data about elements.

(32) The following text has been added to section Methods (element identification evaluation) of the revised manuscript, talking about the classification metrics.

The classifier was evaluated using four standard metrics for classification tasks, namely, mean average precision (mAP) (otherwise known as area under the Precision-Recall curve (AuPR)), area under the Receiver Operating Characteristic curve (AuROC), Accuracy ($A = \frac{TP+TN}{TP+FP+TN+FN}$), and F-score. AuROC is defined as the area under the curve that has true positive rate ($TPR = \frac{TP}{TP+FN}$) on the y-axis and false positive rate ($FPR = \frac{FP}{FP+TN}$) on the x-axis. mAP is defined as the average of the maximum precision ($P = \frac{TP}{TP+FP}$) scores achieved at varying recall levels ($R = TPR$). F-score is defined based on precision and recall ($F = \frac{2P*R}{P+R}$). We compared these metrics for GM12878, H1-hESC, and HFF-hTERT (see Supplementary: Fig. 7,8,9 for more details).

Dataset	Link
GM12878 Hi-C	https://www.ncbi.nlm.nih.gov/geo/query/acc.cgi?acc=GSE63525
H1-hESC Hi-C	https://data.4dnucleome.org/experiment-set-replicates/4DNES2M5JIGV/
WTC11 Hi-C	https://data.4dnucleome.org/experiment-set-replicates/4DNESPDEZNX/
HFF-hTERT Hi-C	https://data.4dnucleome.org/experiment-set-replicates/4DNESVUMGLG2/
GM12878 Hi-C (300M)	https://data.4dnucleome.org/experiment-set-replicates/4DNESJFTAURO/
GM12878 Hi-C (216M)	https://data.4dnucleome.org/experiment-set-replicates/4DNESLQG7ZKJ/
Juicer Tools	https://github.com/aidenlab/juicer/wiki/Juicer-Tools-Quick-Start
RNA-seq (GM12878, H1-hESC, HFF-hTERT)	https://egg2.wustl.edu/roadmap/data/byDataType/rna/expression/
Promoter-Enhancer Interactions (GM12878)	https://github.com/shwhalen/targetfinder
Replication Timing (GM12878)	https://www2.replicationdomain.com
Enhancers (GM12878, H1-hESC, HFF-hTERT)	https://fantom.gsc.riken.jp/5
Transcription Start Sites (GM12878, H1-hESC, HFF-hTERT)	https://www.encodeproject.org/files/ENCFF140PCA
Loop Domains and Subcompartments (GM12878)	https://www.ncbi.nlm.nih.gov/geo/query/acc.cgi?acc=GSE63525
Segway Labels	https://segway.hoffmanlab.org
CTCF and Cohesin Peaks (GM12878)	https://www.encodeproject.org
CTCF Motif	https://meme-suite.org/meme/doc/fimo.html
Transcription Factor Binding Sites	http://humantfs.cabr.utoronto.ca/
Pseudo-Bulk Single-Cell Hi-C	https://www.ncbi.nlm.nih.gov/geo/query/acc.cgi?acc=GSM2254215

Table 3

Figure 11: Accuracy with which representations reproduce the original Hi-C matrix. **a,b)** The Hi-C R-squared computed using the combinations of representations from different methods and selected decoders for replicate 1 and 2 (HFF-hTERT). The horizontal axis represents the distance between positions in Mbp. The vertical axis shows the average R-squared for the predicted Hi-C data. The R-squared was computed on a test set of chromosomes using selected decoders with the representations trained all chromosomes as input. The legend shows the different combinations of methods and decoders, read as *[representation]-[decoder]*. **c,d)** Same as **a,b)**, but for WTC11.

Figure 12: Additional classification metrics in GM12878 for gene expression, replication timing, enhancers, transcription start sites (TSSs), promoter-enhancer interactions (PEIs), frequently interacting regions (FIREs), topologically associating domains (TADs), subTADs, loop domains, TAD boundaries, subTAD boundaries, and subcompartments. **a, b, c**) The area under the receiver operating characteristic curve (AuROC), F-score, and Accuracy for different prediction targets. The y-axis shows the metrics, the x-ticks refer to the prediction targets, and the legend shows the different methods compared with.

Figure 13: Additional classification metrics in H1-hESC for gene expression, enhancers, transcription start sites (TSSs), topologically associating domains (TADs), subTADs, loop domains, TAD boundaries, subTAD boundaries, and subcompartments. **a, b, c**) The area under the receiver operating characteristic curve (AuROC), F-score, and Accuracy for different prediction targets. The y-axis shows the metrics, the x-ticks refer to the prediction targets, and the legend shows the different methods compared with.

Figure 14: Additional classification metrics in HFF-hTERT for gene expression, enhancers, transcription start sites (TSSs), topologically associating domains (TADs), subTADs, loop domains, TAD boundaries, subTAD boundaries, and subcompartments. **a, b, c, d**) mean average precision (mAP), The area under the receiver operating characteristic curve (AuROC), F-score, and Accuracy for different prediction targets. The y-axis shows the metrics, the x-ticks refer to the prediction targets, and the legend shows the different methods compared with.

(33) Figure 15 (Supplementary: Fig. 12) has been added to the supplementary, showing Hi-C-LSTM predictions in scHi-C.

(34) The following text has been added to section Discussion of the revised manuscript, talking about running Hi-C-LSTM on pseudo-bulk single-cell Hi-C data.

Figure 15: Hi-C-LSTM applied to pseudo-bulk single-cell Hi-C (scHi-C) data. A selected portion of the observed scHi-C map (upper-triangle) [1] and the predicted scHi-C map (lower-triangle). The portion is selected from chromosome 21, between 43.2 Mbp to 48.1 Mbp. Hi-C-LSTM does a good job of recapitulating the sparse structures in the observed scHi-C. The ability of Hi-C-LSTM to handle such sparse data alludes to its prowess in reconstructing data from minimal number of data points.

Single-cell Hi-C (scHi-C) datasets are becoming increasingly valuable in providing us with cell level resolution of contacts. Hi-C-LSTM cannot utilize single-cell Hi-C data—unless that data is aggregated into pseudo-bulk Hi-C—because it takes as input a single Hi-C matrix. Additionally, the model would have to be trained on such data by taking into consideration the data resolution, mapping it to appropriate bins and mapping the contacts between those bins. The main challenge when analysing scHi-C is that the data is extremely sparse. Pseudo-bulk scHi-C, where many cells are clustered into groups of similar types and pooled in silico, allows for the statistical validation of chromatin patterns. We trained and evaluated Hi-C-LSTM on a subset of chromosomes (15-22) using pseudo-bulk scHi-C data from Ramani et al. [82] (see Methods for details). A representative heatmap from chromosome 21 shows that

Hi-C-LSTM is able to reconstruct contacts faithfully (Supplementary: Fig. 12), however, the sparsity of scHi-C data might be a potential concern when using the representations from scHi-C models for other downstream tasks like classification and in-silico manipulation.

- (35) The following text has been added to section Methods (datasets) of the revised manuscript, regarding processing of single-cell Hi-C data.

Pseudo-bulk single-cell Hi-C (scHi-C) data was downloaded using the GEO accession number GSM2254215 (see Supplementary: Table 3). We used the validPairs file to filter the data based on chromosome, positions and barcodes. We then obtained the number of reads aligned to hg19 corresponding to each distinct pair of bar codes from the percentages file.

- (36) The following text has been added to section Related Work of the revised manuscript, regarding background on imputation methods and the distinction between imputation and reconstruction.

Another related task is that of imputing unseen Hi-C data sets, for which several methods have been developed. Such imputation methods include SNIPER [11], DeepHiC [22], HiCPlus [23], Higashi [24] and scHiCluster [25]. SNIPER imputes high-coverage Hi-C using moderate-coverage Hi-C at the input. DeepHiC predicts high-resolution Hi-C contact maps from low-coverage sequencing data using generative adversarial networks. HiCPlus infers high-resolution Hi-C matrices from low-resolution Hi-C data using deep convolutional neural networks. Both DeepHiC and HiCPlus, do not form position specific representations that accomplish various downstream tasks, therefore, are not comparable to our method. Higashi enhances scHi-C data quality using hypergraph representation learning. scHiCluster studies cell type-specific chromosome structural patterns in scHi-C. These methods (apart from SNIPER) cannot be used for the task of bulk Hi-C representation learning because they do not form position-specific representations and, in the case of Higashi and scHiCluster, require single-cell data.

Note that, while existing methods for Hi-C representation learning (including SCI, SNIPER and Hi-C-LSTM) utilize a reconstruction loss that aims to reconstruct the input Hi-C data, they cannot in general be used for imputation.

- (37) The following text has been added to section Discussion of the revised manuscript, talking about the motivation behind using reconstruction.

Hi-C-LSTM, in its current form, is not designed to handle data from multiple cell types. We acknowledge that imputation is an important goal that deserves consideration, however, our goal with Hi-C-LSTM is not to impute data in new cell types but rather form cell type and position specific representations. Learning representations that help you reconstruct the Hi-C map can be useful for multiple reasons, namely, (a) the resulting model becomes a contact generation framework, (b) the resulting representations capture conformation defining elements, (c) the representations coupled with the model can be used for in-silico manipulation of genomic elements, and (d) the process can give us insights about which genomic sites are most important in construction. It is also important to note that we learn the representations in the process of reconstructing the matrix, i.e., reconstruction does not bring biological insight but is part of the process that forms the representations. The Hi-C reproduction evaluation shows how well these representations capture the information in the Hi-C matrix.

(38) The following text has been added to section Discussion of the revised manuscript, regarding the extension of the work to handle multiple cell types.

Extending this framework to work with multiple cell types at the same time may be possible with the addition of “cell type id” as an input parameter.

(39) The following text has been added to section Results (Hi-C reproduction) of the revised manuscript, talking about the interactions the model is likely to capture.

Understanding what kind of interactions the model is more likely to capture is vital. TADs are identified with a higher accuracy in all cell types compared to other larger chromatin structures like subcompartments (Fig. 4a,b; Supplementary: Fig. 7,8,9). On the other hand, Promoter-Enhancer interactions are hard to classify in all cell types (Supplementary: Fig. 7,8,9). This means that Hi-C-LSTM representations achieve higher accuracy for medium-scale structures such as TADs than for small-scale structures like promoter-enhancer interactions. This could be due to many factors including data resolution, model architecture, and conservation across cell types.

(40) The following text has been added to section Methods (sequence attribution) of the revised manuscript, talking about the choice of visualization browser.

We visualised our feature importance in the UCSC Genome Browser along with other genomic elements and signals. The 3D genome browser [105] is also an useful tool for visualization for contact map data.

b) It would be of great value if the authors could provide statistical insight about the efficacy of the method recapitulating the 3D structures, breaking it down by length into TADs, subTAD, and smaller loops.

We thank the reviewer for suggesting this. We have now broken down Hi-C-LSTMs classification performance into TADs, subTADs, loop domains, and subcompartments (Changes (26), (27), and (28)). For additional statistical insight we have added Area under the Receiver Operating Characteristic Curve (AuROC), Accuracy, and F-score along with the Area under the Precision Recall curve (AuPR or mAP) (Changes (31) and (32)).

2) This reviewer is under the impression that the authors still do not show convincingly that the model learns anything besides the known presence of the CTCF-motifs in convergent orientation and cohesin complex within the anchor region.

a) In the section “Feature attribution reveals association with genomic elements driving 3D conformation”, the authors show in Fig. 3 that the variation of the aggregated feature importance across relevant genomic regions can distinguish boundaries of domains and genomic regulatory elements. The objective of the method is to provide information about the 1D sequence feature associated with the 3D conformation. Therefore, it is relevant to complete the analysis by adding further motif enrichment analysis in the sequences with high aggregated feature importance. That can provide insight into other TF’s binding sites besides CTCF related to 3D loop formation.

We thank the reviewer for suggesting this. We have now conducted motif enrichment analysis using the aggregated integrated gradients (IG) feature importance scores and have ranked the Transcription Factor Binding Sites (TFBS) according to their importance scores (Changes (41),

(42), and (43)). The ranked list of TFs according to their importance scores is also provided as an additional supplementary file.

Changes made to the paper:

(41) Figure 16 (Fig. 6) has been added to section Results (feature attribution) of the revised manuscript, showing feature attribution results.

(42) The following text has been added to section Results (feature attribution) of the revised manuscript, talking about the motif enrichment analysis.

We analysed importance scores at TF binding sites (TFBS) [48] and saw that some TFBS have a larger positive importance score compared to others (Fig. 6b). Our motif enrichment analysis showed that the top 5 TFs according to importance score were: CTCF, ZNF143, FOXG1, SOX2, and XBP1 (Fig. 6b). As Cohesin is a known partner of CTCF, we looked for Cohesin binding sites in the ranked list and found them in the top 15. The full ranked list of transcription factors is attached as a Supplementary file. All TFs in the top 5 are known to play a role in chromatin conformation.

(43) The following text has been added to section Results (feature attribution) of the revised manuscript, talking about the relevance of TFs found in the top 5 of the ranked list.

Apart from CTCF, the other TFs in the top 5 are also known to play a role in conformation (Fig. 6b). There is a widespread role of C2H2-ZF proteins in chromatin structure and organization [52]. These TFs are known to promote local chromatin loosening, local chromatin condensation [53], and control chromatin accessibility through the recruitment of chromatin-modifying enzymes [52]. We found that ZNF143 binds directly to promoters and contributes to chromatin interactions connecting promoters to distal regulatory enhancers [54]. It has also been shown that ZNF143 plays a partner role in establishing conserved chromatin loops [54]. We also found that chromatin conformation changes considerably near the FOXG1 locus [55]. Moreover, many FOX TFs are considered pioneer factors which open closed chromatin and facilitate the binding of other TFs [56]. As per the last two TFs in the top 5, we found that SOX2 loss is seen to decrease chromatin interactivity genome-wide [57], and that the genomic distribution of XBP1 peaks shows that it binds promoters and potential enhancers [58, 59].

3) *This reviewer considers that the in-silico knockout experiment is incomplete. As noted in above comment 2, the method's novelty relies on providing insight about 3D conformation beyond the well-known CTCF anchors. In the current analysis, the authors confirm that CTCF and cohesin (RAD21 and SMC3) are critical drivers for reducing 3D contact strength. This type of analysis looks encouraging, therefore would be interesting if the authors can extend it by:*

a) Testing the in-silico knockout of CTCF and cohesin out of regions of IG importance peaks and within no loop regions to demonstrate the specificity of CTCF binding that the model picks up.

We thank the reviewer for pointing this out. We initially assumed that knocking out both sides of the loop anchors would be representative of CTCF knockout and knocking out only one side of the loop anchor would be sufficient to be representative of the Cohesin ring breaking. After more careful analysis, we have noticed that its not easy to separate the effect of CTCF and Cohesin at

Figure 16: Hi-C-LSTM representations identify genomic elements involved in conformation through Integrated Gradients (IG) feature importance analysis. a) The IG feature importance averaged across different TADs of varying sizes. The vertical axis indicates the average IG importance at each position and the horizontal axis refers to relative distance between positions in Kbp, upstream/downstream of the TADs. **b)** Violin plots of aggregated feature attribution scores for top ranked transcription factor binding sites (TFBS). The x-axis shows the labels/elements and the y-axis displays the z normalized feature importance scores from Integrated Gradients. Both at loop and non-loop regions, the scores shown are aggregated only at shared sites. **c)** The IG feature importance for a selected genomic locus (chr21 28-29.2Mbp) along with genes, regulatory elements, GC percentage, CTCF signal, and conserved TFBS among others in the UCSC genome browser. We see that the feature importance scores peak at known regulatory elements, higher GC percentage, and CTCF peaks. **d)** Violin plots of aggregated feature attribution scores for selected elements. The x-axis shows the labels/elements and the y-axis displays the z normalized feature importance scores from Integrated Gradients. The scores for CTCF and Cohesin subunits are aggregated genome wide.

loop anchors. Therefore, we have now combined the effect of knockout at loop anchors into one “CTCF+Coheisin” knockout.

We have also completed this analysis by conducting a knockout of CTCF+Coheisin binding sites in non-loop regions where the IG importance scores are high (Change (41)) (we only considered

the case where CTCF and cohesin share sites, and ignored the cases where CTCF binds alone, and cohesin binds alone) (Change (44)). We compared the average difference in predicted contacts between CTCF+Cohesin knockout and no knockout in non-loop regions and loop anchors and found noticeable difference between the two (Changes (45) and (46)).

At TAD boundaries, we contrasted knockout with and without CTCF presence (Change (47)).

Additionally, our analysis revealed that performing knockout has peculiarities and there is more than one right way to do the knockout. We have now included discussion on why knockout is challenging (Change (48)) and on the different ways to do knockout (Changes (49), (50), (51), (52), and (53)).

Changes made to the paper:

- (44) The following text has been added to section Results (feature attribution) of the revised manuscript, talking about the difference in feature importance scores between CTCF and cohesin in loop and non-loop regions.

We found that CTCF+Cohesin sites at loop anchors show 10% higher mean importance score than CTCF+Cohesin sites at non-loop regions (we only considered the case where CTCF and Cohesin share sites) and in both cases they have a spread that is predominantly positive (Fig. 6b). Note that CTCF and Cohesin sites usually overlap, so we analyze them together. Specifically, 98% of loop anchor CTCF ChIP-seq peaks also harbor Cohesin peaks; 92% non-loop CTCF peaks do so [49, 50].

- (45) Figure 17 (Fig. 8) has been added to section Results (in-silico knockout) of the revised manuscript, showing knockout results.

- (46) The following text has been added to section Results (in-silico knockout) of the revised manuscript, talking about the difference in knockout results between CTCF and cohesin in loop and non-loop regions.

After the combined CTCF and Cohesin knockouts, the average contact strength reduces by 7% in a 200 Kbp window when compared to the no knockout case (Fig. 8c). CTCF knockout is seen to affect insulation and reflect possible loss of loops at 200 Kbp (Fig. 8c). The knockout of CTCF and Cohesin subunit binding sites at non-loop regions [49, 50] (just like feature attribution, we only considered the case where CTCF and Cohesin share sites, and ignored the cases where CTCF binds alone, and Cohesin binds alone) produces markedly different effects with 2% lower average inferred strength after knockout at 200Kbp, hinting at the relative importance of loop and non-loop binding factors 8c).

- (47) The following text has been added to section Results (in-silico knockout) of the revised manuscript, talking about knockout of TAD boundaries with and without CTCF.

TADs anchored with CTCF at their boundaries have a differential role to play in conformation compared to the TADs without CTCF. We wanted to check if Hi-C-LSTM can capture this differential behaviour of TADs by knocking out their boundaries. To deal with TADs of varying sizes, we partition the interior of all TADs into 10 equi-spaced bins and average the predicted contacts within these bins. We show these along with the regions outside the TAD boundary 100Kbp upstream and downstream, averaged across all TADs (Fig. 8d). The average difference in inferred Hi-C between the knockout at TAD boundaries and the no

Figure 17: In-silico deletion of transcription factor binding sites (TFBS), orientation replacement of CTCF binding sites and TAD boundaries with and without CTCF is performed and changes in the resulting Hi-C contact matrix is observed. **a)** The average difference in predicted Hi-C contact strength between CTCF+CoheSIN knockout and no knockout in a 2Mb window. We simulate deletion by shifting the downstream representations upward. **b)** Average difference in contact strength of the inferred Hi-C matrix between knockout and no knockout (y-axis) for varying distance between positions in Mbp (x-axis). The knockout experiments include TFBS knockout and convergent/divergent CTCF replacements (legend). **c)** The genome-wide average difference in predicted Hi-C contact strength between TAD boundaries knockout and no knockout with CTCF (upper-triangle) and without CTCF (lower-triangle).

knockout (Fig. 8d) shows largely decreased contacts for both TADs with and without CTCF in a 200Kbp window (3 % lower on average). Within the TAD, however, we see increased contacts for TADs without CTCF (5% higher on average) and decreased contacts with CTCF

(4% lower on average) (Fig. 8d).

- (48) The following text has been added to section Discussion of the revised manuscript, talking about the challenging nature of knockout experiments.

We noticed that performing knockout of specific single genomic sites is not as straightforward as performing insertion of a larger genomic segment (as seen in Duplication) for Hi-C-LSTM. This is primarily because of two reasons. First, because the row representation is fed throughout the column sequence, Hi-C-LSTM decoder is more dependent on the row representations than the column representations. Therefore, Hi-C-LSTM is less susceptible to manipulation of column representations alone, which is the case when inferring contacts for rows around the knockout site, and more reliable for the row pertaining to the knockout site. This issue of robustness to manipulation of column representations is less prominent during insertion because a contiguous segment gets inserted and in the post-insertion genome both the row and the column representations are affected. Second, single locus knockout is harder than knockout of a larger genomic segment because the sequential model is robust to slight perturbations in the input.

- (49) The following text has been added to section Results (in-silico knockout) of the revised manuscript, talking about results from different ways to simulate knockout.

As Hi-C-LSTM is able to perform in-silico insertion/duplication (see Duplication), we wanted to investigate whether knockout or deletion of certain genomic loci would produce reliable changes in the resulting Hi-C contact map.

It is an open question how to simulate small-scale perturbations. We performed knockout using four different techniques at CTCF plus Cohesin binding sites (see Discussion). The difference in inferred Hi-C between the CTCF plus Cohesin knockout and the no knockout using shifted representations (see Methods) shows the decrease in contact strength (7% lower on average) in the immediate neighborhood of the KO site (Fig. 8a). Other ways to simulate knockout like using the padding, zero and average representations (Supplementary: Fig. 16) exploit different properties of the model. We believe there is no one right way to perform knockout, however, we prefer the method of shifting all downstream representations from the knockout site upward (see Methods).

- (50) The following text has been added to section Discussion of the revised manuscript, talking about the different ways to perform knockout.

Moreover, there is no accepted standard way of simulating in-silico knockout in the Hi-C community in the context of manipulating sequential representations. There are four ways one can simulate the knockout of a locus. One, by replacing the representation by the zero representation. Two, by replacing the representation by the average representation in the neighborhood. Three, by replacing the representation by the representation of the padding input, and four, by shifting all downstream representations upward (see Methods). We tried all four techniques and found shifting the representations to be most convincing.

- (51) Figure 18 (Supplementary: Fig. 16) has been added to the supplementary, showing results from other ways to simulate knockout.

Figure 18: The average difference in predicted Hi-C contact strength between CTCF+Coheisin knockout and no knockout in a 2Mb window when simulating knockout using padding, zero, and average representations. **a)** Padding representation: replace the representation at the knockout site with the representation obtained from the embedding layer for the input indice zero (zero is considered as padding). **b)** Zero representation (upper-triangle): replace the representation at the knockout site with the zero representations, and average representation (lower-triangle): replace the representation at the knockout site with the average of the representations 100Kbp upstream and downstream of the knockout site.

(52) Figure 19 (Supplementary: Fig. 13) has been added to the supplementary, showing the different ways to perform knockout.

(53) The following text has been added to section Methods (in-silico perturbation) of the revised manuscript, talking about the different ways to perform knockout.

In a knockout experiment, we chose certain genomic sites (such as CTCF and Coheisin binding sites) and replaced their representations with a different representation depending on the method used to perform the knockout (Supplementary: Fig. 13).

Among the four possible methods to perform knockout, we prefer the method of shifting the representations. Shifting the representations not only captures the true post-duplication genome but also avoids the noise that comes from zeroing or averaging the representations in the neighbourhood (Supplementary: Fig. 13). It also is more interpretable than using the padding representation because we do not fully understand the role of padding representations in recreating the Hi-C matrix.

b) Testing the in-silico knockout of other TF bindings within IG importance peaks with no CTCF to evaluate potential new elements related to 3D contortion.

We thank the reviewer for suggesting this. Using the ranked TF table mentioned in Q2 (attached as an additional supplementary file) as a guide, we knocked-out the next 4 TF binding sites

Figure 19: Different ways to perform knockout. (1) Zero representations: replace the representation at the knockout site with the zero representations. 2) Average representation: replace the representation at the knockout site with the average of the representations 100Kbp upstream and downstream of the knockout site. 3) Shift representations: remove the representation at the knockout site and shift all downstream representations upward. 4) Padding representation: replace the representation at the knockout site with the representation obtained from the embedding layer for the input indice zero (zero is considered as padding).

(TFBS) following CTCF (Change (45)). We compared the average difference in predicted contacts between knockout and no knockout (Change (54)).

Changes made to the paper:

(54) The following text has been added to section Results (in-silico knockout) of the revised manuscript, talking about the TFBS knockout.

Along with CTCF, we knocked out the other 4 TF binding sites (TFBS) in the top 5 TFs according to the ranked list, namely, ZNF143, FOXG1, SOX2, and XBP1 (Fig. 8c). We see that the average predicted contacts after genome-wide knockout partially reflects the importance attributed to each TF by integrated gradients. FOXG1 binding site knockout reduces contacts by 7% on average, XBP1 binding site knockout reduces contacts by 4% on average, whereas ZNF143 and SOX2 binding site knockouts reduce contacts between 4-5% on average at 200Kbp. Most knockouts cause an increase in contacts at 300Kbp and a gradual increase in contacts after 400Kbp. These results validate that Hi-C-LSTM knockout of TFBS captures the general idea of contacts depleting within the domain and connecting regions outside the domain.

4) For validating the prediction accuracy using known structural variations that impact the 3D contact map, the authors used a 2.1 Mbp duplication at the *SOX9* locus in individuals with Cook’s syndrome. This reviewer notes the value of that positive validation, but that is a considerable duplication impacting a large fraction of the chromosome.

a) It will be interesting if the authors can validate their method using narrower rearrangements that disrupt the 3D conformation. This reviewer suggests using the two deletions at anchors of loops that contain the oncogenes *TAL1* and *LMO2* in T-ALL patients [Hnisz, D. et al. *Science*. 2016]. That paper showed that the deletions of the anchors increased interactions between enhancers and promoters that were isolated in the wild-type cells (anchors presence). That result has been validated previously by another machine learning approach [Trieu, T. et al. *Genome Biology*. 2020] thus, the authors will reinforce the utility of their result if they can use their method to evaluate the impact of these two deletions on the 3D contact map of these regions.

We thank the reviewer for suggesting this. Hi-C-LSTM captures the changes after *TAL1* knock-out well, but the changes after *LMO2* knockout to a lesser degree [4]. We took our trained GM12878 model and retrained it on the 5C data from only the two fragments of *TAL1* and *LMO2* neighbourhoods in HEK293T [4]. We then performed knockout of *TAL1* and *LMO2* anchor sites and compared the difference in knockout and no knockout in the observed map versus the predicted map (Changes (55), (56), (57), and (58). The results also point towards the transfer learning ability of Hi-C-LSTM (Change (59)).

Changes made to the paper:

- (55) The following section has been added to the Results of the revised manuscript, elaborating on the anchor deletions experiments.

Simulating loop anchor deletions at the *TAL1* and *LMO2* loci Hi-C-LSTM predicts measured 5C data

To further validate the ability of Hi-C-LSTM to predict experimental perturbations, we simulated the deletion of loop anchor regions at the *TAL1* and *LMO2* neighborhood boundaries in human embryonic kidney cells (HEK-293T) previously conducted by Hnisz et al. [74]. These deletions were observed in T-cell acute lymphoblastic leukemia (T-ALL) patients. The *TAL1* anchor deletion was seen on chromosome 1 in the neighbourhood of 47.7 Mbp (GRCh37/hg19, Fig. 9a), and the *LMO2* anchor deletion was seen on chromosome 11 in the neighbourhood of 34 Mbp (GRCh37/hg19, Fig. 9b) [74]. Both deletions included loop boundary sites. The authors hypothesized that deletions of loop boundary sites at these loci could cause activation of inactive proto-oncogenes within the loops [74]. To simulate a Hi-C experiment on a genome with these deletions, we first obtained the trained model from GM12878 and retrained it on the 5C data from the *TAL1* and *LMO2* segments [74]. We then made a new representation matrix that shifted the representations downstream from the knockout sites upward, and passed this representation matrix through the retrained Hi-C-LSTM decoder to produce a simulated Hi-C matrix (Supplementary: Fig. 14a,b, lower-triangle) (see Methods for more details) and compared this with the 5C experiment performed by Hnisz et al. [74] (Supplementary: Fig. 14a,b, upper-triangle).

They authors saw that the insulated neighborhoods of *TAL1* and *LMO2* were disrupted, which allowed activation of these elements by regulatory elements outside the loop, and

Figure 20: In-silico anchor deletions at the TAL1 and LMO2 loci [74]. **a, c)** TAL1 anchor deletion on chromosome 1. **a)** Observed Hi-C contacts before deletion (upper-triangle), and predicted Hi-C contacts before deletion (lower-triangle). **c)** Scatter plot of differences in contacts after and before TAL1 deletion. The x-axis shows observed differences, and the y-axis shows predicted differences. **b, d)** LMO2 anchor deletion on chromosome 11. **b)** Observed Hi-C contacts before deletion (upper-triangle), and predicted Hi-C contacts before deletion (lower-triangle). **d)** Scatter plot of differences in contacts after and before LMO2 deletion. The x-axis shows observed differences, and the y-axis shows predicted differences.

caused rearrangement of interactions around the neighborhood. We found that Hi-C-LSTM’s predicted contacts correlate with the post-deletion interactions hypothesized by Hnisz et al. To evaluate our predictions, we investigated whether there is a correlation in the differences of knockout and no knockout between the observed and the predicted contacts (Fig. 9c,d). We found a noticeable correlation between Hi-C-LSTM’s prediction differences between knockout and no knockout and the observed assayed contacts for TAL1 (Fig. 9c). The interactions across domain boundaries that did not exist pre-deletion in the TAL1 neighbourhoods were correctly captured by Hi-C-LSTM (Fig. 9c). The correlation for LMO2 was not as strong as TAL1 (Fig. 9d) and the discrepancy was particularly at points where the post knockout contacts were same as the pre-knockout or higher. We see that Hi-C-LSTM accurately predicts

decrease in post knockout contacts as decrease, but wrongly attributes some points of no-change and increase as decrease (Fig. 9d).

These anchor deletion experiments reaffirm that Hi-C-LSTM can perform in-silico alterations with moderate accuracy. Moreover, the results also point to the transfer learning ability of Hi-C-LSTM in cell types with limited data (see Discussion).

- (56) Figure. 21 (Supplementary: Fig. 14) has been added to the supplementary, showing post-deletion Hi-C.

Figure 21: Hi-C-LSTM simulated post-deletion Hi-C compared with observed post-deletion Hi-C. **a)** Observed Hi-C contacts after TAL1 deletion (upper-triangle), and predicted Hi-C contacts after TAL1 deletion (lower-triangle). **b)** Observed Hi-C contacts after LMO2 deletion (upper-triangle), and predicted Hi-C contacts after LMO2 deletion (lower-triangle).

- (57) The following text has been added to section Methods (in-silico perturbation) of the revised manuscript, talking about the anchor deletions experiments.

Our anchor deletion experiment was carried out by first obtaining the trained Hi-C-LSTM model from GM12878, and retraining it on the 5C data from the TAL1 and LMO2 segments in HEK-293T [74]. The TAL1 fragment is on chromosome 1 from 47.5-47.9 Mbp, and the LMO2 fragment is on chromosome 11 from 33.8-34.2 Mbp (GRCh37/hg19). After retraining the model with data from HEK-293T, we made a new representation matrix by shifting all the downstream representations upward (Supplementary: Fig. 13), and passed this representation matrix through the retrained Hi-C-LSTM decoder to produce the inferred Hi-C matrix (Fig. 14).

- (58) The following text has been added to section Methods (datasets) of the revised manuscript, pointing to data used for anchor deletion experiments.

For our anchor deletion experiment, we obtained the 5C data for the TAL1 and LMO2 fragments in chromosome 1 and 11 from Hnisz et al. [74].

- (59) The following text has been added to section Discussion of the revised manuscript, talking about Hi-C-LSTM’s transfer learning ability.

Transfer learning is an important goal for the Hi-C community because of the availability of a variety of disparate and sparse Hi-C datasets. Instead of training a new model from scratch for every new Hi-C, using existing models from other cell types can drastically speed up the training process and also deal with the sparsity of available data. We are able to perform Hi-C inference of fragments in a new cell type (HEK-293T) by using partial 5C fragments as input for retraining (Fig. 9a, b). Hi-C-LSTM accurately captures both the TAL1 and LMO2 5C observed fragments (Fig. 9a, b: upper-triangle) in its predicted Hi-C contacts (Fig. 9a, b: lower-triangle). This will allow the model to be rapidly used in cell types and tasks where the available contact data is scarce.

5) *Minor:*

- a) *In the analysis for Fig. 5a, it is unclear how many regions the authors use to compute the IG importance.*

We thank the reviewer for the question. We have now clarified this in the revised manuscript (Change (60)). Note that Figure 5 has become Figure 6 in the revised version.

Changes made to the paper:

- (60) The following text has been clarified in section Results (feature attribution) of the revised manuscript.

To deal with TADs of varying sizes, we partition the interior of all TADs into 10 equi-spaced bins and average the feature importance signal within these bins. We plot this signal along with the signal outside the TAD boundary 50Kbp upstream and downstream, averaged across all TADs (Fig. 6a)

b) *Fig. 5b should show the CTCF binding track.*

We thank the reviewer for the comment. Figure 5 has now become Figure 6 in the revised version. We have now added the CTCF ChIP-seq Uniform peaks to Fig. 6c (Change (41)). Along with the CTCF track we have now included other interesting data from the UCSC genome browser, such as GC percent, gene interactions, H3K27AC signal, DNA Methylation signal, and conserved Transcription Factor Binding Sites (TFBS) (Change (61)).

Changes made to the paper:

- (61) The following text has been added to section Results (feature attribution) of the revised manuscript, talking about CTCF peaks.

The feature importance peaks also correlate with CTCF peaks and GC percentage peaks (Fig. 6c)

c) *Fig. 5c should show the p-value of the appropriate comparison for each group.*

We thank the reviewer for the comment. We have now added the p-values of relevant comparisons

Table 4: P-values from the T-tests of feature attribution score samples for various elements. Feature scores from each element were tested with scores obtained by sampling randomly from the other elements. **a)** P-values for 3 categories of inactive elements from segway, namely, Repressed, Dead, and Low Regions. **b)** P-values for 6 categories of active elements, namely, Gene Body, TF, Enhancers, TSS, FAIRE, and FIREs. **c)** P-values for pairs of elements. Feature scores from one group were tested with scores from the other group. **d)** P-values for 4 transcription factors (TFs), namely, ZNF143, FOXG1, SOX2, and XBP1. Feature scores from each TF were tested with scores obtained by sampling randomly from the other TFs.

(a)		(b)	
Inactive Elements	P-value	Active Elements	P-value
Repressed	3.45e-9	Gene Body	7.04e-5
Dead	2.79e-7	TF	6.28e-6
Low	1.93e-8	Enhancer	2.18e-4
		TSS	8.61e-4
		FAIRE	6.27e-4
		FIREs	8.17e-5

Element 1	Element 2	P-value
CTCF (weak)	CTCF (strong)	0.08
TAD Boundaries (CTCF+)	TAD Boundaries (CTCF-)	4.62e-4
Loop Domains	Non-loop Domains	8.27e-5
CTCF+Cohesin (loop)	CTCF+Cohesin (Non-loop)	1.26e-5

Transcription Factors	P-value
ZNF143	0.0026
FOXG1	0.0071
SOX2	6.48e-4
XBP1	0.0063

for each group in the supplementary (Changes (62) and (63)).

Changes made to the paper:

- (62) The following text has been added to section Results (feature attribution) of the revised manuscript, referring to the p-value table.
P-values from the relevant comparisons for each group can be referred to in the Supplementary: Table 2.
- (63) Table 4 (Supplementary: Table 2) has been added to the supplementary, showing the p-values of feature attributions from the relevant comparisons.

d) In the analysis for Fig. 6, it is unclear how many regions the authors use for the in-silico knockout test.

We thank the reviewer for the question. In the old version, Fig. 6a,b were showing specific regions on chromosome 21. These two figures are no longer included. Figure 6 has now become Figure 8 in the revised version. In Fig. 8c, we average the results across all transcription factor binding sites (TFBS) (Change (64)). In Fig. 8d, we average across all TADs from the TADKB repository (Change (65)).

Changes made to the paper:

- (64) The following text has been clarified in section Results (in-silico knockout) of the revised manuscript, talking about the knockout of transcription factor binding sites.

We see that the average predicted contacts after genome-wide knockout partially reflects ...

- (65) The following text has been clarified in section Results (in-silico knockout) of the revised manuscript, talking about the knockout of TAD boundaries.

To deal with TADs of varying sizes, we partition the interior of all TADs into 10 equi-spaced bins and average the predicted contacts within these bins. We show these along with the regions outside the TAD boundary 100Kbp upstream and downstream, averaged across all TADs (Fig. 8d).

4 Responses to the Comments from Reviewer 3

The authors use embedding neural network layers to create 1D representation of genomes from their 3D Hi-C heatmaps. The 1D representation can then be converted to 3D interacting frequencies by a long short-term memory (LSTM) neural network model. This method is called Hi-C-LSTM. To compare their method (Hi-C-LSTM) with SNIPER and SCI (which also predict the 1D representation of genomes), the authors divide the trained model into two parts. The first part is 1D representation, which is actually the embedding neural network layer with trained parameters. The second part is a decoder, which is actually the LSTM with trained parameters. The authors find that not only does the 1D representation of Hi-C-LSTM capture more genome information than those of SNIPER and SCI, but also the LSTM as a decoder, predicts the 3D interacting frequencies more precisely than alternative decoders such as a convolutional neural network (CNN) or a fully connected (FC) feed-forward neural network. To further show that the 1D representation of Hi-C-LSTM is more informative than those of SNIPER and SCI, the authors train a boosted decision tree classifier called XGBoost by 1D representations versus gene expression, enhancers, TSSs, and so on. The 1D representation gives better classifications on all genomic phenomena and regions except subcompartments. The authors claim that this is due to the intra-chromosome design of Hi-C-LSTM. There are no experimental confirmations for all claims. By feature attribution analysis, the authors find that genomic elements related to 3D conformation, such as CTCF, cohesin, TAD boundaries, have higher IG scores based on the 1D representation of genomes predicted by Hi-C-LSTM. Finally, by doing in-silico knockout experiments on the 1D representation of genomes, Hi-C-LSTM allows one to predict the resulting 3D conformation without wet experiments. Knockout of CTCF or cohesin binding sites and CTCF orientation replacement give predicted interacting variances consistent loop extrusion model. Moreover, Hi-C-LSTM “accurately” predicts the effects of a 2.1 Mbp duplication at the SOX9 locus revealed in previous wet assays. Here are some points to consider.

1) *In lines 243-290, the authors do in-silico knockout experiments on both CTCF and cohesin binding sites based on the data set in lines 393-396. However, the cohesin does not have known binding motifs. In fact, the binding sites of cohesin highly overlap with those of CTCF because CTCF blocks cohesin. The authors should pay more attention to it.*

We thank the reviewer for pointing this out. We initially assumed that knocking out both sides of the loop anchors would be representative of CTCF knockout and knocking out only one side of the loop anchor would be sufficient to be representative of the Cohesin ring breaking. After more careful analysis, we have noticed that its not easy to separate the effect of CTCF and Cohesin at loop anchors. Therefore, we have now combined the effect of knockout at loop anchors into one “CTCF+Cohesin” knockout (Changes (66), (67), (68), and (69)).

We have also completed this analysis by conducting a knockout of CTCF+Cohesin binding sites in non-loop regions where the IG importance scores are high (we only considered the case where CTCF and cohesin share sites, and ignored the cases where CTCF binds alone, and cohesin binds alone) (Changes (66) and (67)). We compared the average difference in predicted contacts between CTCF+Cohesin binding site knockout and no knockout in non loop regions and loop anchors and found noticeable difference between the two (Changes (68) and (69)).

Additionally, our analysis revealed that performing knockout has peculiarities and there is more than one right way to do the knockout. We have now included discussion on why knockout is challenging (Change (70)) and on the different ways to do knockout (Changes (71), (72), (73), (74), and (75)).

Changes made to the paper:

- (66) Figure 22 (Fig. 6) has been added to section Results (feature attribution) of the revised manuscript, showing feature attribution results.
- (67) The following text has been added to section Results (feature attribution) of the revised manuscript, talking about the difference in feature importance scores between CTCF and cohesin in loop and non-loop regions.
We found that CTCF+Cohesin sites at loop anchors show 10% higher mean importance score than CTCF+Cohesin sites at non-loop regions (we only considered the case where CTCF and Cohesin share sites) and in both cases they have a spread that is predominantly positive (Fig. 6b). Note that CTCF and Cohesin sites usually overlap, so we analyze them together. Specifically, 98% of loop anchor CTCF ChIP-seq peaks also harbor Cohesin peaks; 92% non-loop CTCF peaks do so [49, 50].
- (68) Figure 23 (Fig. 8) has been added to section Results (in-silico knockout) of the revised manuscript, showing knockout results.
- (69) The following text has been added to section Results (in-silico knockout) of the revised manuscript, talking about the difference in knockout results between CTCF and cohesin in loop and non-loop regions.
After the combined CTCF and Cohesin knockouts, the average contact strength reduces by 7% in a 200 Kbp window when compared to the no knockout case (Fig. 8c). CTCF knockout is seen to affect insulation and reflect possible loss of loops at 200 Kbp (Fig. 8c). The knockout of CTCF and Cohesin subunit binding sites at non-loop regions [49, 50] (just like feature attribution, we only considered the case where CTCF and Cohesin share sites, and ignored the cases where CTCF binds alone, and Cohesin binds alone) produces markedly different effects with 2% lower average inferred strength after knockout at 200Kbp, hinting at the relative importance of loop and non-loop binding factors 8c).
- (70) The following text has been added to section Discussion of the revised manuscript, talking about the challenging nature of knockout experiments.
We noticed that performing knockout of specific single genomic sites is not as straightforward as performing insertion of a larger genomic segment (as seen in Duplication) for Hi-C-LSTM. This is primarily because of two reasons. First, because the row representation is fed throughout the column sequence, Hi-C-LSTM decoder is more dependent on the row representations than the column representations. Therefore, Hi-C-LSTM is less susceptible to manipulation of column representations alone, which is the case when inferring contacts for rows around the knockout site, and more reliable for the row pertaining to the knockout site. This issue of robustness to manipulation of column representations is less prominent during insertion because a contiguous segment gets inserted and in the post-insertion genome both the row and the column representations are affected. Second, single locus knockout is harder than knockout of a larger genomic segment because the sequential model is robust to slight perturbations in the input.

Figure 22: Hi-C-LSTM representations identify genomic elements involved in conformation through Integrated Gradients (IG) feature importance analysis. **a)** The IG feature importance averaged across different TADs of varying sizes. The vertical axis indicates the average IG importance at each position and the horizontal axis refers to relative distance between positions in Kbp, upstream/downstream of the TADs. **b)** Violin plots of aggregated feature attribution scores for top ranked transcription factor binding sites (TFBS). The x-axis shows the labels/elements and the y-axis displays the z normalized feature importance scores from Integrated Gradients. Both at loop and non-loop regions, the scores shown are aggregated only at shared sites. **c)** The IG feature importance for a selected genomic locus (chr21 28-29.2Mbp) along with genes, regulatory elements, GC percentage, CTCF signal, and conserved TFBS among others in the UCSC genome browser. We see that the feature importance scores peak at known regulatory elements, higher GC percentage, and CTCF peaks. **d)** Violin plots of aggregated feature attribution scores for selected elements. The x-axis shows the labels/elements and the y-axis displays the z normalized feature importance scores from Integrated Gradients. The scores for CTCF and Cohezin subunits are aggregated genome wide.

(71) The following text has been added to section Results (in-silico knockout) of the revised manuscript, talking about results from different ways to simulate knockout. As Hi-C-LSTM is able to perform in-silico insertion/duplication (see Duplication), we wanted to investigate whether knockout or deletion of certain genomic loci would produce reliable

Figure 23: In-silico deletion of transcription factor binding sites (TFBS), orientation replacement of CTCF binding sites and TAD boundaries with and without CTCF is performed and changes in the resulting Hi-C contact matrix is observed. **a)** The average difference in predicted Hi-C contact strength between CTCF+Cohe sin knockout and no knockout in a 2Mb window. We simulate deletion by shifting the downstream representations upward. **b)** Average difference in contact strength of the inferred Hi-C matrix between knockout and no knockout (y-axis) for varying distance between positions in Mbp (x-axis). The knockout experiments include TFBS knockout and convergent/divergent CTCF replacements (legend). **c)** The genome-wide average difference in predicted Hi-C contact strength between TAD boundaries knockout and no knockout with CTCF (upper-triangle) and without CTCF (lower-triangle).

changes in the resulting Hi-C contact map.

It is an open question how to simulate small-scale perturbations. We performed knockout using four different techniques at CTCF plus Cohe sin binding sites (see Discussion). The

difference in inferred Hi-C between the CTCF plus Cohesin knockout and the no knockout using shifted representations (see Methods) shows the decrease in contact strength (7% lower on average) in the immediate neighborhood of the KO site (Fig. 8a). Other ways to simulate knockout like using the padding, zero and average representations (Supplementary: Fig. 16) exploit different properties of the model. We believe there is no one right way to perform knockout, however, we prefer the method of shifting all downstream representations from the knockout site upward (see Methods).

(72) The following text has been added to section Discussion of the revised manuscript, talking about the different ways to perform knockout.

Moreover, there is no accepted standard way of simulating in-silico knockout in the Hi-C community in the context of manipulating sequential representations. There are four ways one can simulate the knockout of a locus. One, by replacing the representation by the zero representation. Two, by replacing the representation by the average representation in the neighborhood. Three, by replacing the representation by the representation of the padding input, and four, by shifting all downstream representations upward (see Methods). We tried all four techniques and found shifting the representations to be most convincing.

(73) Figure 24 (Supplementary: Fig. 16) has been added to the supplementary, showing results from other ways to simulate knockout.

Figure 24: The average difference in predicted Hi-C contact strength between CTCF+Cohesin knockout and no knockout in a 2Mb window when simulating knockout using padding, zero, and average representations. **a)** Padding representation: replace the representation at the knockout site with the representation obtained from the embedding layer for the input indice zero (zero is considered as padding). **b)** Zero representation (upper-triangle): replace the representation at the knockout site with the zero representations, and average representation (lower-triangle): replace the representation at the knockout site with the average of the representations 100Kbp upstream and downstream of the knockout site.

(74) Figure 25 (Supplementary: Fig. 13) has been added to the supplementary, showing the different ways to perform knockout.

Figure 25: Different ways to perform knockout. (1) Zero representations: replace the representation at the knockout site with the zero representations. 2) Average representation: replace the representation at the knockout site with the average of the representations 100Kbp upstream and downstream of the knockout site. 3) Shift representations: remove the representation at the knockout site and shift all downstream representations upward. 4) Padding representation: replace the representation at the knockout site with the representation obtained from the embedding layer for the input indice zero (zero is considered as padding).

(75) The following text has been added to section Methods (in-silico perturbation) of the revised manuscript, talking about the different ways to perform knockout.

In a knockout experiment, we chose certain genomic sites (such as CTCF and Cohesin binding sites) and replaced their representations with a different representation depending on the method used to perform the knockout (Supplementary: Fig. 13).

Among the four possible methods to perform knockout, we prefer the method of shifting the representations. Shifting the representations not only captures the true post-duplication genome but also avoids the noise that comes from zeroing or averaging the representations in the neighbourhood (Supplementary: Fig. 13). It also is more interpretable than using the padding representation because we do not fully understand the role of padding representations in recreating the Hi-C matrix.

2) *The in-silico knockout experiments on CTCF and cohesin are not validated by wet assays. Many data of experimental knocking out CTCF binding sites are available; however none of current models can explain all experimental data. Maybe the authors can compare their in-silico knockout experiments against these existing data or generate more strictly controlled data.*

We thank the reviewer for their comment. To the best of our knowledge, there is no reliable post-CTCF binding site knockout Hi-C available in our cell types of interest in humans. We have now elaborated on the difference between removal of the CTCF protein itself and knockout of individual CTCF binding sites (Change (76)). This reiterates our previous claim in the discussion section that Hi-C-LSTM can only model *cis*-effects and not *trans*-effects.

Therefore, instead of looking for a post-CTCF binding site knockout Hi-C, we decided to verify our model using anchor deletion data from TAL1 and LMO2 neighbourhoods [4] (Changes (77), (78), (79), (80), and (81)). This further supports our earlier comparison and validation with post-duplication data from Melo et al. [5].

Changes made to the paper:

- (76) The following text has been added to section Discussion of the revised manuscript, elaborating on existing CTCF depletion experiments.

To directly validate the *cis*-knockout of CTCF binding sites, to the best of our knowledge, there is no reliable post-CTCF binding site knockout Hi-C available in our cell types of interest in humans. There are Hi-C experiments available after CTCF protein depletion [76-80]. There are also Cohesin depletion experiments [18]. However, these experiments cannot be used to compare with our binding site knockout experiments as depleting the protein itself is not the same as knocking out the binding site the protein can bind to. There is one work that performs post-CTCF binding site knockout Hi-C, but this experiment is conducted in mice [81]. Therefore, instead of looking for a post-CTCF binding site knockout Hi-C, we decided to further verify our model using data from duplications (see section Duplication) and anchor deletions (see section Anchor).

- (77) The following section has been added to the Results of the revised manuscript, elaborating on the anchor deletions experiments.

Simulating loop anchor deletions at the TAL1 and LMO2 loci Hi-C-LSTM predicts measured 5C data

To further validate the ability of Hi-C-LSTM to predict experimental perturbations, we simulated the deletion of loop anchor regions at the TAL1 and LMO2 neighborhood boundaries in human embryonic kidney cells (HEK-293T) previously conducted by Hnisz et al. [74]. These deletions were observed in T-cell acute lymphoblastic leukemia (T-ALL) patients. The TAL1 anchor deletion was seen on chromosome 1 in the neighbourhood of 47.7 Mbp (GRCh37/hg19, Fig. 9a), and the LMO2 anchor deletion was seen on chromosome 11 in the neighbourhood of 34 Mbp (GRCh37/hg19, Fig. 9b) [74]. Both deletions included loop boundary sites. The authors hypothesized that deletions of loop boundary sites at these loci could cause activation of inactive proto-oncogenes within the loops [74]. To simulate a Hi-C experiment on a genome with these deletions, we first obtained the trained model from GM12878 and re-trained it on the 5C data from the TAL1 and LMO2 segments [74]. We then made a new representation matrix that shifted the representations downstream from the knockout sites

Figure 26: In-silico anchor deletions at the TAL1 and LMO2 loci [74]. **a, c)** TAL1 anchor deletion on chromosome 1. **a)** Observed Hi-C contacts before deletion (upper-triangle), and predicted Hi-C contacts before deletion (lower-triangle). **c)** Scatter plot of differences in contacts after and before TAL1 deletion. The x-axis shows observed differences, and the y-axis shows predicted differences. **b, d)** LMO2 anchor deletion on chromosome 11. **b)** Observed Hi-C contacts before deletion (upper-triangle), and predicted Hi-C contacts before deletion (lower-triangle). **d)** Scatter plot of differences in contacts after and before LMO2 deletion. The x-axis shows observed differences, and the y-axis shows predicted differences.

upward, and passed this representation matrix through the retrained Hi-C-LSTM decoder to produce a simulated Hi-C matrix (Supplementary: Fig. 14a,b, lower-triangle) (see Methods for more details) and compared this with the 5C experiment performed by Hnisz et al. [74] (Supplementary: Fig. 14a,b, upper-triangle).

They authors saw that the insulated neighborhoods of TAL1 and LMO2 were disrupted, which allowed activation of these elements by regulatory elements outside the loop, and caused rearrangement of interactions around the neighborhood. We found that Hi-C-LSTM’s predicted contacts correlate with the post-deletion interactions hypothesized by Hnisz et al. To evaluate our predictions, we investigated whether there is a correlation in the differences of

knockout and no knockout between the observed and the predicted contacts (Fig. 9c,d). We found a noticeable correlation between Hi-C-LSTM’s prediction differences between knockout and no knockout and the observed assayed contacts for TAL1 (Fig. 9c). The interactions across domain boundaries that did not exist pre-deletion in the TAL1 neighbourhoods were correctly captured by Hi-C-LSTM (Fig. 9c). The correlation for LMO2 was not as strong as TAL1 (Fig. 9d) and the discrepancy was particularly at points where the post knockout contacts were same as the pre-knockout or higher. We see that Hi-C-LSTM accurately predicts decrease in post knockout contacts as decrease, but wrongly attributes some points of no-change and increase as decrease (Fig. 9d).

These anchor deletion experiments reaffirm that Hi-C-LSTM can perform in-silico alterations with moderate accuracy. Moreover, the results also point to the transfer learning ability of Hi-C-LSTM in cell types with limited data (see Discussion).

(78) Figure. 27 (Supplementary: Fig. 14) has been added to the supplementary, showing post-deletion Hi-C.

Figure 27: Hi-C-LSTM simulated post-deletion Hi-C compared with observed post-deletion Hi-C.

a) Observed Hi-C contacts after TAL1 deletion (upper-triangle), and predicted Hi-C contacts after TAL1 deletion (lower-triangle). **b)** Observed Hi-C contacts after LMO2 deletion (upper-triangle), and predicted Hi-C contacts after LMO2 deletion (lower-triangle).

(79) The following text has been added to section Methods (in-silico perturbation) of the revised manuscript, talking about the anchor deletions experiments.

Our anchor deletion experiment was carried out by first obtaining the trained Hi-C-LSTM model from GM12878, and retraining it on the 5C data from the TAL1 and LMO2 segments in HEK-293T [74]. The TAL1 fragment is on chromosome 1 from 47.5-47.9 Mbp, and the LMO2 fragment is on chromosome 11 from 33.8-34.2 Mbp (GRCh37/hg19). After retraining

the model with data from HEK-293T, we made a new representation matrix by shifting all the downstream representations upward (Supplementary: Fig. 13), and passed this representation matrix through the retrained Hi-C-LSTM decoder to produce the inferred Hi-C matrix (Fig. 14).

(80) The following text has been added to section Methods (datasets) of the revised manuscript, pointing to data used for anchor deletion experiments.

For our anchor deletion experiment, we obtained the 5C data for the TAL1 and LMO2 fragments in chromosome 1 and 11 from Hnisz et al. [74].

(81) The following text has been added to section Discussion of the revised manuscript, talking about Hi-C-LSTM’s transfer learning ability.

Transfer learning is an important goal for the Hi-C community because of the availability of a variety of disparate and sparse Hi-C datasets. Instead of training a new model from scratch for every new Hi-C, using existing models from other cell types can drastically speed up the training process and also deal with the sparsity of available data. We are able to perform Hi-C inference of fragments in a new cell type (HEK-293T) by using partial 5C fragments as input for retraining (Fig. 9a, b). Hi-C-LSTM accurately captures both the TAL1 and LMO2 5C observed fragments (Fig. 9a, b: upper-triangle) in its predicted Hi-C contacts (Fig. 9a, b: lower-triangle). This will allow the model to be rapidly used in cell types and tasks where the available contact data is scarce.

3) In lines 402-418, the authors illustrate long short-term memory (LSTM) networks. Then LSTM is used in Eq. 3 as a decoder to predict the interacting frequency from the 1D representations of two loci. The symbols used in lines 402-418 and Eq. 3 seem to be inconsistent. LSTM needs initial values for both the cell and memory (or hidden) states, and at each step, it calculates a new pair of the cell and memory states. In Eq. 3, R_1 and R_2 seem to be the current input, so where is the cell state? Also, which gate (the input, output, forget gate, or even none of them) gives the final interacting frequency? Eq. 3 does not clarify this.

We thank the reviewer for pointing this out. We have now fixed this in the revised manuscript. We had previously omitted the cell state from Eq. 3, which we have now added (Change (82)). The reviewer is also right in pointing out that one of the gates modulates the final interacting frequency (output gate). The modulated output by output gate is the result h_t at each time step. We had previously not mentioned the linear layer that takes as input h_t and outputs the interacting frequency. We have now included this in the revised manuscript (Change (83)).

Changes made to the paper:

(82) Equation 1 (Eq. 3) has been modified in section Methods (Hi-C-LSTM) of the revised manuscript.

$$\hat{H}_{i,j} = LSTM((R_i, R_j), h_j, c_j) \quad \text{for } j = 1, 2, \dots, N$$

$$\quad \quad \quad \text{for } i = 1, 2, \dots, N \quad (1)$$

where h_j and c_j are the same as h_{j-1} and c_{j-1} within the frame and are reinitialized at the beginning of each new frame.

(83) The following text has been added to section Methods (LSTM) of the revised manuscript, talking about the final output interacting frequency.

The current memory state (h_t) is fed into a linear layer with a sigmoid function at each step which produces the final output interacting frequency at that step.

4) *I think a simple illustration of the embedding neural network layer is also necessary.*

We thank the reviewer for the comment. We have now added an illustration of the embedding neural network layer and an explanation in the revised manuscript (Changes (84) and (85)).

Changes made to the paper:

(84) The following text has been added to section Methods (Hi-C-LSTM) in the revised manuscript, referring to the embedding neural network layer.

Hi-C-LSTM creates a representation given a pair of genomic positions in the Hi-C contact matrix using an embedding neural network layer (for an illustration see Supplementary: Fig. 15) and predicts the contact strength at that particular pair via a deep LSTM [35] that takes these representations as input (Fig. 2).

(85) Figure 28 (Supplementary: Fig. 15) has been added to the supplementary, showing the embedding neural network layer.

5) *Perhaps Eq. 4 is incorrect.*

We thank the reviewer for pointing this out. We have now fixed this in the revised manuscript (Change (86)).

Changes made to the paper:

(86) Equation 2 (Eq. 4) has been modified in section Methods (Hi-C-LSTM) of the revised manuscript.

$$MSE_i = \frac{1}{N} \left[\sum_{j=1}^N (H_{i,j} - \hat{H}_{i,j})^2 \right] \quad \text{for } i = 1, 2, \dots, N \quad (2)$$

6) *Hi-C-LSTM performs worse in classifying subcompartments than SNIPER and SCI. In my opinion, one possible reason is that the LSTM in this paper reinitializes the hidden states after every frame of length 150. Such a local design may be unsuitable for predicting long-range genomic phenomena such as the subcompartments.*

We thank the reviewer for pointing this out. We agree with the reviewer that having a shorter frame length is one of the reasons our model does not do well at identifying long-range interactions like subcompartments. We have now added discussion pertaining to this in the revised manuscript (Change (87)).

In our future work we plan to aggregate representations from models that work at various resolutions (2Kbp, 5Kbp, 10Kbp, 100Kbp, 1Mbp), which will allow us to detect long range interactions much better (Change (88)). We also plan to use models like Transformers that can work with longer sequences (Change (89)).

Figure 28: An illustration of the embedding neural network layer, where N is the chromosome length, and M is the representation size. The PyTorch Embedding layer we use is essentially just a Linear layer. We could alternatively define it as a linear layer where the number of inputs corresponds to the chromosome length and the number of outputs corresponds to the representation size. Here, each genomic index is represented as a one-hot vector, where the length of the vector is equal to the chromosome length, with a 1 in a unique position, compared to all other genomic indices. The PyTorch Embedding layer simplifies this by requiring just the position index instead of the big one-hot vector.

We conducted an experiment running Hi-C-LSTM at different resolutions, to provide us insight into the effect of resolution on element identification. We trained Hi-C-LSTM and evaluated its classification performance on 3 different resolutions apart from 10Kbp, namely, 2Kbp, 100Kbp, and 500Kbp (Change (90)).

Changes made to the paper:

(87) The following text has been added to section Discussion of the revised manuscript, talking about ways to work with longer sequences.

To reconstruct, we chose a shorter frame length of 150 because (1) LSTMs can typically work with sequences of lengths in the order of 100s but cannot handle very long sequences in the order of 1000s because of issues with gradient propagation. (2) A shorter frame length helps us fit our model in memory and speeds up training time drastically. (3) At our choice of 10Kbp resolution it allows us to identify other important large chromatin structures like loop domains, TADs, and subTADs. A shorter frame length is one of the reasons our model does not do well at identifying long-range interactions like subcompartments, however, our goal is not just to identify long-range interactions but design a model that can identify both short and long range interactions satisfactorily. Hi-C-LSTM is able to achieve this trade-off because of its shorter frame length. In future work, we plan to work with longer sequences efficiently by, (1) Creating hierarchical representations from initial representations. (2) Using models like Transformers that can handle longer sequences. (3) Aggregating representations learnt

at different Hi-C resolutions.

- (88) The following text has been added to section Discussion of the revised manuscript, talking about aggregating representations at different scales.
Third, combining representations from models trained at varying resolutions to form a common representation would allow us to not only discover new elements at different scales but also form a comprehensive scale agnostic representation.
- (89) The following text has been added to section Discussion of the revised manuscript, talking about using Transformers.
Fourth, the success of a LSTM model suggests trying other sequential neural network models that can handle longer sequences such as Transformers [83], coupled with learning hierarchical representations.
- (90) The following section has been added to the Results of the revised manuscript, showing Hi-C-LSTM predictions with different Hi-C resolutions.

Hi-C-LSTM recapitulates structures at different Hi-C resolutions

To check if Hi-C-LSTM works at different resolutions of Hi-C data, in addition to our model trained at 10Kbp, we trained Hi-C-LSTM at three other resolutions of 2Kbp, 100Kbp and 500Kbp. As expected, models at different scales detect different elements, classify differently, and attribute importance to different sites depending on the resolution. The models achieved these by forming representations that allowed them to construct the Hi-C map at the given resolution. We investigate how these representations differ from the ones learned at 10Kbp.

To train the model at 2Kbp, we used only a subset of chromosomes due to memory and compute constraints but trained on the whole genome at other resolutions. A selected portion in chromosome 21 (Fig. 5a) shows that the predicted Hi-C values capture the fine structure of Hi-C even at 2Kbp resolution. The sparsity of available data at 2Kbp is a major constraint in enhancing the performance of the model at this resolution. Hi-C-LSTM captures the Hi-C macro-structures accurately at 100Kbp (Fig. 5c) and 500Kbp (Fig. 5b). This is because operating at this resolution with our sequence length allows it to span entire smaller chromosomes.

We found that representations at different resolutions predict chromatin structures of different scales. The classification performance (as measured in mAP) with gene expression, enhancers, TADs, subTADs, and subcompartments of models trained at different resolutions (Fig. 5d), shows that for small scale phenomena and expression like gene expression and enhancers, the cumulative prediction score worsens with coarser resolution as expected. For enhancers, the prediction score drops by 0.22 units when the resolution goes from 2Kbp to 500Kbp (Fig. 5d). Both with TADs and subTADs, the model at 100Kbp has the best prediction score, closely followed by the model at 10Kbp (Fig. 5d). We attribute this performance to the fact that these resolutions, combined with our frame length of 150, are close to the to the averages sizes of both these elements. The model at 500 Kbp performs best at identifying subcompartments given that the average size of subcompartments is 300Kbp (Fig. 5d). These results point to the idea that aggregating representations learnt at different Hi-C resolutions will likely increase prediction performance across elements of all sizes. Such aggregation will also potentially help in alleviating computational bottlenecks, as a model at a particular

Figure 29: Hi-C-LSTM applied at different resolutions. **a**) Hi-C-LSTM predictions at 2Kbp resolution. A selected portion of the observed Hi-C map (upper-triangle) and the predicted Hi-C map (lower-triangle) in GM12878. The portion is selected from chromosome 21, between 43.2 Mbp to 48.1 Mbp. **b**) Hi-C-LSTM predictions at 500Kbp resolution. The observed Hi-C map (upper-triangle) and the predicted Hi-C map (lower-triangle) in GM12878 for chromosome 21. **c**) Hi-C-LSTM predictions at 100Kbp resolution. The observed Hi-C map (upper-triangle) and the predicted Hi-C map (lower-triangle) in GM12878 for chromosome 21. **d**) The classification performance (as measured in mAP) with gene expression, enhancers, TADs, subTADs, and subcompartments of models trained at different resolutions.

resolution need not take the broader context into account (see Discussion).

7) *This sentence has problems: “we show that our representation captures cell-type-specific functional activity, genomic elements and identifies genomic regions that drive conformation.”*

We thank the reviewer for pointing this out. We have now fixed this in the revised manuscript (Change (91)).

Changes made to the paper:

(91) The following text has been fixed in the revised manuscript.

We show that our representation captures cell-type-specific functional activity, genomic elements, and regions that drive conformation.

8) *Why Hi-C-LSTM cannot identify insulators?*

We thank the reviewer for pointing this out. We believe that insulators don't belong to a separate category of elements but rather elements like loops and TADs partake in insulation. Hi-C-LSTM can identify these elements that give rise to insulation between genomic regions.

We have now broken down Hi-C-LSTMs classification performance into TADs, subTADs, loop domains, and subcompartments. For additional statistical insight we have added Area under the Receiver Operating Characteristic Curve (AuROC), Accuracy, and F-score along with the Area under the Precision Recall curve (AuPR or mAP) (Changes (92), (93), and (94)).

Changes made to the paper:

(92) Figure 30 (Fig. 4) has been added to section Results (classification) of the revised manuscript, showing classification performance in different cell types and experiments with reduced read depth.

(93) The following text has been added to section Results (classification) of the revised manuscript, talking about the classification performance in different cell types and its relationship to reduced read depth.

Similar to reconstruction, when comparing classification performance across cell types, we saw that Hi-C-LSTM accuracy worsened with reduced read depth. However, the classification performance trend over tasks was preserved (Fig. 4c). We include results for all available data sets for each cell type. We omitted WTC11 from this analysis because most data sets are not available (see Methods for details regarding element specific data in cell types). We observed that the accuracy reduced by 0.6 units on average when the reads reduced to 150 million (Fig. 4c). Next, we compared the classification performance of Hi-C-LSTM with other methods (SCI, SNIPER) and baselines (PCA, SUBCOMPARTMENT-ID) in these cell types (Supplementary: Fig. 7,8,9). Similar to R2, we saw that the prediction score trend of methods is preserved across all these cell types.

(94) The following text has been added to section Methods (element identification evaluation) of the revised manuscript, talking about the classification metrics.

The classifier was evaluated using four standard metrics for classification tasks, namely, mean average precision (mAP) (otherwise known as area under the Precision-Recall curve

Figure 30: Genomic phenomena and chromatin regions are classified using the Hi-C-LSTM representations as input. **a)** Prediction accuracy for gene expression, replication timing, enhancers, transcription start sites (TSSs), promoter-enhancer interactions (PEIs), frequently interacting regions (FIREs), loop and non-loop domains, and subcompartments in GM12878. The y-axis shows the mean average precision (mAP), the x-ticks refer to the prediction targets, and the legend shows the different methods compared with. **b)** Same as **a)**, but for targets in H1-hESC. **c)** mAP using Hi-C-LSTM for targets compared across cell types. **d)** The Precision-Recall curves of Hi-C-LSTM for the various prediction targets in GM12878. The y-axis shows the Precision, the x-axis shows the Recall, and the legend shows the prediction targets.

(AuPR), area under the Receiver Operating Characteristic curve (AuROC), Accuracy ($A = \frac{TP+TN}{TP+FP+TN+FN}$), and F-score. AuROC is defined as the area under the curve that has true positive rate ($TPR = \frac{TP}{TP+FN}$) on the y-axis and false positive rate ($FPR = \frac{FP}{FP+TN}$) on the x-axis. mAP is defined as the average of the maximum precision ($P = \frac{TP}{TP+FP}$) scores

achieved at varying recall levels ($R = TPR$). F-score is defined based on precision and recall ($F = \frac{2P \cdot R}{P+R}$). We compared these metrics for GM12878, H1-hESC, and HFF-hTERT (see Supplementary: Fig. 7,8,9 for more details).

9) *Why Hi-C-LSTM maps lost CTCF interaction dots of Hi-C maps?*

We thank the reviewer for the comment. The heatmaps shown in the paper do not fully capture the average model behaviour around CTCF enriched loop boundaries. In order to check whether our model captures CTCF interaction dots on average, we averaged the observed and predicted contacts at all loop domain corners. We see that Hi-C-LSTM is not able to distinctly capture the CTCF interaction dots because of its sequential prediction streaks (Changes (95) and (96)).

Changes made to the paper:

- (95) Figure 31 (Supplementary: Fig. 11) has been added to the supplementary, talking about CTCF interaction dots at loop domain boundaries.

Figure 31: Hi-C-LSTM predictions near CTCF interaction dots. The edges of most domains are usually dotted by the CTCF transcription factor binding and holding the domains together. This results in increased contacts at the very edges of domains compared to their vicinity. **a)** We show the average contact strength at the edges of medium sized domains (300Kbp) and their vicinity in the observed Hi-C. We go up to 50Kbp within the domain and show 50Kbp upstream on both sides. The CTCF interaction dots are visible at the domain edges. **b)** However, the same is not observed for predicted contacts from Hi-C-LSTM. We see that the domain edges are a bit smudged because of the sequential streaks in the Hi-C-LSTM predictions, but the model is able to capture the distinction between the contacts within and outside the domain. Although we lose CTCF interaction dots in the reconstructed output, we are able to accurately predict domain boundaries and CTCF binding sites using Hi-C-LSTM representations.

(96) The following text has been added to section Results (classification) of the revised manuscript, regarding CTCF interaction dots.

One caveat of the model is that it loses CTCF interaction dots at loop boundaries because of its sequential prediction streaks (Supplementary: Fig. 11).

References

- [1] Heil, B. J., Hoffman, M. M., Markowetz, F., Lee, S. I., Greene, C. S. & Hicks, S. C. Reproducibility standards for machine learning in the life sciences. *Nature Methods* , **18**, 1132-1135 (2021).
- [2] Ramani, V. et al. Massively multiplex single-cell Hi-C. *Nature methods* **14**, 263-266 (2017).
- [3] Rao, S. S. et al. A 3D map of the human genome at kilobase resolution reveals principles of chromatin looping. *Cell* **159**, 1665-80 (2014).
- [4] Hnisz, D. et al. Activation of proto-oncogenes by disruption of chromosome neighborhoods. *Science* **351**, 1454-1458 (2016).
- [5] Melo, U. S. et al. Hi-C identifies complex genomic rearrangements and TAD-shuffling in developmental diseases. *The American Journal of Human Genetics* **106**, 872-884 (2020).

Reviewers' Comments:

Reviewer #1:

Remarks to the Author:

In the revision, the authors have tried their best to answer all the questions and comments. I feel the manuscript has been substantially improved. I do not have further questions at this stage.

Reviewer #2:

Remarks to the Author:

In the new revised manuscript, the authors thoroughly addressed the comments that this reviewer suggested for improving the quality of the study.

However, some comments need to be corrected, after which the current manuscript can be positively considered for acceptance.

Major:

In lines 307 and 308 of the text, the authors claim, "We find that ZNF143 binds directly to promoters and contributes to chromatin interactions that connect promoters to distal regulatory enhancers."

This finding does not appear to be a result of the study. The authors should support that statement with evidence from their study.

Minor:

- While citing the panels in Figure 8 there are several errors.

Figure 8b is not cited. It appears that it should be placed on line 409 instead of Fig. 8c.

The text refers to Fig. 8d, that panel 'd' does not exist in Fig 8.

- I suggest revising the citation order of the main and supplementary figures, as well as the citation order of the panel within each figure. It is always good to find that each piece of graphic support follows a consecutive order.

Reviewer #4:

Remarks to the Author:

The authors proposed a recurrent long short-term memory (LSTM) neural network model to identify genomic drivers of 3D genome by producing low-dimensional latent representations that summarize intra-chromosomal Hi-C contacts. The authors have addressed most concerns of reviewers3, but I still have several concerns.

1) For the second comments of reviewer3, I think Hi-C data in the article, Acute depletion of CTCF directly affects MYC regulation through loss of enhancer-promoter looping (GSE138862), could be applied to validate the accuracy of Hi-C-LSTM in predicting the effects of genomic perturbation.

2) Many methods have been proposed to identify genomic features that influence 3D genome, such as multiple logistic regression, enrichment test and non-parametric models. What's the advantage of LSTM-based method than previous methods?

3) If users want to apply Hi-C-LSTM on their own Hi-C data, whether retraining is required?

4) The axis title of Figure 4d was covered and there was a little black line between Figure 4a and Figure 4b.

The authors would like to sincerely thank the reviewers for providing constructive suggestions to improve the overall quality of the manuscript. We have revised the manuscript to address each comment raised by the reviewers.

The revised manuscript is provided as a separate attachment. Text highlighted in blue colour in the revised manuscript refers to all changes that we made from the original revised submission. Item-by-item responses to all the reviewer comments on our original revised submission are provided below. For convenience, the reviewers' comments are given in italics font and our response is given in regular font.

Responses to the Comments from Reviewer 1

In the revision, the authors have tried their best to answer all the questions and comments. I feel the manuscript has been substantially improved. I do not have further questions at this stage.

We thank the reviewer for all their comments. The reviewers comments encouraged us to look into new experiments and improve the quality of the manuscript.

Responses to the Comments from Reviewer 2

In the new revised manuscript, the authors thoroughly addressed the comments that this reviewer suggested for improving the quality of the study. However, some comments need to be corrected, after which the current manuscript can be positively considered for acceptance.

Major Comments:

1) In lines 307 and 308 of the text, the authors claim, "We find that ZNF143 binds directly to promoters and contributes to chromatin interactions that connect promoters to distal regulatory enhancers." This finding does not appear to be a result of the study. The authors should support that statement with evidence from their study.

We thank the reviewer for this out. We have now fixed this wording in the revised manuscript.

Changes made to the paper:

(1) The following sentences have been altered in section Results (Feature Attribution) of the revised manuscript.

ZNF143 (2nd-most important) is a C2H2-ZF protein. It is known to bind directly to promoters, connect promoters to distal regulatory enhancers [61], and plays a partner role in establishing conserved chromatin loops [61]. Similarly, many FOXG1 (3rd-most important) and related TFs are considered pioneer factors which open closed chromatin and facilitate the binding of other TFs [62, 63]. The last two TFs in our top 5, SOX2 and XBP1, are also known to play a role in conformation. SOX2 loss is seen to decrease chromatin interactivity genome-wide [64], and the genomic distribution of XBP1 peaks shows that it binds promoters and potential enhancers [65, 66].

2) While citing the panels in Figure 8 there are several errors. Figure 8b is not cited. It appears that it should be placed on line 409 instead of Fig. 8c. The text refers to Fig. 8d, that panel 'd' does not exist in Fig 8.

We thank the reviewer for pointing this out. We have now fixed this in the revised manuscript.

3) I suggest revising the citation order of the main and supplementary figures, as well as the citation order of the panel within each figure. It is always good to find that each piece of graphic support follows a consecutive order.

We thank the reviewer for pointing this out. We have now revised the order of figures according to their appearance in the text in the revised manuscript.

Responses to the Comments from Reviewer 3/4

The authors proposed a recurrent long short-term memory (LSTM) neural network model to identify genomic drivers of 3D genome by producing low-dimensional latent representations that summarize intra-chromosomal Hi-C contacts. The authors have addressed most concerns of reviewers3, but I still have several concerns.

1) For the second comments of reviewer3, I think Hi-C data in the article, Acute depletion of CTCF directly affects MYC regulation through loss of enhancer–promoter looping (GSE138862), could be applied to validate the accuracy of Hi-C-LSTM in predicting the effects of genomic perturbation.

We thank the reviewer for their comment. The study the reviewer mentions observes effects in cells after depleting the CTCF protein. As discussed in the manuscript, Hi-C-LSTM cannot model effects of protein depletion (trans-effects) but can only model effects caused by genetic variants on the same DNA molecule (cis-effects). To clarify this point, we have now added additional discussion on this in the Introduction of the revised manuscript.

Changes made to the paper:

(2) The following text has been added to Results of the revised manuscript, talking about variant effects Hi-C-LSTM can simulate.

Note that Hi-C-LSTM can simulate only cis effects such as structural variants, but not trans effects that arise from loss of diffusible entities such as transcription factors.

2) Many methods have been proposed to identify genomic features that influence 3D genome, such as multiple logistic regression, enrichment test and non-parametric models. What's the advantage of LSTM-based method than previous methods?

We thank the reviewer for their comment. We agree with the reviewer that there are other classes of methods that identify genomic features influencing the 3D genome, however, there are some key differences. We now highlight these differences in the related work section of the revised manuscript.

Changes made to the paper:

(3) The following text has been added to section Related Work of the revised manuscript, comparing Hi-C-LSTM with existing methods that identify genomic features influencing the 3D genome. *In addition, many approaches have been developed to identify genomic features, such as histone modifications or other ChIP-seq measurements, that influence chromatin conformation [35-40]. This task is similar to conformation representation learning in that it links 1D to 3D genomic features. However, using histone modifications as a summary of the chromatin-defining features of a given locus may not fully encapsulate the conformation.*

3) If users want to apply Hi-C-LSTM on their own Hi-C data, whether retraining is required?

We thank the reviewer for their comment. Depending on the amount of new data the users have, they can either use transfer learning or train the model from scratch. We have now added more discussion related to this in the revised manuscript.

Changes made to the paper:

- (4) The following text has been added to section Discussion of the revised manuscript, elaborating on transfer learning and training on fresh data.

Users can use transfer learning to apply Hi-C-LSTM to new data sets by fine-tuning the pre-trained model on the new data sets. However, if the amount of new data available is large, it may be preferable to train a fresh Hi-C-LSTM model.

4) *The axis title of Figure 4d was covered and there was a little black line between Figure 4a and Figure 4b.*

We thank the reviewer for pointing this out. We have now fixed this in the revised manuscript.

Reviewers' Comments:

Reviewer #4:

Remarks to the Author:

The authors have addressed all my previous concerns. I suggest to accept this article.